# Ice cliff contribution to the tongue-wide ablation of Changri Nup Glacier, Nepal, Central Himalaya

Fanny Brun[1,2], Patrick Wagnon[1,3], Etienne Berthier[2], Joseph M. Shea[3,4,5], Walter W. Immerzeel[6], Philip D.A. Kraaijenbrink[6], Christian Vincent[1], Camille Reverchon[1], Dibas Shrestha[7], and Yves Arnaud[1]

[1]Univ. Grenoble Alpes, CNRS, IRD, Grenoble INP, IGE, F-38000 Grenoble, France
[2]LEGOS, Université de Toulouse, CNES, CNRS, IRD, UPS, F-31400 Toulouse, France
[3]International Center for Integrated Mountain Development, Kathmandu, Nepal
[4]Center for Hydrology, University of Saskatchewan, Saskatoon, Canada
[5]Geography Program, University of Northern British Columbia, Prince George, Canada
[6]Department of Physical Geography, Faculty of Geosciences, Utrecht University, Utrecht, the Netherlands
[7]Central Department of Hydrology and Meteorology, Tribhuvan University, Kathmandu, Nepal

**Correspondence:** Fanny Brun (fanny.brun@univ-grenoble-alpes.fr)

**Abstract.** Ice cliff backwasting on debris-covered glaciers is recognized as an important process, potentially responsible for the so-called "debris-cover anomaly", i.e. the fact that debris-covered and debris-free glacier tongues appear to have similar thinning rates in Himalaya. In this study, we assess the total contribution of ice cliff backwasting to the net ablation of the tongue of the Changri Nup Glacier over two years. Detailed terrestrial photogrammetry surveys were conducted on select ice cliffs in November 2015 and 2016, and the entire glacier tongue was surveyed with unmanned air vehicles (UAVs) and Pléiades tri-stereo imagery in November 2015, November 2016, and November 2017. The total difference between the volume loss from ice cliffs measured with the terrestrial photogrammetry, considered as the reference data, and the UAV and Pléiades was less than 3 % and 7 %, respectively, demonstrating the ability of these datasets to measure volume loss from ice cliffs. For the period November 2015–November 2016 (resp. November 2016–November 2017), using UAV and Pléiades over the entire glacier tongue, we found that ice cliffs, which cover  7 % (resp. 8 %) of the map view area, contribute to 23 $\pm$ 5 % (resp. 24 $\pm$ 5 %) of the total net ablation of Changri Nup Glacier tongue. Ice cliffs have a net ablation rate 3.1 $\pm$ 0.6 (resp. 3.0 $\pm$ 0.6) times higher than the average glacier tongue surface. However, on Changri Nup Glacier, ice cliffs cannot compensate for the reduction of ablation due to debris-cover. Reduced ablation and lower emergence velocities on debris-covered glacier tongues could be responsible for the debris-cover anomaly.

## 1 Introduction

Ablation areas in High Mountain Asia (HMA) are heavily debris-covered, meaning that a potentially large part of melt water originates from ice ablation of debris-covered glacier tongues (Kraaijenbrink et al., 2017). Numerous studies have demonstrated that a debris layer thicker than 5–10 cm has a dominant insulating effect and dampens the ablation of ice beneath it (e.g., Østrem, 1959; Nicholson and Benn, 2006; Reid and Brock, 2010; Reznichenko et al., 2010; Lejeune et al., 2013). Yet counter-intuitively, similar thinning rates (change in glacier surface elevation over time) have been observed for clean ice and debris-covered ice

at similar elevations across HMA (Gardelle et al., 2013; Kääb et al., 2012), in the Khumbu region (Nuimura et al., 2012), the Kangri Karpo Mountains (Wu et al., 2018), for the Kanchenjunga Glacier (Lamsal et al., 2017) and the Siachen Glacier (Agarwal et al., 2017). This has been referred to as the "debris-cover anomaly" (Pellicciotti et al., 2015).

Two main hypotheses have been proposed to explain this anomaly. First, while ablation rates are reduced by thick debris, ice cliffs could be "hot spots" of ablation and thus contribute disproportionally to the tongue-averaged ablation (Sakai et al., 1998, 2002; Reid and Brock, 2014; Immerzeel et al., 2014; Pellicciotti et al., 2015; Steiner et al., 2015; Buri et al., 2016a). Additionally, other processes linked to supraglacial and englacial water systems could lead to substantial ablation (e.g., Sakai et al., 2000; Miles et al., 2016; Benn et al., 2017; Watson et al., 2018). Second, debris-covered tongues could have a lower emergence velocity compared with debris-free tongues (Anderson and Anderson, 2016; Banerjee, 2017). As a result, even though debris-covered tongues could have less negative surface mass balance compared to clean ice glaciers, their thinning rates (surface mass balance rate minus emergence velocity) would be similar.

In order to partially test the first hypothesis, there is a need to calculate the total contribution of the additional melt processes to the tongue-wide surface mass balance. In this work, we focus on the ice cliff contributions, as the processes related to the glacial water system are currently not quantifiable at the scale of a glacier tongue. For simplicity, hereafter we use the term *net ablation* instead of surface mass balance as we focus only on the ablation areas. We introduce the variable $f_C$, defined as the spatially integrated ratio between the net ablation from all ice cliffs and the glacier tongue net ablation, to quantify the enhanced ablation due to the presence of ice cliffs:

$$f_C = \frac{\dot{b}_C}{\dot{b}_T} = \frac{\Delta V_C}{A_C} \times \frac{A_T}{\Delta V_T} \tag{1}$$

where $\dot{b}$ is the net ablation, $\Delta V$ is the volume loss and $A$ is the area, in each case the subscript refers to the cliffs ($C$) or the glacier tongue ($T$). Additionally, we define the quantity $f_C^*$, which is the spatially integrated ratio between net ablation from all ice cliffs, and the net ablation on all non-cliff areas on the glacier tongue (noted with the subscript $NC$):

$$f_C^* = \frac{\dot{b}_C}{\dot{b}_{NC}} = \frac{\Delta V_C}{A_C} \times \frac{A_{NC}}{\Delta V_{NC}} = \frac{\Delta V_C}{A_C} \times \frac{A_T - A_C}{\Delta V_T - \Delta V_C} = f_C \frac{\Delta V_T}{\Delta V_T - \Delta V_C} \frac{A_T - A_C}{A_T} \tag{2}$$

$f_C^*$ has the advantage of not including the ice cliff contributions in the total tongue ablation, it is thus useful for modeling studies where sub-debris and cliff ablation are estimated independently or in order to scale the ice cliff ablation from the sub-debris ablation. $f_C$ has the advantage of being directly linked to the total ice cliff contributions to ablation. $f_C^*$ is expected to be larger than $f_C$, and both terms refer to a glacier-wide value.

Most previous attempts to estimate the value of $f_C$ were based on modelling and found values of ~6 (Reid and Brock, 2014), ~10 (Sakai et al., 2000), and around 14 (Buri et al., 2016b). Fewer studies assessed $f_C^*$ and found values of ~7 (Juen et al., 2014) and ~12 (Sakai et al., 2000). Two studies have quantified $f_C$ using direct observations: Brun et al. (2016) found $f_C = 6$ over Lirung Glacier by extrapolating volume losses measured from very high resolution photogrammetry on a limited number of cliffs and Thompson et al. (2016) found a value of 8 by digital elevation model (DEM) differencing at Ngozumpa Glacier in the Nepalese Himalaya.

The emergence velocity ($w_e$) of ice for debris-covered glaciers has been found to be significantly different from zero for some cases, but it has been neglected in the calculation of $f_C$, for all the above-mentioned studies. Values of $w_e$ equal to 5.1–5.9 $\pm$ 0.28 m a$^{-1}$ (Nuimura et al., 2011), 0.41 $\pm$ 0.05 m a$^{-1}$ (Vincent et al., 2016) and 0.00–0.35 $\pm$ 0.10 m a$^{-1}$ (Nuimura et al., 2017) have been found for, respectively, the debris-covered tongues of Khumbu, Changri Nup and Lirung glaciers in Nepal. However, we stress the fact that these emergence velocities have been measured at different locations of these debris-covered tongues (in particular close to the clean ice/debris transition on Khumbu Glacier), on glaciers with very different dynamics. Neglecting the emergence velocities (i.e., comparing thinning rates instead of ablation rates) introduces a systematic overestimation of $f_C$. This is due to the fact that cliffs ablate at higher rates than the rest of the glacier tongue: ice cliff thinning rates are thus less influenced than the thinning rates of debris-covered ice when neglecting the emergence velocity. As a consequence, the ratio of the cliff thinning rate divided by the mean tongue thinning rate will overestimate $f_C$. To correctly estimate $f_C$ and the fraction of total ice cliff net ablation, thinning rates need to be corrected with the emergence velocity.

Recent studies advocate the use of terrestrial photogrammetry to understand patterns of ice cliff retreat (e.g., Watson et al., 2017). Nevertheless, these data can only be collected in the field with some difficulty, and thus can only be acquired on a limited number of cliffs. Remote sensing platforms (unmanned aerial vehicles [UAVs], satellites) offer the potential to provide high resolution topographic data with a glacier-wide or region-wide coverage but have not yet been evaluated for detailed multi-temporal monitoring of ice cliffs. Here we test the possibility to use gridded elevation data (i.e. DEMs) obtained from both UAV and Pléiades imagery to assess the total ice cliff contribution to the tongue-wide net ablation.

In this study, we use three very high resolution topographic datasets based on terrestrial photogrammetry, UAV imagery, and Pléiades imagery collected over the tongue of Changri Nup Glacier, Nepal between 2015 and 2017. From the terrestrial photogrammetry, 3D models of 12 cliffs are created to calculate reference ice cliff volume losses from 2015 to 2016. We introduce a new method based on DEM differencing, which takes into account geometric changes induced by glacier flow, and in particular by emergence velocity, and apply it to the UAV and Pléiades imagery. The new method is validated with the terrestrial photogrammetric estimates and applied to the entire Changri Nup Glacier tongue in order to evaluate the fraction of the tongue-wide net ablation due to ice cliffs.

## 2   Study area

This study focuses on the debris-covered part of the tongue of the Changri Nup Glacier, Everest region, Nepal (Fig. 1). The glacier accumulates partly through avalanche deposition from the surrounding steep slopes (up to ∼6700 m a.s.l.) and flows down to 5250 m a.s.l. The local equilibrium line altitude (ELA) was evaluated around 5600 m for the nearby debris-free West Changri Nup Glacier (Sherpa et al., 2017). We use the same glacier tongue outline as Vincent et al. (2016), which was derived from a combination of UAV imagery, field measured velocity fields and field expertise. It is different from the outline available in the Randolph Glacier Inventory 6.0 (Pfeffer et al., 2014), which includes the nearby West Changri Nup Glacier (Sherpa et al., 2017). The debris-covered part of the tongue has an area of 1.49 $\pm$ 0.16 km$^2$ (Fig. 1). We focus first on 12 ice cliffs that were ground-surveyed (Table 1 and Fig. 1), before extending the analysis to more than 140 ice cliffs of various sizes (Fig.

1). The map view area of these cliffs were $70 \pm 14 \times 10^3$ m$^2$, $72 \pm 14 \times 10^3$ m$^2$ and $70 \pm 14 \times 10^3$ m$^2$ in November 2015, in November 2016 and in November 2017, respectively (see section 4.4.4 for the uncertainty assessment of the cliff map view areas).

## 3 Data

### 3.1 Terrestrial photogrammetry

We collected terrestrial photographs during two field campaigns: 24–28 November 2015 and 9–12 November 2016. We surveyed a total of 12 cliffs (Table 1) with methods similar to Brun et al. (2016) and Watson et al. (2017). Between 200 and 400 photographs of each cliff were taken from various camera positions using a Canon EOS5D Mark II digital reflex camera with a Canon 50 mm f/2.8 fixed focal length lens (Vincent et al., 2016). For each cliff, we derive point clouds (PCs) and triangulated irregular networks (TINs) with Agisoft Photoscan 1.3.4 professional edition (Agisoft, 2017). In order to align the photographs and georeference the final point clouds and derived products, between 7 and 17 ground control points (GCPs) made of pink fabric were spread around each cliff. GCP positions were surveyed with a Topcon differential global positioning system (DGPS) unit with a precision of ~10 cm. All markers were used as GCPs and therefore no independent markers were available for validation. After optimization of the photographs alignment, the marker residuals were on average 27 cm for the 2015 campaign and 18 cm for the 2016 campaign. The 3D area of the surveyed cliffs ranged from 600 m$^2$ to more than 11 000 m$^2$ (Table 1).

### 3.2 UAV photogrammetry

UAV imagery of Changri Nup Glacier was obtained in November 2015, November 2016, and November 2017 using the Sony Cyber-shot WX DSC-WX220 mounted on the fixed-wing eBee UAV manufactured by senseFly (Table 2). The Structure from Motion (SfM) procedure that we implemented in Agisoft Photoscan Professional version 1.2.6 to process the imagery is equivalent to the procedure used in Vincent et al. (2016) and Kraaijenbrink et al. (2016). From the PCs, we produced a 10 cm resolution orthomosaic and a 20 cm DEM for each year.

In 2015, five separate flights with the eBee were performed to cover the surface of the glacier over the course of 3 days, i.e. 22–24 November. The data was georeferenced using a set of 24 GCPs that were spatially well-distributed and measured using the Topcon DGPS (Fig. S1). Based on 10 additional independent GCPs, the error of the UAV products was determined to be 4 cm horizontal and 10 cm vertical, which is in the range of expected accuracy (Gindraux et al., 2017). For a detailed description of the data processing we refer to Vincent et al. (2016) and Kraaijenbrink et al. (2016).

On 10 November 2016 we surveyed Changri Nup in three successful flights with the eBee UAV. To georeference the 2016 UAV imagery, we distributed a total of 17 markers on the glacier and measured their coordinates with the Topcon DGPS. Unfortunately, due to time constraints, the resulting spatial distribution of the markers was suboptimal (Fig. S1). Using only these markers as GCPs had considerable consequences for processing accuracy, and we therefore defined 16 additional virtual

tie points for which we sampled the coordinates from the November 2015 UAV orthomosaic and DEM (Fig. S1). For the tie points, we selected specific features on boulders that were clearly identifiable on both the 2015 and 2016 image sets. The use of virtual tie points requires stable terrain (Immerzeel et al., 2014), i.e. the coordinates of the features should not change over time. We have therefore only selected points in stable areas in the vicinity of the glacier, which we determined from visual

inspection of the Pléiades orthoimages and DEMs.

In 2017, we performed the same measurements as in 2015 in three separate flights on 23 November, using 30 GCPs (Fig. S1). The residuals, based on 6 independent check points were 10 cm in horizontal and 14 cm in vertical .

### 3.3  Pléiades tri-stereo photogrammetry

Three triplets of Pléiades images were acquired over the study area on 22 November 2015, on 13 November 2016 and on 24

October 2017 (Table 3). The along track angles of the acquisitions gave base to height ratios ensuring suitable stereo capabilities (e.g., Belart et al., 2017). For each acquisition, we derived a 2 m DEM and a 0.5 m orthoimage using the Ames Stereo Pipeline (ASP; Shean et al., 2016) using only the rational polynomial coefficients (RPCs) provided with the imagery (no GCP) and the same processing parameters as Marti et al. (2016). We used the *stereo* routine of ASP to derive one PC from each triplet of images, which was gridded into a single 2 m DEM using the *point2dem* routine. The orthoimages were generated from the

image closest to the nadir using the *mapproject* function and a 2 m resolution DEM, which was gap-filled with 4 and 8 m DEM resolutions derived similarly. This ensured sharp and gap-free images.

Each Pléiades orthoimage was co-registered to the corresponding UAV orthomosaic, by matching boulders on stable ground visually. We check the accuracy of this co-registration by calculating the median displacement on a 2.4 km$^2$ stable area off-glacier. An east to west residual displacement of 0.05 m and a north to south residual displacement of -0.09 m was identified

after co-registration. This absolute co-registration was needed to compare the UAV and Pléiades datasets, but would not be necessary while working with Pléiades data only. In the latter case, the robustness of the Pléiades processing based on RPCs only would be sufficient to co-register the images and DEMs relatively using automatic co-registration methods. Each Pléiades DEM was shifted with the same horizontal displacement as the corresponding orthoimage (Table 3). Automatic co-registration methods applied to the manually-shifted DEMs (Berthier et al., 2007; Nuth and Kääb, 2011) resulted in no improvement of

the standard deviation of elevation changes on stable terrain. Thus, no further horizontal shift was applied. The vertical shift between the two Pléiades DEMs was calculated as the median elevation change on stable terrain and was equal to -7.43 m and -3.31 m for the periods November 2015–November 2016 and November 2016–November 2017, respectively. These vertical offsets are quite large but expected, as the DEMs are derived from the orbital parameters only (Berthier et al., 2014). We corrected these offsets by subtracting them from the elevation difference map. We tested the dependency of the elevation changes

over stable terrain to the slope, aspect and curvature and found no dependency to these parameters (Fig. S2).

For these three datasets, the duration between acquisition dates were 350 to 381 days. All displacements and volumes have been linearly adjusted (divided by the number of days between the acquisition dates and multiplying by the total number of days in a year) to obtain annual velocities and change rates.

### 3.4 Update of existing datasets

We updated two datasets from Vincent et al. (2016): the glacier surface velocity and the cross sectional ice thickness data.

### 3.4.1 Surface velocity fields

Surface velocity fields were derived from the correlation of the Pléiades orthoimages and UAV orthomosaics using COSI-
corr (Leprince et al., 2007). The UAV orthomosaics were resampled to a resolution of 0.5 m to match one of the Pléiades
orthoimages. For both data sets we choose an initial correlation window size of 256 pixels and a final size of 16 pixels
(Kraaijenbrink et al., 2016). The step was set to 16 pixels, leading to a final grid spacing of 8 m.

The raw correlation outputs were filtered to retain pixels with a signal to noise ratio larger than 0.9. We manually removed
pixels at ice cliff locations, as cliff retreat lead to large geometric changes and therefore poor correlation. These outputs were
filtered with a 9×9 pixel window moving median filter and then gap-filled with a bilinear interpolation (Fig. 2). The patterns of
displacement from UAV and Pléiades are in very good agreement. The velocities measured with Pléiades match well with the
field data (ablation stake displacements measured with a DGPS between November 2015 and November 2016), with the notable
exception of a stake located where the velocity gradient is high and for which the correlation between the Pléiades images could
not work due to snow, leading to a poor bilinear interpolation (Fig. S3). Nevertheless, the maximum displacement is lower in
the remote sensing data (around 11 m a$^{-1}$), than the 2011–2015 field data (around 12 m a$^{-1}$; Vincent et al., 2016). This is due
to a slowdown of the glacier observed also in the 2015–2016 field data.

### 3.4.2 Cross section ice thickness

A cross sectional profile of ice thickness has been measured upstream of the debris-covered tongue (Fig. 1) in October 2011,
with a ground penetrating radar (GPR) working at a frequency of 4.2 MHz (Vincent et al., 2016). The original cross-sectional
area was 79 300 m$^2$ in 2011 and 78 200 m$^2$ in 2015 (Vincent et al., 2016). Between November 2015–November 2016 and
November 2016–November 2017, the cross sectional area decreased from $S_{2015-2016}$ = 76 900 m$^2$ to $S_{2016-2017}$ = 76 340 m$^2$
(with $S_{yr1-yr2}$ being the mean cross sectional area between the year 1 and year 2), based on the 0.86 m a$^{-1}$ thinning rate
measured over the November 2015–November 2017 period along the profile. Following Azam et al. (2012), who measured the
ice thickness of Chhota Shigri Glacier (15.48 km$^2$ flowing from 5830 to 4050 m a.s.l. with a maximum measured ice thickness
of ∼270 m) using the same methods, we estimated that the absolute uncertainty on the ice thickness is ± 15 m, which leads to
an uncertainty on the cross sectional area ($\sigma_S$) of ± 10 000 m$^2$, as the length of the cross-section is 670 m.

## 4 Methods

### 4.1 Emergence velocity

The emergence velocity refers to the upward flux of ice relatively to the glacier surface in an Eulerian reference system (Cuffey
and Paterson, 2010). For the case of a glacier in steady-state (i.e., no volume change at the annual scale), the emergence

velocity balances exactly the net ablation for any point of the glacier ablation area (Hooke, 2005). For a glacier out of its steady state (such as Changri Nup Glacier) the thinning rate in the ablation area is the sum of the net ablation and the emergence velocity (Hooke, 2005). On debris-covered glaciers, while the thinning rate is relatively straightforward to measure from DEM differences, for example, the ablation is highly spatially variable and difficult to measure (e.g., Vincent et al., 2016). In order

to evaluate the mean net ablation of Changri Nup Glacier tongue from the thinning rate, we estimate the mean emergence velocity ($w_e$) for the period November 2015–November 2016 and for the period November 2016–November 2017 using the flux gate method of Vincent et al. (2016). As the ice flux at the glacier front is 0, the average emergence velocity downstream of a cross-section can be calculated as the ratio of the ice flux through the cross-section ($\Phi$ in m$^3$ a$^{-1}$), divided by the glacier area downstream of this cross-section ($A_T$ in m$^2$):

$$w_e = \frac{\Phi}{A_T} \tag{3}$$

This method requires an estimate of ice flux through a cross-section of the glacier, and is based here on measurements of ice depth and surface velocity along a profile upstream of the debris-covered tongue (Figs. 1 and 2). The ice flux is the product of the depth-averaged velocity ($\bar{u}$ in m a$^{-1}$) and the cross-sectional area. For the period November 2015–November 2016 (resp. November 2016–November 2017), the glacier slowed down compared with the 2011–2015 period and the centerline velocity

was equal to 10.8 m a$^{-1}$ (resp. 11.1 m a$^{-1}$), leading to an assumed mean surface velocity along the upstream profile of 8.1 $\pm$ 0.6 m a$^{-1}$ (resp. 8.3 $\pm$ 0.6 m a$^{-1}$), as the mean surface velocity along the cross-section is usually 70 to 80 % of the centerline velocity (e.g., Azam et al., 2012; Berthier and Vincent, 2012). We used the relationship between the centerline velocity and the mean velocity, instead of an average of the velocity field along the cross section, because the image correlation was not successful on a relatively large fraction ($\sim$ 30 %) of the cross section. Converting the surface velocity into a depth-averaged

velocity requires assumption of the basal sliding and a flow law (Cuffey and Paterson, 2010). Little is known about the basal conditions of Changri Nup Glacier, but Vincent et al. (2016) assumed a cold base, and therefore no sliding. This leads to $\bar{u}$ being approximated as 80 % of the surface velocity, additionally assuming $n = 3$ in Glen's flow law (Cuffey and Paterson, 2010). As an end-member case, assuming that the motion is entirely by slip implies $\bar{u}$ equals to the surface velocity (Cuffey and Paterson, 2010). Consequently, we followed Vincent et al. (2016) and assumed no basal sliding, but we took the difference

between the two above-mentioned cases as the uncertainty on $\bar{u}$. This leads to $\bar{u} = 6.5 \pm 1.6$ m a$^{-1}$ (resp. 6.6 $\pm$ 1.7 m a$^{-1}$) for the period November 2015–November 2016 (resp. November 2016–November 2017).

Assuming independence for the cross-sectional area ($\sigma_S$) and the depth-averaged velocity ($\sigma_{\bar{u}}$), the uncertainty on the ice flux ($\sigma_\Phi$) can be estimated as:

$$\frac{\sigma_\Phi}{\Phi} = \sqrt{\frac{\sigma_{\bar{u}}^2}{\bar{u}} + \frac{\sigma_S^2}{S}} \tag{4}$$

Given the above mentioned values for the depth-averaged velocity, the cross-sectional area and the associated uncertainties, the relative uncertainty of the ice flux is $\sim$30 %. As a result, for the period November 2015–November 2016 (resp. November 2016–November 2017), the incoming ice flux was thus 499 700 $\pm$ 150 000 m$^3$ a$^{-1}$ (resp. 503 840 $\pm$ 150 000 m$^3$ a$^{-1}$). The glacier tongue area was considered unchanged at 1.49 $\pm$ 0.16 km$^2$, corresponding to $w_e = 0.33 \pm 0.11$ m a$^{-1}$ (resp. 0.34 $\pm$

0.11 m a$^{-1}$). It is notoriously difficult to delineate debris-covered glacier tongues (e.g., Frey et al., 2012). In this case, we assumed an uncertainty in the outline position of $\pm$ 20 m, leading to a relative uncertainty in the glacier area of 11 %, which is higher than the 5 % of Paul et al. (2013). In this case, the uncertainty on the glacier outline is not the main source of uncertainty in $w_e$, but, if we had used automatically delineated outlines, this would be an important source of uncertainty. The updated emergence velocity is $\sim$20 % lower than estimated for the 2011-2015 period (Vincent et al., 2016), due to both the thinning and deceleration of the glacier at the cross-section. As the difference in $w_e$ between November 2015–November 2016 and November 2016–November 2017 is insignificant, we consider $w_e$ to be constant and equal to $w_e = 0.33 \pm 0.11$ m a$^{-1}$ for the rest of this study. It is noteworthy that $w_e$ is likely to be spatially variable, however, we have no means to assess its spatial variability.

## 4.2 Ice cliff backwasting calculation

### 4.2.1 Point cloud deformation

Every point on the glacier surface moves with a horizontal velocity $u_s$, along a surface slope $\alpha$ and is advected upwards following the vertical velocity $w_s$ (Fig. 3; Hooke, 2005; Cuffey and Paterson, 2010):

$$w_s = u_s \tan \alpha + w_e \tag{5}$$

When DEM differencing is applied, observed thinning rates at every point on the glacier surface is a combination of net ablation and displacement caused by glacier flow. In order to measure only the volume loss associated with the net ablation, we deformed the PCs, by displacing its individual points, for the datasets acquired in November 2015 and in November 2016, in order to account for three-dimensional glacier flow between November 2015 and November 2016 and between November 2016 and November 2017, respectively. For the terrestrial photogrammetry and UAV data, we applied these deformations directly to each point of the PCs. For the Pléiades data, we artificially oversampled the DEM on a 0.5 m resolution grid and converted this DEM to a PC, using the *gdal_translate* function. All the points of the PCs were displaced in $x$, $y$ and $z$ direction:

$$
\begin{cases}
x_{t+dt} = x_t + u_{s,x}dt \\
y_{t+dt} = y_t + u_{s,y}dt \\
z_{t+dt} = z_t + w_s dt
\end{cases}
\tag{6}
$$

where $u_{s,x}$ and $u_{s,y}$ are the $x$ and $y$ components of the horizontal velocity, $dt$ is the duration between the two acquisitions and $z$ is the glacier surface elevation.

Even though $w_e$ is likely to be spatially variable, we consider it to be homogeneous over the whole ablation tongue. The horizontal velocity $u_s$ was directly taken from the bilinear interpolation of the Pléiades velocity field (Fig. 2). The term $u_s \tan \alpha$, can be expressed as:

$$u_s \tan \alpha = \frac{z(x + u_{s,x}dt, y + u_{s,y}dt) - z(x, y)}{dt} \tag{7}$$

As the ice flows along the longitudinal gradient instead of the rough local surface slope, we extracted z from a version of the Shuttle Radar Topography Mission (SRTM) DEM smoothed with a Gaussian filter using a 30 pixel kernel size (Fig. S4).

For the Pléiades and UAV data, we then gridded the deformed PCs using the *point2dem* ASP function (Shean et al., 2016) and derived the associated maps of elevation changes (Figs. 4 and 5).

### 4.2.2 Ice cliff volume change from TINs

In order to measure the volume changes due to cliff retreat from the TINs derived from terrestrial photos, we applied the method from Brun et al. (2016) with some methodological improvements. First, the field of displacement was assumed to be homogeneous at the cliff scale in Brun et al. (2016). In this study, we use interpolated values of the local field of displacement with a resolution of 8 m. This would be an important methodological refinement for ice cliffs on fast flowing glaciers with a rotational component, but has minor influence for the cliffs of interest in this study (Fig. S5a). Second, we added more analogous points in the cliff edge triangulation method. Analogous points are points that are assumed to match in the two acquisitions (e.g. the corners of cliffs; Fig. S5b). Brun et al. (2016) discretized the triangulation problem assuming that the final number of points was equal on the upper and on the lower side of the cliff outline (i.e. implicitly assuming that the two corners of the cliffs were the only analogous points). In this study, the operator can choose how many analogous points are needed to link the two cliff outlines. Consequently, the method is now able to handle larger geometry changes than previously, under the assumption that some analogous parts of the cliffs are identifiable on both cliff outlines.

### 4.2.3 Ice cliff volume change from DEMs

We measured the cliff volume change from DEMs simply as the mean elevation change corrected from glacier flow below a cliff mask multiplied by the projected map view area of the mask. The cliff mask was defined as the union of the shapefiles of the cliff outlines, and is called the cliff footprint and noted $A_{2D}$ hereafter. The cliff outlines were manually delineated both on the Pléiades and UAV orthoimages for November 2015, November 2016 and November 2017. For each acquisition, we used deformed outlines of November 2015 and November 2016 cliffs when working with the corresponding deformed DEM difference. We manually edited the cliff mask to make sure we included the terrain along which the cliff retreated. In particular, this implied linking the corners of the cliff outlines of the two acquisitions in many cases (Fig. S5c).

### 4.3 Sources of uncertainty on the ice cliff backwasting

The main sources of uncertainties on the volume loss estimates are (1) the uncertainty on the spatial distribution of the emergence velocity ($\sigma_e$); (2) uncertainties of the horizontal surface displacement ($\sigma_d$); (3) uncertainty introduced by the displacement along the slope ($\sigma_w$); (4) uncertainties of the cliff outlines delineation ($\sigma_m$) and (5) uncertainties of the various representations of the glacier surface in TINs and DEMs ($\sigma_z$). The first, second and third sources of uncertainties are common to the three datasets and the third and fourth ones are specific to each dataset. We assume these five sources of uncertainty to be independent.

### 4.3.1 Emergence velocity

We calculated a mean emergence velocity for the tongue of $0.33 \pm 0.11$ m a$^{-1}$, but as the spatial variability was unknown extreme values of emergence velocities were tested to estimate $\sigma_e$. We choose 0.00 m a$^{-1}$ as a lower limit because the emergence velocity is positive in the ablation area (Hooke, 2005; Cuffey and Paterson, 2010). For a thinning glacier, the net ablation is higher than the emergence velocity (e.g., Hooke, 2005), consequently, the net ablation can be used as a proxy for the upper bound for the emergence velocity. The maximum net ablation measured with stakes within the period 2014–2016 on the tongue of Changri Nup was chosen as an upper limit equal to 2.22 m a$^{-1}$ (Vincent et al., 2016). We tested these values on the terrestrial photogrammetry-based volume change estimate of each cliff (Fig. 6a). Except for cliff 11, the relative volume change that resulted from the test was always below +40 % for an increase in the emergence velocity and -5 % for a decrease in the emergence velocity. Cliff 11 likely exhibits a high sensitivity to the emergence velocity due to its relatively shallow slope and its very small volume loss (Table 1 and S1). The tested range of values of emergence velocities is rather extreme for the case of Changri Nup Glacier, and we therefore assumed that the uncertainty due to the emergence velocity was equal to the median of the relative volume change for an increase in the emergence velocity (23 %). As a consequence, $\sigma_e = 0.23V$, where $V$ is the cliff volume change.

### 4.3.2 Horizontal displacement

The quality of the horizontal surface displacement derived from Pléiades orthoimages was evaluated by comparison with field measurements of the surface displacement. The median of the absolute difference between the 16 field measurements (stakes and marked rocks) and the corresponding Pléiades measurements was 30.8 cm. We therefore assumed that the uncertainty introduced by the horizontal displacement ($\sigma_d$) is 30 cm. The conversion into volumetric uncertainty, $\sigma_d$, was made by multiplying this uncertainty by the cliff 3D area ($A_{3D}$) for the terrestrial photogrammetry and by the cliff footprint area ($A_{2D}$) for the UAV and Pléiades (Table 1).

### 4.3.3 Displacement along the glacier slope

The uncertainty on $u_s \tan \alpha$ depends mostly on the uncertainty on the mean slope of the surrounding glacierized surface (Hooke, 2005). We evaluated kernel sizes of 5 and 60 pixels to filter the SRTM DEM and found respective mean elevation changes on the cliff mask of -0.51 and -0.33 m a$^{-1}$. As these values correspond to relatively sharp and very smooth DEMs, half of the difference between these two values (10 cm) is a good proxy for the uncertainty due to this correction. We converted this uncertainty into a volumetric uncertainty ($\sigma_w$) by multiplying it by the cliff 3D area ($A_{3D}$) for the terrestrial photogrammetry and by the cliff footprint area ($A_{2D}$) for the UAV and Pléiades.

### 4.3.4 Cliff mapping

The uncertainty on the cliff mapping from Pléiades orthoimages was empirically assessed by asking eight different operators (most of the co-authors of this study) to map six cliffs for which we had reference outlines from the terrestrial photogrammetry.

The operators had access to the Pléiades orthoimage of November 2016 and to the corresponding slope map. We calculated a normalized length difference defined as the difference between the area mapped by the operator and the reference area divided by the outline perimeter. The median normalized length difference ranged between -0.7 and 1.7 m, and was on average equal to 0.6 m, meaning that the operators systematically overestimated the cliff area. The mean of the absolute value of the median normalized length difference was 0.8 m, which was used as an estimate for the cliff area delineation uncertainty. We conservatively assumed the same value for the Pléiades orthoimages and UAV orthomosaics, even though it should be lower for the UAV orthomosaics because of their higher resolution. For the terrestrial photogrammetry data, we assumed no uncertainty on the cliff area. The volumetric uncertainty $\sigma_m$ was obtained by multiplying this value by the perimeter of cliff footprint and by the mean elevation change from DEM differences for UAV and Pléiades.

### 4.3.5 Accuracy of the topographic data

The uncertainty on the vertical accuracy of the terrestrial photogrammetry was directly estimated as the mean of the GCPs residual of all cliffs (0.21 m). For the UAV and Pléiades orthoimages we followed the classical assumption of partially correlated errors (Fischer et al., 2015; Rolstad et al., 2009) and therefore $\sigma_z$ is given by:

$$
\sigma_z = \begin{cases} A_{2D}\sigma_{\Delta h}\sqrt{\dfrac{A_{cor}}{5A_{2D}}} & ; A_{2D} \geq A_{cor} \\ A_{2D}\sigma_{\Delta h} & ; A_{2D} < A_{cor} \end{cases}
\tag{8}
$$

where $A_{cor} = \pi L^2$, with $L$ being the decorrelation length and $\sigma_{\Delta h}$ being the normalized median of absolute difference (NMAD; Höhle and Höhle, 2009) of the elevation difference on stable ground. We experimentally determined $L = 150$ m for both the UAV and Pléiades data, even though the spherical model was not fitting the Pléiades semi-variogram very well. We found $\sigma_{\Delta h} = 0.27$ m for the UAV and 0.36 m for Pléiades.

Under the assumption that the different sources of uncertainty are independent, the final uncertainty on the volume estimate $\sigma_V$ is:

$$
\sigma_V = \sqrt{\sigma_e^2 + \sigma_d^2 + \sigma_w^2 + \sigma_m^2 + \sigma_z^2}
\tag{9}
$$

## 5 Results

### 5.1 Comparison of TIN based and DEM based estimates

The volume changes estimated from terrestrial photogrammetry (our reference) and from UAV / Pléiades data are in good agreement and within error bars (Table S2 and Fig. 7). The total volume loss estimated for these twelve cliffs for the period November 2015–November 2016 is 193 453 ± 19 647 m³ a⁻¹ using terrestrial photogrammetry and 188 270 ± 20 417 m³ a⁻¹ and 181 744 ± 19 436 m³ a⁻¹ using UAV and Pléiades, respectively. The total relative difference is therefore -3 % for the UAV and -7 % for Pléiades, which is smaller than the uncertainty on each estimate (∼10 %, calculated as the quadratic

sum of the twelve individual cliff uncertainty estimates, assumed to be independent). The total Pléiades and UAV estimates are lower than the reference estimate, nevertheless, this is probably due to the estimate of the largest cliff (cliff 01), as there is no systematic under estimation of the volume for individual cliffs (Fig. 7).

## 5.2 Sensitivity to the emergence velocity

As Changri Nup Glacier is a slow flowing glacier, the emergence velocity is small and the associated uncertainty is low (Fig. 6a). Nevertheless, with our dataset it is possible to explore more extreme emergence velocities up to 5 m a$^{-1}$, which is a value inferred for a part of the Khumbu Glacier tongue and which is also the maximum emergence velocity measured on a debris-covered tongue, to our knowledge (Nuimura et al., 2011). Our results show that, as a rule of thumb, every 1 m a$^{-1}$ error on the emergence velocity would increase the one-year volume change estimate by 10 % (Fig. 6b). It is noteworthy that the main source of uncertainty on the cliff volume change is the uncertainty on the emergence velocity.

## 5.3 Importance of the glacier flow corrections

In order to check the internal consistency of the glacier flow correction, we calculated the mean glacier tongue net ablation (calculated as the mean rate of elevation change minus the emergence velocity) before and after corrections. For the period November 2015–November 2016, without flow correction the mean tongue net ablation was equal to -1.07 ± 0.27 m a$^{-1}$ and -1.18 ± 0.36 m a$^{-1}$ for the UAV and Pléiades DEM differences, respectively. After the glacier flow correction (Eq. 3), the mean tongue net ablation was equal to -1.10 ± 0.27 m a$^{-1}$ and -1.20 ± 0.36 m a$^{-1}$ for the UAV and Pléiades data, respectively. The very good consistency between each estimate gave confidence in the fact that our glacier flow correction conserves mass. The same consistency was found for the period November 2016–November 2017.

For individual cliffs, the contribution of the glacier flow corrections were small relative to the uncertainties (Fig. 7), except for cliff 11 and 12 that experienced a small volume change. These two cliffs are also located in the fastest flowing part of the glacier tongue. The low magnitude of the glacier flow corrections is a result of (1) the small displacements of most of the cliffs and (2) the vertical displacement due to slope, which tended to compensate for the emergence velocity (Fig. 3). Nevertheless, for the two smallest and fast moving cliffs (cliffs 11 and 12), these corrections were much larger and resulted in improved estimates of volume change for both Pléiades and UAV data (Fig. 7).

## 5.4 Total contribution of ice cliffs to the glacier tongue net ablation for the period November 2015–November 2016

In addition to the 12 cliffs mapped in the field, we manually mapped 132 additional ice cliffs from the Pléiades and UAV orthoimages and slope maps. The total map view cliff footprint area from November 2015 and November 2016 was $113 \pm 21 \times 10^3$ m$^2$, i.e. 7.4 % of the total tongue map view area. Averaged over this cliff mask, the UAV (respectively Pléiades) rate of elevation change corrected from glacier flow and emergence was -3.88 ± 0.27 m a$^{-1}$ (respectively -3.91 ± 0.36 m a$^{-1}$). This corresponds to a total average volume loss at ice cliffs of $440 \pm 54 \times 10^3$ m$^3$ a$^{-1}$.

The three largest cliffs contribute to almost 40 % of the total net ablation from cliffs (Fig. 8). As there is some variability in the rate of cliff thinning, the volume change of each cliff is not always directly related to its area (Figs. 8 and 9). Nevertheless, the largest cliffs dominate the volume loss, as 80 % of the total cliff contribution originates from the 20 largest cliffs in our study and all the cliffs below 2 000 m$^2$ (i.e., the 120 smallest cliffs) contribute to less than 20 % of the total volume loss (Fig. 8).

For the same period the tongue-averaged rate of elevation change was -0.79 $\pm$ 0.21 m a$^{-1}$ (average of the UAV and Pléiades thinning rates) which, after adding the emergence velocity, corresponds to a net glacier tongue ablation of 1.12 $\pm$ 0.21 m a$^{-1}$ or a volume loss of $1.9 \pm 0.2 \times 10^6$ m$^3$ a$^{-1}$. Consequently, the fraction of total net glacier tongue ablation due to cliffs was found to be 23 $\pm$ 5 % for both methods although these cliffs covered only 7.4 % of the tongue area. The factors $f_C$ and $f_C^*$ were thus equal to 3.1 $\pm$ 0.6 and 3.7 $\pm$ 0.7, respectively.

## 5.5 Total contribution of ice cliffs to the glacier tongue net ablation for the period November 2016–November 2017

For the period November 2016–November 2017, we relied on the Pléiades and UAV data only. The cliff footprint area from November 2016 and November 2017 was $120 \pm 21 \times 10^3$ m$^2$, i.e. 7.8 % of the total tongue area. Averaged over this cliff mask, the UAV (respectively Pléiades) rate of elevation change corrected for glacier flow and emergence was -4.76 $\pm$ 0.27 m a$^{-1}$ (respectively -4.43 $\pm$ 0.36 m a$^{-1}$). The average from the Pléiades and UAV data gives a total ice cliff volume loss of $550 \pm 66 \times 10^3$ m$^3$ a$^{-1}$. In the meantime the tongue-average rate of elevation change was -1.18 $\pm$ 0.21 m a$^{-1}$ (average of the UAV and Pléiades thinning rates), corresponding to a net glacier tongue ablation of 1.51 $\pm$ 0.21 m a$^{-1}$, after correction for the emergence, or a total volume loss of $2.3 \pm 0.2 \times 10^6$ m$^3$ a$^{-1}$. Consequently, the ice cliffs contributed to 24 $\pm$ 5 % of the net glacier tongue ablation and the factors $f_C$ and $f_C^*$ were thus equal to 3.0 $\pm$ 0.6 and 3.6 $\pm$ 0.7, respectively.

## 6 Discussion

### 6.1 Cliff evolution and comparison of two years of acquisition

The total ice cliff covered area did not vary significantly from year to year, ranging from $70 \pm 14 \times 10^3$ m$^2$ in November 2015 and 2017 to $71 \pm 14 \times 10^3$ m$^2$ in November 2016. The twelve individual cliffs surveyed showed substantial variations in area within the course of one year, with a maximum increase of 57 % for the large cliff 06 and a decrease of 34 % for cliffs 03 and 09 (Table S2). The total area of these twelve cliffs increased by 8 % in one year. Interestingly, over the same period, Watson et al. (2017) observed only declining ice cliff area on the tongue of Khumbu Glacier ($\sim$6 km away), suggesting a lack of regional consistency. All the large cliffs (most of them are included in the twelve cliffs surveyed with the terrestrial photogrammetry) persisted over two years of survey, including the south or south-west facing ones (Table 1), although south facing cliffs are known to persist less then non south facing ones (Buri and Pelliciotti, 2018). However, we observed the appearance and disappearance of small cliffs, and marginal areas became easier to classify as either ice cliff or debris-covered areas, highlighting the challenge in mapping regions covered by thin debris (e.g., Herreid and Pelliciotti, 2018).

We calculated backwasting rates for the twelve cliffs monitored with terrestrial photogrammetry for the period November 2015–November 2016 (Table 1). The backwasting rate is sensitive to cliff area changes (because it is calculated as the rate of volume change divided by the mean 3D area) and should be interpreted with caution for cliffs that underwent large area changes (e.g., cliffs 01, 02, 03, 06, 09 and 11; Table S2). The backwasting rates ranged from $1.2 \pm 0.4$ to $7.5 \pm 0.6$ m a$^{-1}$, reflecting
the variability in terms of ablation rates among the terrain classified as cliff (Fig. 9). The lowest backwasting rates are observed for cliffs 11 and 12, located on the upper part of the tongue, roughly 100 m higher than the other cliffs (Fig. 1 and Table 1). The largest backwasting rates were observed for cliff 01, which expanded significantly between November 2015 and November 2016. The backwasting rates are lower than those reported by Brun et al. (2016) on Lirung Glacier (Langtang catchment) for the period May 2013–October 2014, which ranged from 6.0 to 8.4 m a$^{-1}$ and lower than those reported by Watson et al.
(2017) on Khumbu Glacier for the period November 2015–October 2016, which ranged from 5.2 to 9.7 m a$^{-1}$ (we reported the values for cliffs which survived over their entire study period only). These differences are likely due to temperature differences between sites. Indeed, the cliffs studied here are at higher elevation (5320–5470 m a.s.l.) than the two other studies (4050–4200 m a.s.l. for Lirung Glacier and 4923–4939 m a.s.l. for Khumbu Glacier).

While a comparison between only two years of data cannot be used to extrapolate our results in time, we note the similarity
between the total ice cliff contribution to net ablation ($23 \pm 5$ % and $24 \pm 5$ % in November 2015–November 2016 and November 2016–November 2017, respectively). In contrast, total net ablation of the Changri Nup Glacier tongue was $\sim$25 % higher for the period November 2016–November 2017 than for the period November 2015–November 2016. While a difference in meteorological conditions between these two years is a likely cause of the greater ablation totals, the ice cliffs seem to contribute a constant share to the total ablation.

## 6.2  Influence of the emergence velocity and glacier flow correction on $f_C$ and $f_C^*$

In most studies quantifying ice cliff ablation (Brun et al., 2016; Thompson et al., 2016), the glacier thinning rate was assumed to be directly equal to the net ablation rate, i.e. emergence velocity was assumed to be zero. If we make the same assumption (but include the corrections for horizontal displacement and the vertical displacement due to the slope), we find a mean thinning rate of $0.80 \pm 0.10$ m a$^{-1}$ for the tongue and of $3.59 \pm 0.17$ m a$^{-1}$ for the cliffs (average of UAV and Pléiades data) for the
period November 2015–November 2016. In this case, the factor $f_C$ would be $4.5 \pm 0.6$ (and $f_C^*$ would be $5.4 \pm 0.7$), which is 50 % higher than the actual value. The cliffs would be found to contribute to $\sim$34 % of the tongue ablation. For the period November 2016–November 2017, the factor $f_C$ would be $3.6 \pm 0.6$ (and $f_C^*$ would be $4.3 \pm 0.7$), which is 20 % higher than the actual value. The cliffs would be found to contribute to $\sim$29 % of the tongue ablation. This might partially explain why previous studies found significantly higher values of $f_C$, and stresses the need to estimate and take into account the ice flow
emergence, even for almost stagnant glacier tongues like Changri Nup Glacier (see Discussion below).

The values of $f_C$ and $f_C^*$ not corrected from the emergence velocity can be compared to the previous observational estimates. Both Brun et al. (2016) and Thompson et al. (2016) found values higher than our estimates. Part of the difference might arise

from the different climatological settings, as Lirung and Ngozumpa glaciers are located at lower elevation than Changri Nup Glacier.

## 6.3 Ice cliff ablation and the debris-cover anomaly

Between November 2011 and November 2015, Vincent et al. (2016) quantified the reduction of area-averaged net ablation over the glacier tongue due to debris-cover. They obtained a tongue-wide net ablation of -1.2 m w.e. a$^{-1}$ and -3.0 m w.e. a$^{-1}$ with and without debris, respectively. As ice cliffs ablate at -3.5 m w.e. a$^{-1}$, $\sim$3.6 times faster than the non-cliff terrain of the debris-covered tongue for the period November 2015–November 2016, and $\sim$1.2 times faster than the tongue if it was entirely debris-free, approximately 75 % of the tongue would have to be covered by ice cliffs to compensate for the lower ablation rate under debris and to achieve the same overall ablation rate as a clean ice glacier under similar conditions. Since ice cliffs typically cover a very limited area (Herreid and Pellicciotti, 2018), it is unlikely that they can enhance the ablation of debris-covered tongues enough to reach the level of ablation of ice-free tongues.

Other ablation-related processes such as supra-glacial ponds (Miles et al., 2016) or englacial ablation (Benn et al., 2012) may contribute to higher ablation rates than what can be expected on the basis of the Østrem curve. Yet the contribution of these processes is not sufficient to enhance the ablation of the debris-covered tongue of Changri Nup Glacier at the level of clean ice ablation, as Vincent et al. (2016) already showed that the insulating effect of debris dominates for this glacier. As a consequence, and based on this case study, we hypothesize that the reason for similar thinning rates over debris-covered and debris-free areas, i.e. the "debris-cover anomaly", is largely related to a reduced emergence velocity compensating for a reduced ablation due to the debris mantle.

This hypothesis currently applies to the Changri Nup Glacier tongue only, and it is unclear if it can be extended to the debris cover anomaly identified at larger scales. The high quality data available for Changri Nup Glacier are not available for other glaciers at the moment, and consequently we provide a theoretical discussion below.

The mass conservation equation (e.g., Cuffey and Paterson, 2010) gives the link between thinning rate ($\overline{\frac{\partial h}{\partial t}}$ in m a$^{-1}$), ablation rate and emergence velocity for a glacier tongue:

$$\overline{\frac{\partial h}{\partial t}} = -\frac{1}{\rho}\dot{b} + \frac{\Phi}{A} \qquad (10)$$

where $\Phi$ (m$^3$ a$^{-1}$) is the ice flux entering in the tongue of area $A$ (m$^2$), $\rho$ is the ice density (kg m$^{-3}$), and $\dot{b}$ is the area-averaged tongue net ablation (kg m$^{-2}$ a$^{-1}$ or m w.e. a$^{-1}$). Consider two glaciers with tongues that are either debris-covered (case 1- referred hereafter as "DC") or debris-free (case 2 – referred hereafter as "DF"), and similar ice fluxes entering at the ELA i.e., $\Phi_{\text{DC}} = \Phi_{\text{DF}}$. The ice flux at the ELA is expected to be driven by accumulation processes, and consequently it is reasonable to assume similarity for both debris-covered and debris-free glaciers. There is a clear link between the glacier tongue area and its mean emergence velocity: the larger the tongue, the lower the emergence velocity. These theoretical considerations have been developed by Banerjee (2017) and Anderson and Anderson (2016), the latter demonstrating that debris-covered glacier lengths could double, depending on the debris effect on ablation in their model. Real-world evidence for such differences in debris-covered and debris-free glacier geometry remain largely qualitative. For instance, Scherler et al. (2011) found lower

accumulation-area ratios for debris-covered than debris-free glaciers. Additionally, based on the data of Kraaijenbrink et al. (2017), we found a negative correlation ($r$ = -0.36, p < 0.01) between the glacier minimum elevation and the percentage of debris cover (Fig. 10). The combination of these two observations hints at both reduced ablation and a larger tongue for debris-covered glaciers.

Consequently, the qualitative picture we can draw is that the ablation area of glaciers with considerable debris-cover is usually larger than for those without ($A_{DC} > A_{DF}$), leading to lower emergence velocity ($w_{e,DC} = \Phi/A_{DC} < \Phi/A_{DF} = w_{e,DF}$). If the glacier is in equilibrium, in both cases, the thinning rate at any elevation is 0, because the emergence velocity compensates the surface mass balance, but with lower magnitudes for both variables ($w_e$ and $\dot{b}$) in case of a debris-covered tongue (Fig. 11). In an unbalanced regime with consistent negative mass balances, as mostly observed in High Mountain Asia (Brun et al.,

2017), similar thinning rates between debris-free and debris-covered tongues could be the combination of reduced emergence velocities and lower ablation roughly summing up to similar thinning rates as debris-free glaciers (Fig. 11). Additionally, there are evidences of slowing down of debris-covered tongues and detachment from their accumulations area, both leading to reduction in ice flux and consequently in $w_e$ (Neckel et al., 2017).

    In conclusion, our field evidence shows that enhanced ice cliff ablation alone could not lead to a similar level of ablation

for debris-covered and debris-free tongues. While we acknowledge the existence of other processes which can substantially increase the ablation of debris-covered tongues, we highlight the potentially important share of the emergence velocity in the explanation of the so-called 'debris-cover anomaly', which partly originates from a confusion between thinning rates and net ablation rates.

### 6.4   Applicability to other glaciers

Determining the total ice cliff contribution to the net ablation of the tongue (i.e., the $f_C$ factor defined in this study) of a single glacier has limited value by itself, because we do not know the glacier-to-glacier variability. In particular, it is too early to conclude if the spread of $f_C$ in the literature reflect the inconsistencies amongst the different methods used or the glacier-to-glacier variability. For instance, the $f_C$ values from models (Sakai et al., 1998; Juen et al., 2014; Buri et al., 2016b; Reid and Brock, 2014) are not directly comparable with the observations (Brun et al., 2016; Thompson et al., 2016), because

they usually require additional assumptions about e.g., the sub-debris ablation or emergence velocity. The definition of debris-covered tongues, the nature of their surface and their hypsometry might have a considerable effect on $f_C$.

    A significant obstacle to applying our method to other glaciers is the need to estimate the emergence velocity, which requires an accurate determination of the ice fluxes entering the glacier tongues. The measurement of ice thickness with GPR systems is already challenging for debris-free glaciers, as it requires to drag emitter, receiver and antennas along transects of the glacier

surface. It is even more challenging for debris-covered glaciers, as the hummocky surface prevents the operators from dragging a sledge. More field campaigns dedicated to ice thickness and velocity measurements (Nuimura et al., 2011, 2017) or the development of airborne ice thickness retrievals through debris are needed, as stressed by the outcome of the Ice Thickness Models Intercomparison eXperiment (Farinotti et al., 2017). The precise retrieval of emergence velocity pattern using a network

of ablation stakes combined with DGPS is a promising alternative, in particular if combined with detailed ice flow modeling (e.g., Gilbert et al., 2016).

## 7 Conclusions

In this study, we estimate the total contribution of ice cliff to the total net ablation of a debris-covered glacier tongue for two consecutive years, taking into account the emergence velocity. Ice cliffs are responsible for 23-24 $\pm$ 5 % of the total net ablation for both years, despite a tongue-wide net ablation approximately 25 % higher for the second year. On Changri Nup Glacier, the fraction of total net ablation from ice cliffs is too low to explain by itself the so-called "debris-cover anomaly". Other contributions, such as ablation from supra-glacial lakes, or even along englacial conduits, are potentially large and have to be quantified, but for the specific case of Changri Nup Glacier they are not large enough to compensate for the reduced ablation (Vincent et al., 2016). Consequently, we hypothesize that the "debris-cover anomaly" could be a result of lower emergence velocities and reduced ablation, which leads to *thinning rates* comparable to those observed on clean ice glaciers. However, ice cliffs are still hot-spots of ablation and consequently of enhanced thinning; without them, the thinning rates of debris-covered and clean ice might not be similar.

Our method requires high-resolution UAV or satellite stereo imagery, and is restricted to glaciers where thickness estimates at a cross section upstream of the debris-covered tongue are available and emergence velocity can be estimated. A comparison of cliff ablation enhancement factor ($f_C$ or $f_C^*$) values calculated for other debris-covered glaciers under our suggested framework would be informative, in order to compare estimates of ice cliff ablation for other and potentially much larger debris-covered tongues. Though our results cover only two years of data, the area occupied by ice cliffs and their relative contribution to ablation ($f_C$) remained almost constant while net ablation totals differed by 25%. The main limitation of our study is its short spatial and temporal extent. It would be very worthwhile to obtain longer-term and multiple sites quantification of the relative ice-cliff contribution to net ablation. Then a compilation of these data would allow to develop empirical relationships for cliff enhanced ablation, which could be included into debris-covered glacier mass balance models.

In line with a previous study (Vincent et al., 2016), we advocate for the abandonment of the term "debris-cover anomaly", which is based on a confusion between thinning rate and net ablation, and we stress the need for more research about the emergence velocity of debris-covered (and nearby debris-free) tongues. Two research directions could be (a) to measure cross sectional ice thicknesses for multiple debris-covered glaciers and (b) to install dense networks of ablation stakes to assess the spatial variability of ice flow emergence.

*Data availability.* Data are available upon request to F.B.

*Author contributions.* F.B., P.W. and E.B. designed the study. F.B. and C.R. processed the terrestrial photogrammetry data, F.B. and E.B. the Pléiades data, P.K., W.I. and F.B. the UAV data. P.W., E.B., C.V., J.S., D.S. and F.B. collected the field data. All authors interpreted the results. F.B. led the writing of the paper and all other co-authors contributed to it.

*Competing interests.* The authors declare no competing interest.

5 *Acknowledgements.* This work has been supported by the French Service d'Observation GLACIOCLIM, the French National Research Agency (ANR) through ANR-13-SENV-0005-04-PRESHINE, and has been supported by a grant from Labex OSUG@2020 (Investissements d'avenir – ANR10 LABX56). This study was carried out within the framework of the Ev-K2-CNR Project in collaboration with the Nepal Academy of Science and Technology as foreseen by the Memorandum of Understanding between Nepal and Italy, and thanks to contributions from the Italian National Research Council, the Italian Ministry of Education, University and Research and the Italian Ministry of Foreign

10 Affairs. Funding for the UAV survey was generously provided by the United Kingdom Department for International Development (DFID) and by the Ministry of Foreign Affairs, Government of Norway. This project has received funding from the European Research Council (ERC) under the European Union's Horizon 2020 research and innovation programme (grant agreement No 676819). E.B. acknowledges support from the French Space Agency (CNES) and the Programme National de Télédétection Spatiale grant PNTS-2016-01. The International Centre for Integrated Mountain Development is funded in part by the governments of Afghanistan, Bangladesh, Bhutan, China, India,

15 Myanmar, Nepal, and Pakistan. The views expressed are those of the authors and do not necessarily reflect their organizations or funding institutions. F.B., P.W., C.V. and Y.A are parts of Labex OSUG@2020 (ANR10 LABX56). We thank the editor and four anonymous reviewers for their constructive comments, which greatly improved this article.

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

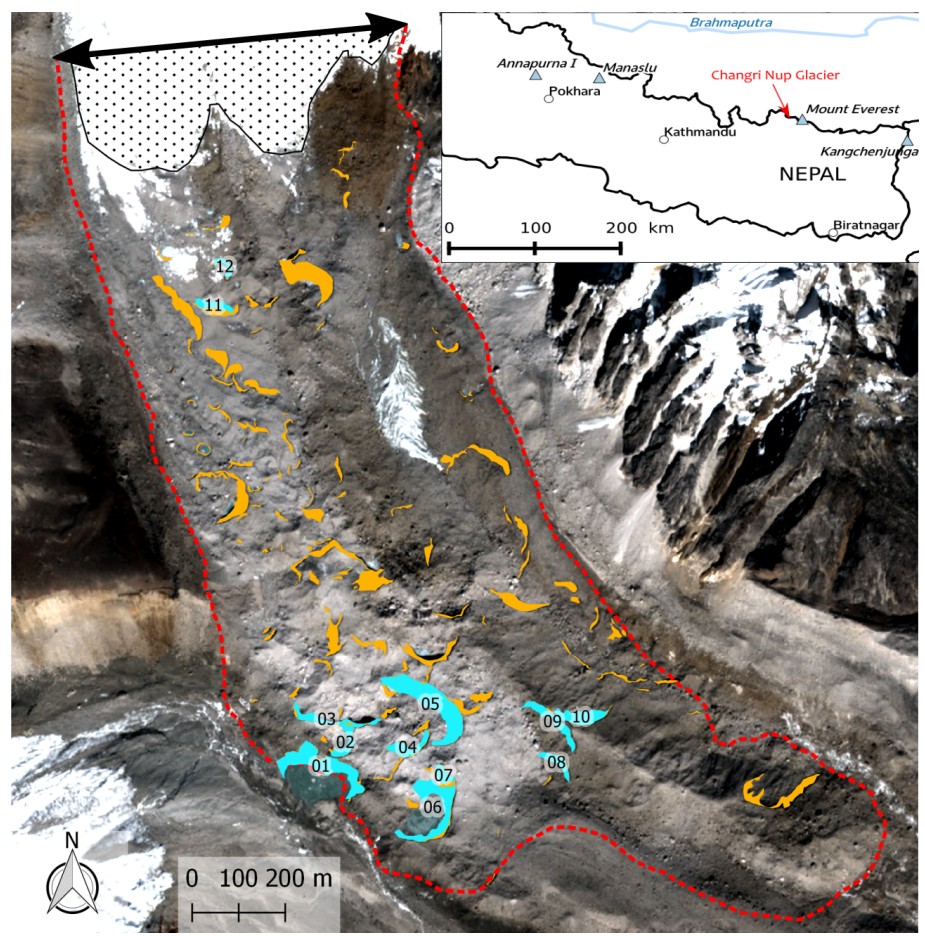

**Figure 1.** Map of Changri Nup Glacier tongue (red outline). The light blue shapes are the twelve cliffs surveyed with the terrestrial photogrammetry and the orange shapes are all the cliffs of the tongue. The background image is the Pléiades images of November 2016 (copyright: CNES 2016, Distribution Airbus D&S). The ice thickness was measured along the black double-headed arrow in 2011 (Vincent et al., 2016). The dotted area is the debris-free part of the tongue (November 2017).

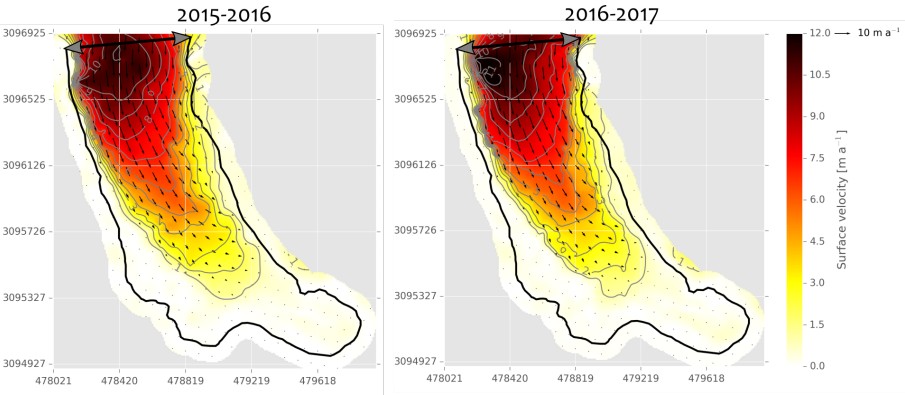

**Figure 2.** Annual horizontal velocity fields deduced from the correlation of Pléiades orthoimages. Coordinates are in UTM 45/WGS 84. The black line is the tongue outline. The missing data in the velocity fields were filled using linear interpolation.

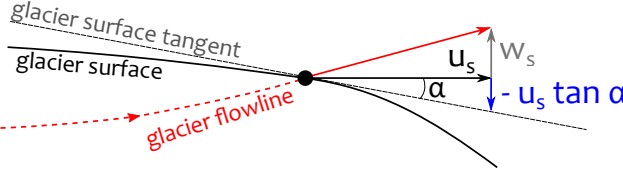

**Figure 3.** Definition of the different flow components, $u_s$ is the horizontal velocity, $w_s$ the vertical velocity and $\alpha$ the angle of the glacier surface tangent; adapted from Hooke (2005).

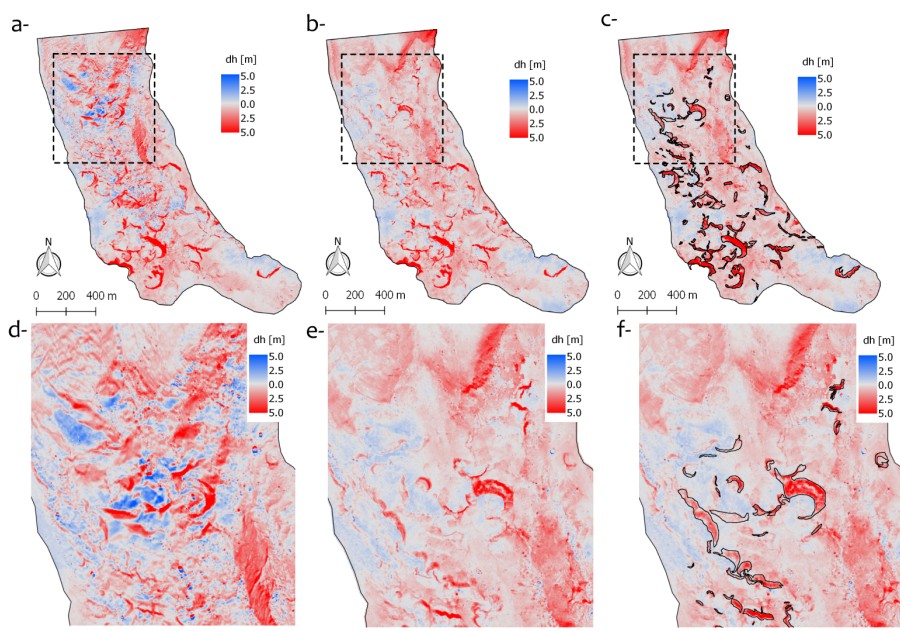

**Figure 4.** Panels showing maps of elevation change from UAV (a, d) before flow correction and (b, c, e, f) after flow correction over the period 23/11/2015–16/11/2016. Black outlines on panels c and f are the cliff footprints. Panels d, e and f are zooms of the panels a, b and c, respectively.

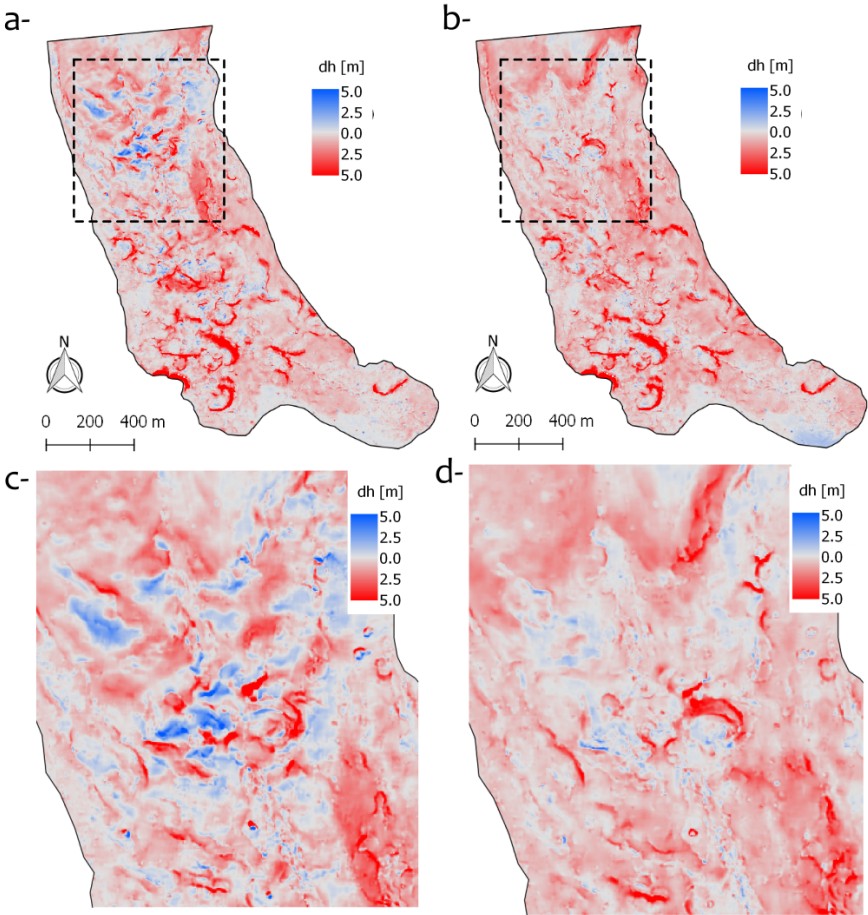

**Figure 5.** Panels showing maps of elevation change from Pléiades (a, c) before flow correction and (b, d) after flow correction over the period 22/11/2015–13/11/2016. Panels c and d are zooms of the panels a and b, respectively.

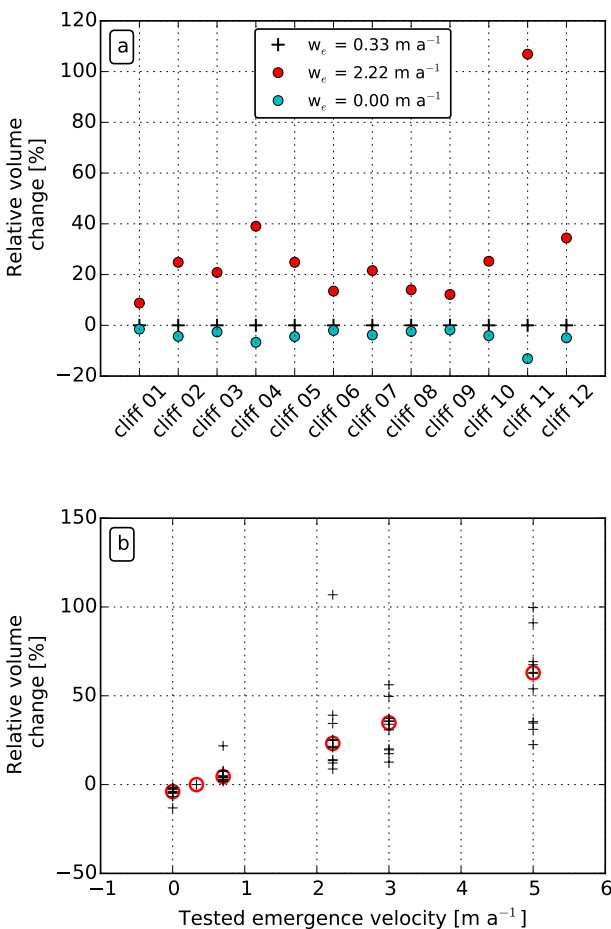

**Figure 6.** Sensitivity of the volume change estimate to the emergence velocity for each cliff with two tested emergence velocities (a) and for all cliffs with various emergence velocities tested (b). The relative volume change is the tested volume change minus the reference volume change (obtained for $w_e = 0.33$ m a$^{-1}$), divided by the reference volume change and multiplied by 100. In the lower panel, each cross represents a cliff and the open circles represent the median, note that cliff 11 relative volume change is not visible for emergence velocities of 3.0 and 5.0 m a$^{-1}$, because it is equal to 153 and 255 %, respectively. The volume estimates are from terrestrial photogrammetry data, for the period November 2015–November 2016.

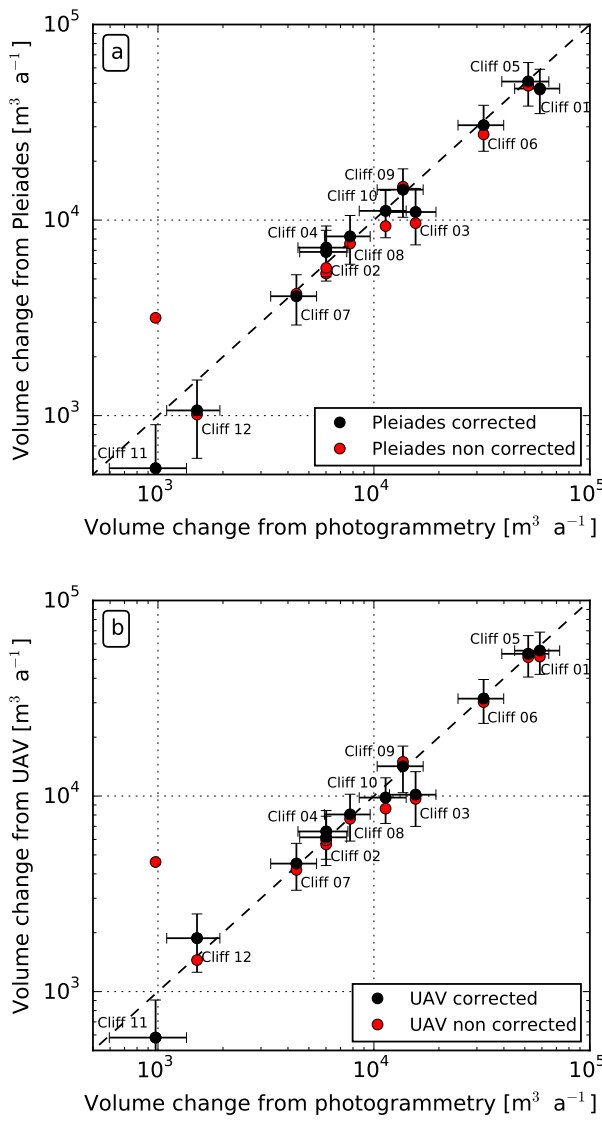

**Figure 7.** Comparison of the ice cliff volume changes estimated from DEM differences between UAV (a) or Pléiades (b) and terrestrial photogrammetry, for the period November 2015–November 2016. Note the log scale. For each panel, "corrected" means taking into account the geometric corrections due to glacier flow and "non corrected" means neglecting them.

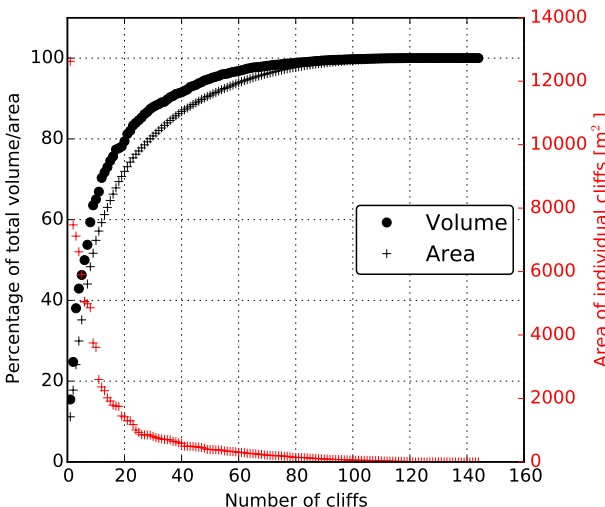

**Figure 8.** Individual ice cliff contributions for the period November 2015–November 2016 based on the UAV data. The left axis shows the cumulative volume (black dots) and area (black crosses), expressed as a percentage of the total volume or area, respectively.

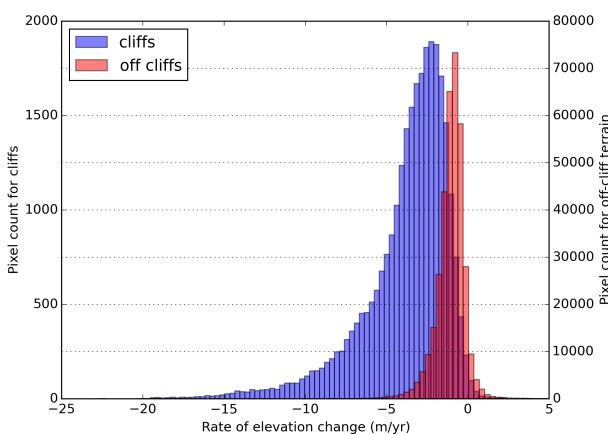

**Figure 9.** Rate of glacier surface elevation change for cliff and off-cliff terrain (Pléiades DEM difference November 2015–November 2016, corrected from flow). Note the strongly different Y axis.

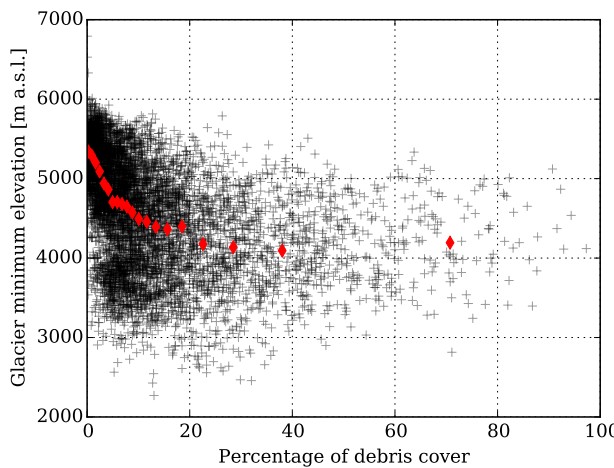

**Figure 10.** Glacier minimum elevation as a function of the percentage of debris cover for the glaciers larger than 2 km$^2$ in High Mountain Asia (6571 glaciers in total). The black crosses represent individual glaciers and the red diamonds shows the mean of the glacier minimum elevation for each five percentile of debris cover. For instance, the first diamond represent the mean of the glacier minimum elevation for glaciers with a percentage of debris cover between 0 (minimum) and 0.51 % (5th percentile).

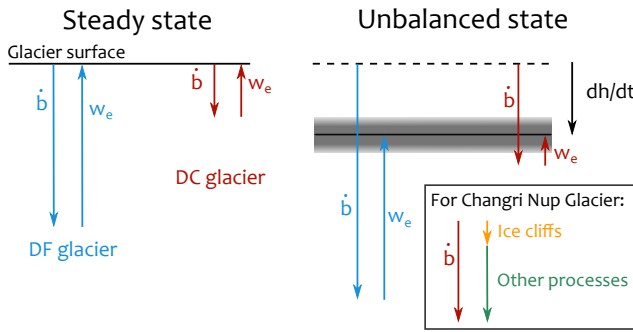

**Figure 11.** Conceptual representation of the interplay of net ablation ($\dot{b}$) and emergence velocity ($w_e$) for debris-free (DF, blue color) and debris-covered (DC, brown color) glacier tongues. In the left panel both glaciers are at equilibrium (no thinning) and in the right panel their tongues are thinning at roughly the same rate $\overline{\partial h / \partial t}$, shown by the grey shaded area. In the unbalanced state, the values are scaled according to Vincent et al. (2016). For the steady state, we assumed a similar emergence velocity for the debris-free tongue. The inset shows the share of the ice cliffs versus the other processes for the tongue-wide ablation on Changri Nup Glacier tongue. It is noteworthy that this representation is only conceptual, that it is based on our current understanding of the interplay of ablation and ice dynamics of a single, small glacier tongue (Changri Nup), and that the emergence velocity values are very poorly constrained.

**Table 1.** Characteristics of the 12 surveyed cliffs. The 3D mean area was calculated as the mean of the November 2015 and 2016 areas, which were measured from the PCs obtained with the terrestrial photogrammetry on CloudCompare. The perimeter was calculated from the cliff footprint of November 2015 and 2016. The backwasting rate was calculated as the ratio between the cliff backwasting volume obtained from terrestrial photogrammetry and the 3D mean area, for the period November 2015–November 2016. The cliffs are usually not perfectly planar and they exhibit multiple aspects. The main aspects were calculated by fitting a plan through the cliff PC or through parts of the PC in CloudCompare, the main aspect is in bold when it was possible to determine it.

| Cliff ID | 3D mean area [m$^2$] | Cliff footprint [m$^2$] | Footprint perimeter [m] | Elevation [m a.s.l.] | Backwasting rate [m a$^{-1}$] | Main aspects (slope [degree]) |
|---|---|---|---|---|---|---|
| Cliff 01 | 7543 | 6575 | 711 | 5330 | 7.5 ± 0.6 | **SW** (44°) / S (46°) / W (39°) / NE (59°) |
| Cliff 02 | 1315 | 1406 | 260 | 5343 | 4.4 ± 0.5 | SW (25°) / NW (29°) |
| Cliff 03 | 3033 | 1821 | 479 | 5347 | 4.9 ± 0.5 | **N** (69°) |
| Cliff 04 | 1851 | 1774 | 286 | 5352 | 3.1 ± 0.4 | **N** (42°) / NW (57°) / E (36°) |
| Cliff 05 | 11294 | 8592 | 607 | 5353 | 4.4 ± 0.5 | **SW** (44°) / NW (51°) |
| Cliff 06 | 5267 | 5064 | 639 | 5331 | 5.9 ± 0.5 | N (60°) / W (52°) / S (45°) / SW (86°) |
| Cliff 07 | 752 | 979 | 153 | 5350 | 5.6 ± 0.5 | **SW** (41°) |
| Cliff 08 | 1282 | 1307 | 227 | 5325 | 5.8 ± 0.5 | S (58°) / SW (59°) |
| Cliff 09 | 2408 | 2263 | 386 | 5350 | 5.4 ± 0.5 | **SW** (60°) / S (46°) |
| Cliff 10 | 2426 | 2521 | 284 | 5338 | 4.5 ± 0.5 | **N** (35°) |
| Cliff 11 | 775 | 630 | 194 | 5452 | 1.2 ± 0.4 | **N** (38°) |
| Cliff 12 | 587 | 653 | 165 | 5464 | 2.5 ± 0.4 | W (58°) / SW (50°) / S (40°) |

**Table 2.** Characteristics of the three UAV flights. The horizontal and vertical residuals are assessed on independent additional GCPs. The virtual GCPs are reference points taken in stable ground from the 2015 UAV DEM and orthomosaic, and used as GCPs to derive the 2016 UAV DEM and orthomosaic. For the 2015 and 2017 campaigns, the GCPs were in sufficient number and consequently we did not use virtual GCPs. For the 2016 campaign, we used all the available GCPs to derive the DEM, and consequently could not evaluate the residuals.

| Date of acquisition | Number of images | Number of GCPs | Number of virtual GCPs | Horizontal residuals (cm) | Vertical residuals (cm) |
|---|---|---|---|---|---|
| 22–24/11/2015 | 582 | 24 | 0 | 4 | 10 |
| 16/11/2016 | 475 | 17 | 16 | N/A | N/A |
| 23/11/2017 | 390 | 30 | 0 | 11 | 14 |

**Table 3.** Characteristics and IDs of the Pléiades images. Horizontal shifts relative to the UAV ortho-images are also given.

| Date of acquisition | Base to height ratio (B/H ) | Shift eastward (m) | Shift northward (m) |
|---|---|---|---|
| 22/11/2015 | 0.36;0.26;0.10 | -4.3 | 0.3 |
| 13/11/2016 | 0.47;0.28; 0.20 | 6.6 | 3.7 |
| 24/10/2017 | 0.34;0.25;0.09 | 1.0 | 4.2 |