# Peer review of "Can ice-cliffs explain the "debris-cover anomaly"? New insights from Changri Nup Glacier, Nepal, Central Himalaya"

_The Cryosphere, 2018_

## Referee Comment (RC1) · Anonymous Referee #1 · 17 Apr 2018

Brun et al combine several state-of-the-art observational datasets with a novel correction for glacier dynamics (based on unique field observations) to measure volume losses due to bare ice cliffs exposed on Changri Nup Glacier in Nepal. This is an important question, as recent studies have suggested that ice cliffs play an important role in bringing the thinning rates of debris covered glaciers to parity with those of clean ice glaciers (unexpectedly). The study finds that ice cliffs indeed account for a disproportionate amount of mass loss in the debris-covered ablation area of Changri Nup, but that emergence velocity has been neglected in assessments of the 'debris covered glacier anomaly'.

[Figure]

I am impressed with the careful processing of the field and remote-sensing observations, in particular with the correction of point clouds for glacier flow and the treatment of uncertainty in general, and I find this study to be an excellent combination of high-resolution topographic datasets and robust processing to measure changes of highly dynamic features. I am particularly pleased to see attention given to emergence velocity, an aspect of glacier dynamics and mass balance that is often neglected in contemporary studies due to the recent emphasis on remote sensing observations. I have concern with the strength of the authors' refutation of the 'debris covered glacier anomaly' based on observations from a single glacier; I rather think they have highlighted the (largely unacknowledged) importance of emergence velocity, but have not demonstrated that this is the dominant or general mechanism by which debris covered glaciers thin at rates comparable to clean ice glaciers in High Mountain Asia. I suggest the authors consider some textual revision in order to better balance the focus of their discussion and conclusions with the focus of their highly sophisticated processing.

Major points:

1. The manuscript is not balanced in terms of the focus of its methods, results, and discussion. The manuscript is mostly aimed at assessing the contribution of ice cliffs to mass balance; the gold-standard methods are targeted specifically to assess this using multiple (perhaps redundant) high-resolution datasets, yet once the authors have a number for the ice cliff net ablation, the discussion is nearly all about the importance of emergence velocity. This feels like an afterthought (i.e. determination of emergence velocity itself is not given much attention in the background and methods, but this is the main topic in the discussion, whereas ice cliffs received little attention); this disparity is awkward. In particular, additional attention needs to be paid to the uncertainty in both the original emergence velocity dataset (per Vincent et al 2016) and particularly with respect to the 'updated' estimate. For example, what about uncertainties in ice thickness retrieval and differences in emergence velocity due to profile orientation? What about the uncertainty of thermal regime and its effect on column-averaged ice velocity?

If emergence velocity is to be a major outcome of the manuscript, its uncertainty needs to be more carefully assessed.

2. I think some adjustment to the title and latter discussion is necessary: I do not think the authors are able to answer the title question using data from Changri Nup alone. First, the authors provide no evidence that Changri Nup fits within the 'debris cover anomaly' framework (that Changri Nup is thinning at a comparable rate to debris-free ice at a similar elevation). This is partly due to the hypsometric differences of debris-covered and debris-free ice in the Solukhumbu region, but this is largely why the debris-cover anomaly has been determined from numerable populations of glaciers, which will exhibit a variety of hypsometric distributions. It could be possible to assess the thinning rates (and melt rates) just below the GPR transect, where debris and ice surfaces exist at the same elevations – does Changri Nup actually show evidence of comparable thinning rates for debris and ice? However, I am doubtful that this would be satisfactory, as Vincent et al (2016) has already demonstrated that melt rates at Changri Nup would be very different beneath debris and clean ice; it seems that the hypsometric parity of thinning rates for debris-covered and debris-free ice does not hold for this particular location, but for larger regions.

Put differently, there is circular logic at play – it is already known that subdebris melt rates are not equivalent to clean-ice melt rates at this location, so no amount of ice cliff melt could bring the subdebris mass balance to the same level. A way forward is to emphasize that both processes are important: neglecting emergence velocity, one does underestimate melt rates, but similarly one does if neglecting ice cliffs. However, emergence velocity has been neglected, and the Changri Nup data is the first field data to demonstrate the effects theorized by Banerjee (2018). Thus, a meaningful question is how much are the competing hypotheses responsible for boosting the thinning rate of debris-covered glaciers? I.e. how much of a boost in lowering is due to cliffs vs how much is because of emergence velocity? Or, how much 'additional' melt would be needed from cliffs to lead to thinning (or b_dot) -equivalence? Twice as much? Three

times? Can you guess how much ablation ponds are responsible for (realising that this is just part of your non-cliff net ablation, and does not affect the role of emergence velocity)?

Minor points

Some nomenclature formality is needed for the cliff area terms. Variably through the manuscript there are 'planar' (cliffs are often considered inclined planes, so this is confusing), '2D' and '3D' areas of cliffs. Please clarify this early on in the manuscript, and ensure consistency.

P1 L20. Suggest 'have been found' in place of 'were found' for correct tense

P2 L5-11. It may be useful to use the same order for the hypotheses here as for the rest of the text, e.g. you first discuss how to test the cliff hypothesis before considering the role of emergence velocity.

P2 L6-8. This is the thesis of this paper (that emergence velocity is a major player), which it supports very well. Here, however, this is an hypothesis – that differences in emergence velocity 'can/could lead to comparable thinning rates despite differences in surface ablation. The two studies referenced are hypothetical, idealised flow-models.

P2 L10. This seems to refer to surface ablation only, yet Sakai et al 2000, Miles et al 2016, and Watson et al 2017 (ESPL) also indicate that ponds could potentially lead to significant internal ablation (which would also contribute to lowering as in Thompson et al 2016).

P2 L12. It follows that you also need to determine the melt contribution of supraglacial ponds in order to resolve this

P2 L17. In the formulation of Equation 1, does the 'tongue' area include or exclude the ice cliff areas? That is, does p compare ice-cliff ablation to the overall surface mass balance, or to the non-cliff ablation? Is this consistent between the studies mentioned?

[Figure]

P2 L24. Please also mention the source data and method for Brun et al 2016 if you are going to for Thompson et al 2016.

P2 L28. Suggest 'positive emergence velocities will increase the ...' as it is more concrete than 'affect'

P3 L5-10. It is necessary to make some mention of your emergence velocity correction in this paragraph.

P3 L28. 'GCPs' should be singular or possessive here.

P4 L15. 'equal' should be 'equivalent'

P5 L3. Incomplete sentence. 'This ensured our study/our analysis to...'

P6 L9. I don't believe the accuracy of this cross-sectional area. The uncertainty with respect to radar velocity in ice alone is greater than the stated value. The stated uncertainty equates to 10cm of uncertainty in ice thickness all along the cross section. Please ensure that your corrected uncertainty is propagated to your uncertainty in emergence velocity as well.

P6 L19. Constant and equal over the lower glacier for both periods of study, you mean. As the flux gate method can only give you a mean emergence velocity for the lower glacier, but please mention how it is expected to vary in space, and how this might affect your results for ice cliffs and for the whole glacier.

P9 L2. Your kernel sizes are with units of pixels, correct?

P11 L10. Can you calculate or estimate the 3D area of these cliffs in order to calculate a mean backwasting rate for comparison to other studies? As the rate of elevation change over a cliff-affected area is heavily influenced by, e.g. their height and slope, the backwasting rate is perhaps easier to compare between studies (or indeed between years, as your 2016-2017 data is quite different).

P11 L18-19. For p, it makes sense to me that the comparison would be cliff area to

non-cliff area, rather than cliff area to the whole area. Please check what prior studies have used for this calculation.

P11 L25. Why the much higher melt rates in 2016-2017?

P12 L8. 'Mean tongue' is not a sensible term. Consider 'relative to the whole tongue'

P12 L10-18. Neglecting the emergence velocity, what portion of the glacier's total ablation would be accounted for by ice cliff melt? Perhaps it would likewise be useful to compare the area-averaged losses due to ice cliffs and emergence velocity – are they of comparable magnitude?

P12 L19. Consider 'the' debris-cover anomaly

P12 L22. This emphasizes the problem with your p calculation – it is not comparing ice cliff to debris, but ice cliff to drbis-and-cliff mixtures. Your values of p will increase with this correction. I.e. total melt due to cliffs was 440000m3 for 2015-2016, and they covered an area of 113000m2. Total melt for the whole glacier was 1,918,000m3 over an area of 1.49 km2. Thus the non-cliff melt was 1478000m3 over an area of 1.377km2. And thus p is 3.6 (20% higher). Can you also calculate what p would be neglecting your emergence velocity estimation (for comparison to the studies mentioned?

P12 L29. This is a very good point, but highlights a key difficulty for the paper. The authors have not demonstrated that the 'debris-cover anomaly' is applicable to Changri Nup at all! That is to say – the authors have not demonstrated that Changri Nup's debris-covered area is indeed thinning at a rate comparable to clean ice glaciers at the same elevation (the point of the debris-cover anomaly). Vincent et al 2016 has already demonstrated that the surface mass balance of Changri Nup is lower than it would be if debris were not present. Here you demonstrate that ice cliffs cannot bring the debris area's mass balance to the same level, but does Changri Nup even fit the debris-cover anomaly in the first place? This is not so problematic for your analyses and paper, but for the generalisation of your results to other areas (P13 L1-2 especially)

P13 L4. I think this section needs to be tidied up with respect to nomenclature, in particular replace 'tongue' with 'ablation area'.

P13 L8. This hypothetical analysis is very worthwhile, but as stated in the text, 'has already been shown by Banerjee (2018)'. Please properly reference that study early in this section (you can state that you provide the first field evidence supporting this hypothesis) and reduce this text accordingly. I recommend that you expand the discussion of the responsibility of reduced emergence velocity vs enhanced ablation (how important are cliffs and ponds for mass balance, then?) or consider more fully how the mass balance and emergence velocity (thus thinning rates) of both systems will continue to evolve. Is the apparent parity of thinning rates a temporary feature in this evolution, or should we expect this to perpetuate?

P13 L22-23. The manuscript has demonstrated that emergence velocities (and the difference between emergence velocity for clean-ice and debris-covered areas) are a key part of the debris-cover anomaly, but the manuscript has certainly not demonstrated that this is always (or even generally!) the reason for the thinning rate parity. Consequently I respectfully but strongly think that your statement should be modified, e.g. 'In conclusion, we have demonstrated that emergence velocity differences are as important as ice cliffs and supraglacial ponds in the calculation of melt rates for debris-covered glaciers, and that the 'debris cover anomaly' is in part due to the confusion of thinning rates and net ablation.'

P13 L24. This section is very out of place with regards to the underlying theme of the manuscript, especially as your discussion up until now focuses on cliffs not being important. I suggest as a segue to emphasize that melt rates are substantially higher than without ice cliffs, and that the primary analysis of the study is thus of benefit for modelling studies (otherwise why automatically delineate cliffs at all?).

P14 L6. Please include a mention of where Brun et al (2016) falls in this spectrum.

P14 L7-10. This is an important consideration that should be expanded upon. Your

analysis including flow correction is without a doubt more sophisticated and 'correct' than prior efforts, but it extremely limited in its transferability because of the field data requirements. While emergence velocity is clearly an important and neglected aspect of studies addressing debris-covered glacier mass balance, it is extremely difficult to assess (and thus also the reliance on overall thinning rates rather than mass balance). It is not enough to say 'more data would be helpful' when you advocate abandonment of an entire train of thought; rather, I think it is important to acknowledge why such data do not already exist (why debris thickness has prevented widespread ice thickness measurement through debris), and to address alternative methods of assessing emergence velocity (e.g. networks of ablation stakes combined with dGPS).

P14 L24. There is no discussion of this point, but I think it would be useful to expand upon (briefly). What do we do with your results? How does this affect models of debris-covered glacier mass balance and/or dynamics?

Figure 6. These are normalised change in the volume change (rather than cliff volume), correct?

Figure 10. I like the simplicity of Figure 10, but it is deceptive in its simplicity (the scales are of course arbitrary). It would be worthwhile to emphasize that this is one hypothetical transient state (another would be to double b_dot for debris free glaciers, the end-member with no increase in w_e in either case). It also would be worthwhile to highlight here the fraction of b_dot due to ice cliffs (the focus of the study), and to emphasize that w_e is the least measured aspect of the chart.

---

## Referee Comment (RC2) · Anonymous Referee #2 · 27 Apr 2018

This study evaluated ablation at ice cliff of Changri, debris-covered glacier tongue using UAV-image and dem (and also Preades). They consider the emergence velocity and also evaluated several kinds of errors carefully. They concluded that recent elevation changes at tongue of Changri Glacier is mainly due to lower emergence velocities, not ablation at ice cliffs. In particular, Figure 4 and 5 are very impressive for me, because it's ideal data to analyze ablation process of debris-covered glaciers (Off course we have to consider distribution of emergence velocity). I think this result can be analyzed for other target. I'm looking forward to read other papers. I have some comments as follows. I hope my comments will help to improve your manuscript.

[Figure]

<Specific comments> Page2 L21 '35 (Sakai et al., 1998)' » Please refer Table 2 in Sakai et al.(2000) p = 256/26=9.8. The value 35 calculated from Sakai et al.(1998) is inaccurate value.

Page 2 L26 'but it has typically been neglected in the calculation of p.' » Which previous study neglected emergence velocity? Please address the references.

Page 2 L26 '5.7–6.4 $\pm$ 3.9 m a$-1$' » 6.4 $\pm$ 3.9 m a$-1$ is the value of mass balance in Nuimura et al.(2011)

Page 2 L28-31 'Emergence velocities will affect the thinning rates of debris-covered ice and ice cliffs equally. But since the cliffs ablate at higher rate, their thinning rate is relatively less influenced than the thinning rate of debris-covered ice. As a consequence, the ratio of the cliff thinning rate divided by the mean tongue thinning rate will overestimate p.' » Those explanation is a little bit ambiguous expression. Please write more clearly.

Page 3 '2 Study area' »There are basic information of study area in Vincent et al.(2016). But, I recommend that ELA around the Changri glacier and altitudes (Max, min) information are necessary, here.

Page 6 L6 I cannot find out the location of cross section in Fig. 1 or 2.

Page 7 section 4.2 and 4.3 Ice cliff is unstable. Sometimes they disappear or newly emerge in one melting season. Are there any ice cliffs diminished or emerged? And you have neglected those ice loss in this study?

Page 8 '4.4.1. Emergence velocity' 'The debris-covered part of the tongue has an area of 1.49 $\pm$ 0.16 km2'(Page3 line 16) The uncertainty is induced assuming that there are 20 m uncertainty in the glacier outline Vincent et al.(2016). I think estimation of the tongue area is difficult. I have never been to the Changri Glacier, therefore, I'm not sure the confidence of glacier outline at the terminus. But, in general, it is difficult to estimate outline of glacier terminus at debris-covered glacier. Then, we have to

measure ice depths at two cross sections, and calculate emergence velocity between the two cross sections to avoid large error due to glacier area estimation. You can discuss.

Page8 L13-14 ' The maximum net ablation measured with stakes within the period 2014–2016 on the tongue of Changri Nup was chosen as an upper limit equal to 2.22 m a−1' » Please explain why you can choose the maximum net ablation measured with stakes can be assumed to be the maximum emergence velocity.

P13 L3-23 and Fig. 10 ãĂĂIn this discussion, you have compared debris-covered and debris-free glaciers in equilibrium and transient (shrinking) regime. But, your target is ice loss at ice cliff. Almost assumptions are based on part of other studies. Further, I cannot accept some assumptions, Ex. ice flux is same at both debris-covered and debris-free glaciers. Usually, debris-covered glaciers are large, and debris-free glaciers is small. Further, each altitude are different. Then, I think we cannot discuss without the observation of debris-free glaciers.

Reference Sakai A, Takeuchi N, Fujita K, Nakawo M (2000) Role of supraglacial ponds in the ablation process of a debris-covered glacier in the Nepal Himalayas. International Association of Hydrological Sciences, Publication No. 264 (Symposium at Seattle 2000 - Debris-covered glacier), 119-130.

---

## Referee Comment (RC3) · Anonymous Referee #3 · 30 Apr 2018

— SUMMARY ————-

Brun et al. present an analysis aiming at determining the actual role of ice cliffs in the so-called "debris-cover anomaly". The analysis in performed on Changri Nup Glacier, Everest region, and combines both in-situ and remote-sensing data for a two-year period.

The results are bold: The debris-cover anomaly - the authors say - does not exist, but is the result of confusion around the concepts of "thinning rates" and "net ablation". The "anomaly" – the authors explain – only comes to happen because past studies failed to account for emergence velocities. In reality, the similar thinning rates observed for

debris-covered and debris-free glaciers are a result of the difference between net ablation and ice flow emergence being coincidentally similar for the two types of glaciers.

As much as I agree with the first part of the interpretation, I do believe that the second part is too weakly backed-up: The authors try to generalize their field-based results in the discussion section, but the result is not convincing. Some general statements (e.g. debris covered glacier have smaller accumulation-area-ratio and are generally smaller) should be better corroborated (inventories for doing that exist by now).

A part from the above, the manuscript has a very high standard: The topic is of high relevance and actuality, the introduction is well written, the text is easy to follow, the technical analysis is clearly done by experts, the relevant literature is cited in an exemplarily manner, and the figures are illustrative.

I am fully convinced that the manuscript will be an important addition to the glaciological literature once it is revised.

— GENERAL COMMENTS ———-

1) Sample size

The fact that all claims are built on one glacier (sample size = 1) is a clear handicap for the general conclusions the authors are aiming at. For making the point that much of the debris-cover anomaly is due to a confusion of concepts, the sample size is not an issue. This can be shown with one single example, and this is really the paper's merit. Where it becomes more difficult is when the authors stat arguing that "[for debris-covered glaciers,] the combination of reduced emergence velocities and lower ablation coincidently sum up to similar thinning rates as [for] debris-free glaciers" (p. 13, L. 19-20). Arguing for a regional-scale "coincidence" seems at least adventurous with the single data-point at hand.

I see two ways of solving this: Either (1) the authors try to get hold of published data that exist for other glaciers, or (2) they refine their theoretical argumentation (Sec.

6.3) and back up some of the as-yet little-supported claims (see next comment) with remote-sensing and inventory data.

2) "Theoretical considerations"

The "theoretical considerations" presented in Sec. 6.3 is the part of the manuscript that I found the least convincing. Unfortunately, it is the crucial one.

The problem is that the arguments seem to be much based on qualitative considerations, whilst the author's point is much focused on quantitative statements. Two examples: (1) "debris-covered glaciers have lower accumulation-area ratios than debris-free glaciers" (p. 13 L. 9). The claim is backed up with a reference (Scherler et al., 2011) but it would be ways more convincing to have some actual numbers corroborating this. These can either be re-presented from the original publication or re-compiled from inventory data and remote sensing products (I'm fully aware that the second option would be much more work-intensive). Ideally, a distribution of AARs would be shown for both debris-free and debris-covered glaciers, and the difference quantified. (2) "the glacier response time of a debris covered glacier is longer compared with a debris-free glacier (Rowan et al., 2015), therefore the clean tongue will shrink faster than the debris-covered tongue, further enhancing the difference between ADC and ADF." (p. 13 L. 15-16). Well, again, although a reference is given, it would be so much more convincing having two distributions shown, and a difference quantified. In this case, however, I'm not even sure whether this is necessary since what actually would matter are the present, actual sizes of the debris-free and debris-covered tongues – and not their response times.

I think that fixing this section is the only major work that is in front of the authors.

3) Introduction

A rather minor issue: The concept of "emergence velocity" is, obviously, of central importance to the paper. Since one of the main conclusions is that there is confusion

around the term, I think it would make much sense to provide a clear definition in the introduction. Some indication on how the quantity is typically calculated from field data (or other types of data) may also be helpful for the one or the other reader.

— LINE-BY-LINE COMMENTS ————-

What follows is a series of line-by-line comments of various nature, ranging from comprehension questions to stylistic corrections and including some specific suggestions for issues the authors may want to think about or change.

P.2 L.14: Maybe a detail but I see a danger of the "p" being referred to as "p-value" at some stage. This would obviously be extremely misleading, since the term is reserved for something very specific in statistics.

P.2 L.15: I was confused by the mixture for plurals ("cliffs") an singulars ("cliff"). At that stage, I even briefly asked myself if "p" was something defined at the cliff-scale (i.e. one "p" for every cliff). Please avoid the confusion by using consistent wording.

P.3, L15: Remove "in the same outline" (there is no danger of misunderstanding that)

P.4, L.7: "using a [not "the"] Structure from Motion algorithm"

P.5, L.13-14: I was wondering whether the relatively large offset determined for stable terrain (-7 or so meters) requires a short comment/explanation?

P.5, L.30: "The velocities measured with Pléiades match well with the field data" –> I may have missed it, but I don't think any in-situ velocity measurements were described so far? (The only reference to such measurements seems to be at P.3 L.14, but I understood that info only to be a side note on how the glacier outline was derived in another publication?)

P.6, L15-16: Can a word be spent in discussing the implication of assuming a homogeneous w_e? That quantity is a distributed field, and I have the impression that assuming a similar w_e for all ice-cliffs that are considered is an important assumption?

Some discussion is found later, but here is where the question arises

P.6, L.26: I'm not sure to understand what "deformed" means in this case. "Deformed" how?

P.7, L.19-20. I found the concept of analogous points somewhat abstract. Would it make sense to provide a figure with a visual example?

P.8, L.2-3: Also in this case, a visualization would probably make it easier to understand what is meant exactly.

P.9, L.26-27: "We experimentally determined L = 150 m for the UAV and L = 150 m for the Pléiades data" –> "We experimentally determined L = 150 m for both the UAV and Pleiades data" (or should one of the two "150" read something else?)

P.11, L.9: The unit of -3.88+/-0.27 is missing

P.11, L.10: Here and elsewhere: In light of the estimated uncertainty, it would make sense to state 440+/-54 x 10ˆ3 m3/a (instead of 439 689 +/- 54 000 m3/a).

P.11, L.26: I'm not following: Is the stated value (1.51+/-0.21 m/a) already corrected for emergence?

P.12, L.7-8: The last part of the sentence is rather involved. Can't you simply say that the cliffs seem to contribute a constant share to the total ablation?

P.12, L.14: Here and below: Consider replacing "original" with "actual".

P.12, L.15: Remove "Doing the same".

P.12, L.17: Replace "emergence velocity" with "ice flow emergence" (saying "the influence of velocity" sounds somewhat odd).

P.12, L.24-25 An alternative (simpler?) wording would be "Since ice cliffs typically cover a very limited area, thus, it is unlikely that they can explain the debris-cover anomaly."

P.12, L.27: Check the wording: "englacial hydrology" is not an "ablation-related processes" (it's rather a "discipline", as e.g. glaciology)

P.12, L.28, sentence starting with ".Yet this does not. . ." –> Split the sentence somewhere; it is very long.

P.13, L.1-23: This is the part that really needs revision.

P.13, L.7: The unit of "density" should be kg/m3 (not m2)

P.13,L.10: Well, the comparison is somewhat "cheated", as it should certainly include areas of the same size.

P.13, L.12: Not entirely sure what "both variables" is referring to. To mass balance and emergence velocity?

P.13, L.20: If I read this correctly, you imply that the similar thinning rates for debris-free and debris-covered glaciers are only observed now, and that this was different in the past and will be different in the future. Is this correct? If so, state that explicitly.

P.13, L.30-31: I don't understand the sentence. Especially the two "in" within the parenthesis create confusion.

P.13, L.32: Why "nevertheless"? What's the logical link to the previous sentence?

P.14, L.2: "i.e. the p factor defined in this study" should live in a parenthesis (the sentence is difficult to understand at the moment). Also try to split the sentence as it is very long.

P.14, L.5: "models [. . .] are not directly comparable with the observations" –> Explain (or at least give a hint) why not.

P.14, L.5: Not sure to understand what you mean with "flow components". Please clarify. In light of the claims provided above, moreover, I'm not sure to understand the latter part of the sentence (the one that "advocates for a more consistent framework"). This may be clearer after revision.

P.14, L.10: As much as I agree with the statement, I don't think it is appropriate making it "yours": Geophysicists are working on that since years, after all.

P.14, L.16: "or englacial conduits" –> That's a very speculative claim, isn't it? I would suggest to flag it as such.

P.14, L.18: "we hypothesize that" –> In the last section (Sec. 6) the claims were stated in a much more decided way. Why this caution here? The text should be coherent in what the level of trust in the results is concerned.

P.14, L.18-19: "the debris-cover anomaly could be a result of lower emergence velocities and reduced ablation" –> This is basically the main claim of the paper. Whether it is suitable of having it in the conclusion section or not very much depends on how convincing Sec. 6 will be after revision.

P.14, L.22: ". . .our suggested framework would inform estimates of ice cliff ablation. . ." –>not sure to understand how "inform" is used here. Can you reword?

P.14, L.24: "[. . . it] is required to include these results into debris-covered glacier mass balance models" –> I'm not entirely sure but it looks like you advocate for mass balance models to include a "p"-factor? If this is actually your message, please be more explicit in saying that.

P.14, L.28-29: If what I understood the message correctly, the sentence could be adjusted to "Two research directions could be (a) to extensively measure ice thicknesses and (b) to install networks of stake measurements to assess the spatial variability of ice flow emergence."

— COMMENTS TO FIGURES ————-

Fig. 1: (a) The red box in the upper-right inset is misleading, since it is not the part enlarged in the main figure. Consider replacing it with a red dot or similar. (b) Last parenthesis of the caption: Why "measured"? (The word can simply be removed.)

Fig. 2: Please tell what coordinates are used. The black line is the same as the red-dashed one in Fig. 1, I guess? (Readers shouldn't be guessing ;-) )

Fig. 3: (a) In the caption, spell out what U_s, w_s, etc are. (b) "Local slope" is misleading; it looks more like the local tangent of the surface (and my guess is that "\alpha = local slope").

Fig. 4+5: (a) Can the colour-bar be stretched and more values be added? At the moment, it is difficult to tell what colour corresponds to, e.g. -2.5 m. (b) What is the meaning of "raw" in "raw elevation change"? (c) Please indicate the time span between the UAV surveys (or the dates of the surveys as such). (d) "from flow for" –> Do you mean "for iceflow from"? (e) For consistency, the last sentence should read "Zoom in the dashed rectangle of panel a (c,d)".

Fig. 6: (a) Please (re-) state what "normalized" means in this case. (b) State the period over which the changes refer to. (c) "In the latter" –> "In panel b" (d) "because it is more than 150%" –> "(since the value is >150%)"

Fig. 7: State the period over which the changes refer to.

Fig. 8: The red markers should be crosses (area), and not dots (volume).

Fig. 10: (a) Please simplify the third sentence (the one starting with "In the transient state,..."). As far as I understand, it simply means that all but the blue w_e are taken from Vincent et al (2016)? (b) "ratio of net ablation" –> Not sure to understand that. A "ratio" is between something and something else, I would say.

Tab. 1: (a) "A_SD" never shows up in the table. Thus, no need of introducing the symbol. (b) "The main aspects" –> Why plural? If there is a share of aspects implied in what the table shows, please state that.

Tab. 2: (a) I don't understand the meaning of the reference. Just remove? (b) Please explain in the caption what "virtual" means. (c) In the caption, provide a hint for why some values are "N/A".

Tab. 3: What is "B/H"? The caption should tell.

[Figure]

---

## Referee Comment (RC4) · Anonymous Referee #4 · 3 May 2018

Summary:

The authors estimate the total ablation associated to supraglacial ice cliff melt on the debris-covered tongue of Changri Nup Glacier, Nepalese Himalaya, based on high-resolution topographical data. They use terrestrial photogrammetry surveys on selected cliffs for validation and UAV- and satellite-imagery for the entire cliff population on that glacier for two consecutive years. They then derive the contribution of ice cliff melt relative to glacier melt on the tongue of Changri Nup Glacier by taking into account emergence velocity and estimate ice cliff melt to be ∼3 times higher than the average melt of the glacier tongue. They conclude that ice cliffs cannot explain the debris-cover

anomaly and that the anomaly in turn could be a result of lower emergence velocities and reduced ablation.

General comments:

The main outcome of this study is that UAV- and especially high-resolution satellite imagery can be used to estimate glacier-wide volume losses associated to ice cliff melt, as the authors showed by a sophisticated analysis of various topographic datasets. The second important conclusion of the study is the fact that emergence velocities have so far not been considered carefully enough in terms of glacier ablation estimates and should be investigated further. However, as much as I appreciate the topic of the paper including all its careful analysis, I think the authors' main conclusion of explaining the debris-cover anomaly with reduced emergence velocities in general is not appropriate and a bit out of the context of the paper, given the limited sample size of just one glacier tongue. I suggest the authors adapt the title accordingly and focus more on the nice outcome of the cliff volume loss estimates at the glacier scale, especially in the discussion of the manuscript. Further, I think the balance between ice cliffs and emergence velocity is not given, as the main part of the paper regarding methods description and results, is mostly about ice cliffs and in contrast the discussion/conclusion part is mostly about emergence velocities. The processing of the ice cliff data is well described in general and the results are elaborated carefully by taking into account uncertainties. I miss a better description of one of the key improvements compared to an earlier study, the correction for distortion. Further, I am not convinced that the calculation of the p-factor is feasible or should be done in a different way. All in all my impression of the paper in terms of quality, writing, and relevance in the context of the actual literature is good and it fits into the scope of The Cryosphere, but I have major comments that should be addressed. If these are addressed, I am sure the paper will be a useful contribution to the glaciological community. See further comments below.

Main issues:

[Figure]

**1) Explanation of "debris-cover anomaly"**

I like that you bring the emergence velocity component into the focus of the analysis and interpretation of debris-covered glaciers. It is clear that this upward movement of parts of the glacier tongue has been neglected so far in most of the studies related to debris-covered glaciers especially. In the case of Changri Nup Glacier the emergence velocity is, based on your observations, very small and thus similar downwasting rates of debris-free and debris-covered glacier surfaces might be explained by this differential emergence. However, I am not convinced by the reasoning of the authors to generalize this result and explain the debris-cover anomaly solely by the confusion of glacier elevation changes and net ablation. It is not clear if Changri Nup Glacier has ablation rates similar to that of debris-free glaciers (of the same elevation range) at all, but this would be the case for the debris-cover anomaly. For sure emergence is an important component and might explain many issues related to this topic. But I think, just based on one single glacier making a general conclusion is too ambitious and therefore the title of the manuscript should be adapted accordingly, away from the debris-cover anomaly more towards the volume loss of the ice cliffs. The study still presents interesting points and provides nice results, such as: estimation of the volume loss associated to ice cliff melt at the glacier scale; incorporating rotational behavior of ice cliffs in volume loss estimates; sophisticated comparison to net ablation by taking into account also glacier emergence.

**2) Calculation of p**

I am not sure if your calculation of the p ratio makes much sense. Like this you compare ice cliff melt to the melt of the entire debris-covered glacier tongue. This means you compare ice cliff melt to a mixture of subdebris- and ice cliff melt (the glacier tongue). Wouldn't it be more feasible to compare ice cliff melt to subdebris melt (i.e, exclude ice cliff areas)? In order to be comparable to the previous studies you cited, you should check if they also compared ice cliff melt to the ice cliff-subdebris mixture or to subdebris melt only. Also, it is not clear how many ice cliffs you detected on the entire

debris-covered glacier tongue in both seasons, does this number vary? Did you observe the formation of new features over time or disappearance of them? This might also affect the p ratio, in case the cliff areas are not excluded in its calculation. Additionally, could you derive melt rates for the cliff surface (perpendicular to their surface) instead of pure elevation change rates? This would be helpful in order to compare your results to previous studies.

3) Correction with local field of displacement

From the text I cannot follow how the rotational component of the ice cliffs/glacier surface are implemented in the calculations of the ice cliff volume changes (Section. 4.2). Since this is a main improvement of this study compared to an earlier one using a similar method, much more emphasis should be given to explain this implementation. How do you use the local field of displacement? Where does the 3D flow field that you use for the correction come from? This is a key comment that should be addressed for the paper clarity. A schematic figure would be helpful. Also, can you estimate how much the difference is compared to the previous method used in Brun et al. 2016? More importantly, can you validate the new method to show that it is appropriate and sound? From the short description provided: i) the method cannot be reproduced; ii) there is no evidence that it is the correct approach. You should discuss this new method in the discussion part of the manuscript, as it seems to be a clear advancement from previous studies, where a simple homogeneous direction of displacement was assumed.

4) Uncertainty of emergence velocity assumptions

As mentioned above in the general comments, the balance between ice cliff associated melt estimates and emergence velocity assumptions is not well elaborated in the manuscript. E.g. I miss a more in depth discussion of the uncertainties related to the calculation of the emergence velocities in the methods section, such as the one associated with having only one cross section profile for glacier thickness estimates, or general uncertainties in ice flux assumptions (glacier thickness measurements, bed topography, subglacial conditions, distribution of emergence etc.). As stated in line 19, p. 10, "the main source of uncertainty on the cliff volume change is the uncertainty on the emergence velocity", but this is not discussed later on in the text. Also, the reduction in emergence velocity of ∼20% compared to the period 2010-2015 (line 16, p. 6) is striking, isn't it? Can you try to explain it more convincingly? If glacier emergence is so (relatively) variable, this might also have implications on your assumption of generally explaining the debris-cover anomaly by lower emergence velocities.

---

## Author Comment (AC1) · 4 Jul 2018

**Response to anonymous referee 1**

We thank anonymous referee 1 for the detailed review. In this response, the reviewer's comments are in black standard font. Our response is in standard blue font and the modifications to the manuscript are in blue bold font.

Brun et al combine several state-of-the-art observational datasets with a novel correction for glacier dynamics (based on unique field observations) to measure volume losses due to bare ice cliffs exposed on Changri Nup Glacier in Nepal. This is an important question, as recent studies have suggested that ice cliffs play an important role in bringing the thinning rates of debris covered glaciers to parity with those of clean ice glaciers (unexpectedly). The study finds that ice cliffs indeed account for a disproportionate amount of mass loss in the debris-covered ablation area of Changri Nup, but that emergence velocity has been neglected in assessments of the 'debris covered glacier anomaly'.

I am impressed with the careful processing of the field and remote-sensing observations, in particular with the correction of point clouds for glacier flow and the treatment of uncertainty in general, and I find this study to be an excellent combination of high-resolution topographic datasets and robust processing to measure changes of highly dynamic features. I am particularly pleased to see attention given to emergence velocity, an aspect of glacier dynamics and mass balance that is often neglected in contemporary studies due to the recent emphasis on remote sensing observations. I have concern with the strength of the authors' refutation of the 'debris covered glacier anomaly' based on observations from a single glacier; I rather think they have highlighted the (largely unacknowledged) importance of emergence velocity, but have not demonstrated that this is the dominant or general mechanism by which debris covered glaciers thin at rates comparable to clean ice glaciers in High Mountain Asia. I suggest the authors consider some textual revision in order to better balance the focus of their discussion and conclusions with the focus of their highly sophisticated processing.

We thank the anonymous referee 1 for her/his positive appreciation of our work and we understand her/his concern about the lack of balance in the focus of our discussion and the critics of the extrapolation of our findings based on a single case. We respond on the specific points raised hereafter.

Major points:
1. The manuscript is not balanced in terms of the focus of its methods, results, and discussion. The manuscript is mostly aimed at assessing the contribution of ice cliffs to mass balance; the gold-standard methods are targeted specifically to assess this using multiple (perhaps redundant) high-resolution datasets, yet once the authors have a number for the ice cliff net ablation, the discussion is nearly all about the importance of emergence velocity. This feels like an afterthought (i.e. determination of emergence velocity itself is not given much attention in the background and methods, but this is the main topic in the discussion, whereas ice cliffs received little attention); this disparity is awkward. In particular, additional attention needs to be paid to the uncertainty in both the original emergence velocity dataset (per Vincent et al 2016) and particularly with respect to the 'updated' estimate. For example, what about uncertainties in ice thickness retrieval and differences in emergence velocity due to profile orientation? What about the uncertainty of thermal regime and its effect on column-averaged ice velocity? If emergence velocity is to be a major outcome of the manuscript, its uncertainty needs to be more carefully assessed.

We agree with the reviewer. However, we think that many recent studies, based on DEM differences often neglected the ice dynamics. This article was an opportunity to stress the potential influence of

the emergence velocity, and consequently to stress the fact that thinning rates and ablation rates are very different. We address the reviewer's comment by two developments in the text:

1- We changed the structure of the text in order to better emphasize the emergence velocity calculation. The section 3.4.2 is now titled "Ground penetrating radar" and we added more methodological development and background about the emergence velocity in a new subsection within the method section, which is now divided as: 4.1 Emergence velocity; 4.2 Ice cliff backwasting calculation; 4.3 Sources of uncertainty. We substantially enriched the 'update' estimate. We corrected the uncertainty of the GPR estimate (see below and thanks for pointing this out!) and tested different glacier thermal regime hypothesis.

Section 4.1 now reads: **"The emergence velocity refers to the upward flux of ice relative to the glacier surface in an Eulerian reference system (Cuffey and Paterson, 2010). For the case of a glacier in steady-state (i.e., no volume change at the annual scale), the emergence velocity balances exactly the net ablation for any point of the glacier ablation area (Hooke, 2005). For a glacier out of its steady state (as Changri Nup Glacier) the thinning rate observed in the ablation area is the sum of the net ablation and the emergence velocity (Hooke, 2005). On debris-covered glaciers, while the thinning rate is relatively straightforward to measure from DEM differences, for example, the ablation is highly spatially variable and difficult to measure (e.g., Vincent et al., 2016). In order to evaluate the mean net ablation of Changri Nup Glacier tongue from the thinning rate, we estimate the mean emergence velocity ($w_e$) for the period November 2015-November 2016 and for the period November 2016--November 2017 using the flux gate method of Vincent et al. (2016). As the ice flux at the glacier front is 0, the average emergence velocity downstream of a cross-section can we calculated as the ratio of the ice flux through the cross-section ($\Phi$ in m³ a⁻¹), divided by the glacier area downstream of this cross-section ($A_T$ in m²):**

$$w_e = \frac{\Phi}{A_T}$$

**This method requires an estimate of ice flux through a cross-section of the glacier, and is based here on measurements of ice depth and surface velocity along a profile upstream of the debris-covered tongue (Figs. 1 and 2). The ice flux is the product of the depth-averaged velocity ($\overline{u}$ in m a⁻¹) and the cross-sectional area. For the period November 2015-November 2016 (resp. November 2016-November 2017), the glacier slowed down compared with the 2011-2014 period and the centerline velocity was equal to 10.8 m a⁻¹ (resp. 11.1 m a⁻¹), leading to an assumed mean surface velocity along the upstream profile of 8.1 ± 0.6 m a⁻¹ (resp. 8.3 ± 0.6 m a⁻¹), as the centerline velocity is usually 70 to 80 % of the mean surface velocity along the cross-section (e.g., Azam et al., 2012; Berthier and Vincent, 2012). We used the relationship between the centerline velocity and the mean velocity, instead of an average of the velocity field along the cross section, because the image correlation was not successful on a relatively large fraction (~ 30 %) of the cross section. Converting the surface velocity into a depth-averaged velocity requires assumptions about e basal sliding and a flow law (Cuffey and Paterson, 2010). Little is known about the basal conditions of Changri Nup Glacier, but Vincent et al. (2016) assumed a cold base, and therefore no sliding. This leads to $\overline{u}$ being approximated as 80 % of the surface velocity, additionally assuming n = 3 in Glen's flow law (Cuffey and Paterson, 2010). As an end-member case, assuming that the motion is entirely by slip implies $\overline{u}$ equals to the surface velocity (Cuffey and Paterson, 2010). Consequently, we followed Vincent et al. (2016) and assumed no basal sliding, but we took the difference between the two above-mentioned cases as the uncertainty on $\overline{u}$. This leads to $\overline{u}$ = 6.5 ± 1.6 m a⁻¹ (resp. 6.6 ± 1.7 m a⁻¹) for the period November 2015-November 2016 (resp. November 2016-November 2017).**

**Assuming independence for the cross-sectional area ($\sigma_S$) and the depth-averaged velocity ($\sigma_{\overline{u}}$), the uncertainty on the ice flux ($\sigma_\Phi$) can be estimated as:**

$$\frac{\sigma_\Phi}{\Phi} = \sqrt{\frac{\sigma_{\bar{u}}{}^2}{\bar{u}} + \frac{\sigma_S{}^2}{S}}$$

Given the above mention values for the depth-averaged velocity, the cross-sectional area and the associated uncertainties, the relative uncertainty of the ice flux is ~30 %. As a result, for the period November 2015-November 2016 (resp. November 2016-November 2017), the incoming ice flux was thus 499 700 ± 150 000 m³ a⁻¹ (resp. 503 840 ± 150 000 m³ a⁻¹). The glacier tongue area was considered unchanged at 1.49 ± 0.16 km², corresponding to $w_e$ = 0.33 ± 0.11 m a⁻¹ (resp. 0.34 ± 0.11 m a⁻¹). It is notoriously difficult to delineate debris-covered glacier tongues (e.g., Frey et al., 2012). In this case, we assumed an uncertainty in the outline position of ± 20 m, leading to a relative uncertainty in the glacier area of 11 %, which is higher than the 5 % of Paul et al. (2013). In this case, the uncertainty on the glacier outline is not the main source of uncertainty in $w_e$, but for automatically delineated glacier outlines, this would be an important source of uncertainty. The updated emergence velocity is ~20 % lower than estimated for the 2011-2015 period (Vincent et al., 2016), due to both the thinning and deceleration of the glacier. As the difference in $w_e$ between November 2015-November 2016 and November 2016-November 2017 is insignificant, we consider $w_e$ to be constant and equal to $w_e$= 0.33 ± 0.11 m a⁻¹ for the rest of this study. It is noteworthy that some spatial variability is expected for $w_e$, however, we have no means to assess it."

2- We now describe in more detail the cliff evolution (number of cliffs, backwasting rates, area changes) in section 6.1 of the discussion and we shortened and substantially rewrote section 6.3. Section 6.1 is now entitled "**Cliff evolution and** comparison of two years of acquisition", and the two first paragraphs read as:

"**The total ice cliff covered area did not vary significantly from year to year, ranging from 70 ± 14 × 10³ m² in November 2017 to 72 ± 14 × 10³ m² in November 2016. The twelve individual cliffs surveyed showed large variations in area within the course of one year, with a maximum increase of 57 % for the large cliff 06 and a decrease of 34 % for cliff 03 and 09 (Table S2). The total area of these twelve cliffs increased by 8 % in one year. Interestingly, over the same period, Watson et al. (2017) observed only declining ice cliff areas on the tongue of Khumbu Glacier (~6 km away). All the large cliffs (most of them are included in the twelve cliffs surveyed with the terrestrial photogrammetry) persisted over these two years of survey, including the south or south-west facing ones (Table 1) , although south facing cliffs are known to persist less then non south facing ones (Buri and Pellicciotti, 2018). However, we observed the appearance and disappearance of small cliffs, and marginal areas became easier to classify as either ice cliff or debris-covered areas, highlighting the challenge in mapping regions covered by thin debris (e.g., Herreid and Pellicciotti, 2018).**

We calculated backwasting rates for the twelve cliffs monitored with terrestrial photogrammetry for the period November 2015--November 2016 (Table 1). The backwasting rate is sensitive to cliff area changes (because it is calculated as the rate of volume change divided by the mean 3D area) and should be interpreted with caution for cliffs that underwent large area changes (e.g., cliffs 01, 02, 03, 06, 09 and 11; Table S2). The backwasting rates ranged from 1.2 ± 0.4 to 7.5 ± 0.6 m a⁻¹. The lowest backwasting rates are observed for cliffs 11 and 12, located on the upper part of the tongue, roughly 100 m higher than the other cliffs (Fig. 1 and Table 1). The largest backwasting rates were observed for cliff 01, which expanded significantly between November 2015 and November 2016. The backwasting rates are lower than those reported by Brun et al. (2016) on Lirung Glacier (Langtang catchment) for the period May 2013-October 2014, which ranged from 6.0 to 8.4 m a⁻¹ and lower than those reported for surviving cliffs by Watson et al. (2017) on Khumbu Glacier for the period November 2015-October 2016, which ranged from 5.2 to 9.7 m a⁻¹. These**

**differences are likely due to temperature differences between sites. Indeed, the cliffs studied here are at higher elevation (5320-5470 m a.s.l.) than the two other studies (4050--4200 m a.s.l. for Lirung Glacier and 4923-4939 m a.s.l. for Khumbu Glacier).**"

2. I think some adjustment to the title and latter discussion is necessary: I do not think the authors are able to answer the title question using data from Changri Nup alone.
We modified the title of the article, which now reads "Ice cliff contribution to the tongue-wide ablation of Changri Nup Glacier, Nepal, Central Himalaya"

First, the authors provide no evidence that Changri Nup fits within the 'debris cover anomaly' framework (that Changri Nup is thinning at a comparable rate to debris-free ice at a similar elevation). This is partly due to the hypsometric differences of debris-covered and debris-free ice in the Solukhumbu region, but this is largely why the debris-cover anomaly has been determined from numerable populations of glaciers, which will exhibit a variety of hypsometric distributions.
The "debris cover anomaly" (i.e. similar thinning rates over debris-covered and debris free glaciers at similar elevations, although ablation is expected to be reduced over debris covered glaciers compared with debris-free glaciers) is to our opinion an interesting but fuzzy concept, which has been used to motivate previous studies that looked for processes responsible for enhanced ablation on debris-covered tongues. Based on the data of Brun et al. (2017), we show that the thinning rates of debris-covered areas are comparable to thinning rates of debris-free areas for glaciers in the Khumbu region (Figure R1). The thinning rate of Changri Nup Glacier agrees well with this regional pattern and therefore we do conclude that the tongue of Changri Nup Glacier is a representative "debris-anomaly" glacier.

[Figure]

*Figure R1: rate of elevation change for debris-free and debris-covered ice in the Khumbu region, based on Brun et al. (2017) data. The brown histogram represents the hypsometry of the debris-covered ice and it is stacked above the blue histogram, which represents the hypsometry of the debris-free ice. The thinning rate for the debris-covered part of Changri Nup is overlaid in grey.*

It could be possible to assess the thinning rates (and melt rates) just below the GPR transect, where debris and ice surfaces exist at the same elevations – does Changri Nup actually show evidence of comparable thinning rates for debris and ice?

We do not think that comparing melt rate just beneath the GPR section to compare clean ice and debris covered ice is possible, because the transition between clean ice and debris-covered ice is very smooth and it is hard to distinguish between the two categories. Moreover, this area is very small and it is not representative of heavily debris-covered tongues.

However, I am doubtful that this would be satisfactory, as Vincent et al (2016) has already demonstrated that melt rates at Changri Nup would be very different beneath debris and clean ice; it seems that the hypsometric parity of thinning rates for debris-covered and debris-free ice does not hold for this particular location, but for larger regions.

Put differently, there is circular logic at play – it is already known that subdebris melt rates are not equivalent to clean-ice melt rates at this location, so no amount of ice cliff melt could bring the subdebris mass balance to the same level. A way forward is to emphasize that both processes are important: neglecting emergence velocity, one does underestimate melt rates, but similarly one does if neglecting ice cliffs. However, emergence velocity has been neglected, and the Changri Nup data is the first field data to demonstrate the effects theorized by Banerjee (2018). Thus, a meaningful question is how much are the competing hypotheses responsible for boosting the thinning rate of debris-covered glaciers? I.e. how much of a boost in lowering is due to cliffs vs how much is because of emergence velocity? Or, how much 'additional' melt would be needed from cliffs to lead to thinning (or b_dot) -equivalence? Twice as much? Three times?

The point raised by the reviewer is interesting but we believe that it is not possible to address it (at least using our data), for two main reasons: first, we do not know much about the emergence (of both categories of glaciers), and consequently it is not possible to answer directly the question "how much is because of emergence velocity" raised by the reviewer. Second, the cliff melt is highly localized, whereas the emergence velocity effect is spatially distributed. Consequently, we cannot really calculate the values suggested by the reviewer, as they are expected to be very different for each glacier, because they depend on the relative areas and they depend on the ice dynamics. Instead, we calculated the area covered by cliffs which produce ablation similar to a debris-free tongue (P12-L22-24).

Can you guess how much ablation ponds are responsible for (realising that this is just part of your non-cliff net ablation, and does not affect the role of emergence velocity)?

We tried to map the area occupied by ponds, but it turns out that this task was not straightforward. Based on the UAV orthomosaic, we could not distinguish between very shallow ponds/supraglacial river systems, which have a limited contribution to ablation, and ponds that are deep enough to develop vertical mixing and therefore enhanced ablation. We mapped approximately 20 000 m² of surface covered by ponds (i.e. approximately 1.5 % of the tongue area) on the November 2017 imagery. It is noteworthy that a single pond contributed to half of this total by itself (area of 9 600 m²). This pond is located at the bottom of cliff 06 and triggered a large calving event between November 2015 and 2016. We cannot say much more about pond ablation with this dataset.

Minor points

Some nomenclature formality is needed for the cliff area terms. Variably through the manuscript there are 'planar' (cliffs are often considered inclined planes, so this is confusing), '2D' and '3D' areas of cliffs. Please clarify this early on in the manuscript, and ensure consistency.

We have removed "planar" from the manuscript and replaced it with "map view", following (Herreid and Pellicciotti, 2018). The cliff 2D area was defined as the cliff footprint (P5 L28).

P1 L20. Suggest 'have been found' in place of 'were found' for correct tense

Modified accordingly

P2 L5-11. It may be useful to use the same order for the hypotheses here as for the rest of the text, e.g. you first discuss how to test the cliff hypothesis before considering the role of emergence velocity.

Modified accordingly

P2 L6-8. This is the thesis of this paper (that emergence velocity is a major player), which it supports very well. Here, however, this is an hypothesis – that differences in emergence velocity 'can/could lead to comparable thinning rates despite differences in surface ablation. The two studies referenced are hypothetical, idealised flow-models.

We added "**could**"

P2 L10. This seems to refer to surface ablation only, yet Sakai et al 2000, Miles et al 2016, and Watson et al 2017 (ESPL) also indicate that ponds could potentially lead to significant internal ablation (which would also contribute to lowering as in Thompson et al 2016).

We separated this sentence into two parts: the first one mentions only the cliffs (we removed the reference to Miles et al 2016) and the second one mentions the supraglacial and englacial ablation due to water circulation: "**Other processes linked to supraglacial and englacial water circulation could lead to substantial ablation** (e.g., Benn et al., 2017; Miles et al., 2016; Sakai et al., 2000; Watson et al., 2018)."

P2 L12. It follows that you also need to determine the melt contribution of supraglacial ponds in order to resolve this

Modified accordingly. "**In order to partially test the first hypothesis, there is a need to calculate the total contribution of the additional melt processes to the tongue-wide surface mass balance. In this work, we focused on the ice cliff contribution, as the other processes are currently not quantifiable at the scale of a glacier tongue.**"

P2 L17. In the formulation of Equation 1, does the 'tongue' area include or exclude the ice cliff areas? That is, does p compare ice-cliff ablation to the overall surface mass balance, or to the non-cliff ablation? Is this consistent between the studies mentioned?

This point was also mentioned by other reviewers. The $p$ factor is now named $f_C$ factor to avoid a confusion with the "$p$-value" (comment from reviewer 3). The $f_C$ factor has the same definition as $p$, but we added the definition of a new factor, named $f_C^*$, which is the ratio of the cliff ablation divided by the non-cliff terrain ablation (denoted by the subscript NC):

$$f_C^* = \frac{\Delta V_C}{A_C} \frac{A_{NC}}{\Delta V_{NC}} = f_C \frac{\Delta V_T}{\Delta V_T - \Delta V_C} \frac{A_T - A_C}{A_T}$$

Based on our data, for Changri Nup Glacier, $\frac{\Delta V_T}{\Delta V_T - \Delta V_C} = \frac{1}{1 - 0.23} = 1.30$ and $\frac{A_T - A_C}{A_T} = \frac{1 - 0.07}{1} = 0.93$, consequently $f_C^* = 1.2\, f_C$.

For the consistency between the studies mentioned, we interpreted the studies as follow:
- Juen et al. 2014 : "Although the ice cliffs occupy only 1.7% of the debris covered area, the melt amount accounts for approximately 12% of the total sub-debris ablation" -> $f_C^* = \frac{12}{1.7} = 7.1$
- Reid and Brock 2014: "Analysis of the DEM indicates that ice cliffs account for at most 1.3% of the 1m pixels in the glacier's debris-covered zone, but application of a distributed model indicates that ice cliffs account for ~7.4% of total ablation." -> $f_c = \frac{7.4}{1.3} = 5.7$

- Buri et al. 2016: "Although only representing 0.09% of the glacier tongue area, the total melt at the two cliffs over the measurement period is 2313 and 8282m$^3$, 1.23% of the total melt simulated by a glacio-hydrological model for the glacier's tongue. -> $f_c = \frac{1.23}{0.09} = 13.7$

- Sakai et al 1998: From the abstract: "The ice cliff melt amount reaches 69% of the total ablation at debris covered area, although the area of ice cliffs occupies less than 2% of the debris covered area" -> $f_c = \frac{69}{2} = 35$

- Sakai et al 2000: From their Table 2: ratio of the "absorbed heat at each type of surface during the observation period (167 days)", including the "whole debris-covered zone" -> $f_c = \frac{256}{26} = 9.8$ or looking only at the debris -> $f_C^* = \frac{256}{21} = 12.2$

- Brun et al 2016: "The ice cliffs lose mass at rates six times higher than estimates of glacier-wide melt under debris, which seems to confirm that ice cliffs provide a large contribution to total glacier melt." -> $f_c = 6$

- Thompson et al 2016: "Although ice cliffs cover only ~5% of the area of the lower tongue, they account for 40% of the ablation." -> $f_c = \frac{40}{5} = 8$

As most of the studies were already within the framework of the original definition of $p/f_C$, we decided to keep the focus on this factor, instead of $f_C^*$. This example demonstrates the importance of a consistent framework for comparing these studies.

P2 L24. Please also mention the source data and method for Brun et al 2016 if you are going to for Thompson et al 2016.
Modified accordingly

P2 L28. Suggest 'positive emergence velocities will increase the : : :' as it is more concrete than 'affect'
This paragraph has been reworked. It now reads: "**Neglecting the emergence velocities (i.e. comparing thinning rates instead of ablation rates) introduces a systematic overestimation of $f_c$. This is due to the fact that cliffs ablate at higher rate than the rest of the glacier tongue: ice cliff thinning rates are thus less influenced than the thinning rates of debris-covered ice when neglecting the emergence velocity. As a consequence, the ratio of the cliff thinning rate divided by the mean tongue thinning rate will overestimate $f_c$. To correctly estimate $f_c$ and the fraction of total ice cliff net ablation, thinning rates need to be corrected with the emergence velocity.**"

P3 L5-10. It is necessary to make some mention of your emergence velocity correction in this paragraph.
Modified accordingly. "We introduce a new method based on DEM differencing, which takes into account geometric changes induced by glacier flow, **and in particular by the emergence**, and apply it to the UAV and Pléiades imagery."

P3 L28. 'GCPs' should be singular or possessive here.
Modified accordingly

P4 L15. 'equal' should be 'equivalent'
Modified accordingly

P5 L3. Incomplete sentence. 'This ensured our study/our analysis to: : :'
Modified accordingly

P6 L9. I don't believe the accuracy of this cross-sectional area. The uncertainty with respect to radar velocity in ice alone is greater than the stated value. The stated uncertainty equates to 10cm of uncertainty in ice thickness all along the cross section. Please ensure that your corrected uncertainty is propagated to your uncertainty in emergence velocity as well.

We apologize for this mistake and thank a lot the reviewer for pointing it out!

The section has been quite modified following the reviewer's major comments. The uncertainty on the GPR data is ± 15 m.

The new section 3.4.2 ("**Ground penetrating radar data**") now reads: "**A cross sectional profile of ice thickness has been measured upstream of the debris-covered tongue (Fig. 1) in October 2011, with a ground penetrating radar (GPR) working at a frequency of 4.2 MHz (Vincent et al., 2016). The original cross-sectional area was 79 300 m² in 2011 and 78 200 m² in 2015 (Vincent et al., 2016). Between November 2015-November 2016 and November 2016-November 2017, the cross sectional area decreased from $S_{2015-2016}$ = 76 900 m² to $S_{2016-2017}$ = 76 340 m² (with $S_{yr1-yr2}$ being the mean cross sectional area between the year 1 and year 2), based on the 0.86 m a$^{-1}$ thinning rate measured over the November 2015-November 2017 period along the profile. The uncertainty on the ice thickness is ±15 m (Azam et al., 2012), which leads to an uncertainty ($\sigma_S$) of ± 10 000 m², as the length of the cross-section is 670 m.**"

P6 L19. Constant and equal over the lower glacier for both periods of study, you mean. As the flux gate method can only give you a mean emergence velocity for the lower glacier, but please mention how it is expected to vary in space, and how this might affect your results for ice cliffs and for the whole glacier.

We added: "**It is noteworthy that $w_e$ is likely to be spatially variable, however, we have no means to assess its spatial variability.**"

P9 L2. Your kernel sizes are with units of pixels, correct?

It is in pixel, this is added in the text.

P11 L10. Can you calculate or estimate the 3D area of these cliffs in order to calculate a mean backwasting rate for comparison to other studies? As the rate of elevation change over a cliff-affected area is heavily influenced by, e.g. their height and slope, the backwasting rate is perhaps easier to compare between studies (or indeed between years, as your 2016-2017 data is quite different).

We added a supplementary table (Table S2), which shows the cliff 3D area in 2015 and 2016 and we calculated the backwasting distance in Table 1. We calculated the backwasting as individual cliff volume loss from terrestrial photogrammetry (i.e., only for the period Nov. 2015 – Nov. 2016), divided by the mean 3D area. The backwasting rate is compared with other studies in section 6.1.

P11 L18-19. For p, it makes sense to me that the comparison would be cliff area tonon-cliff area, rather than cliff area to the whole area. Please check what prior studies have used for this calculation.

For a comparison with previous studies, see our response above. We added the results for $f_C^*$ as well.

P11 L25. Why the much higher melt rates in 2016-2017?

The difference in mass balance between 2015-16 and 2016-17 is also observable in the glacier-wide mass balance of the near-by debris-free West Changri Nup Glacier (-0.76 and -2.56 m w.e. yr$^{-1}$, for 2015-16 and 2016-17, respectively) (P. Wagnon, unpublished data). The exact reasons explaining such large differences need to be analyzed but are not related to air temperature almost similar between both years (-3.6°C measured at the AWS at 5360 m a.s.l. on West Changri Nup, in both

years, from 1 November to 31 October). The mean summer temperatures (1 April – 30 September) are also very similar (0.3°C for 2016 versus 0.1°C for 2017). The difference might come from other meteorological variables, but this has not been analyzed in details yet, and it is not the scope of this present paper.

P12 L8. 'Mean tongue' is not a sensible term. Consider 'relative to the whole tongue'
Modified accordingly

P12 L10-18. Neglecting the emergence velocity, what portion of the glacier's total ablation would be accounted for by ice cliff melt? Perhaps it would likewise be useful to compare the area-averaged losses due to ice cliffs and emergence velocity – are they of comparable magnitude?
We added this calculation and calculated the $f_C^*$ ratios when neglecting the emergence, the section now reads as:
"In this case, the factor $f_c$ would be 4.5 ± 0.6 (and $f_C^*$ would be 5.4 ± 0.7), which is 50 % higher than the actual value. The cliffs would be found to contribute to ~34 \% of the tongue ablation. For the period November 2016--November 2017, the factor $f_c$ would be 3.6 ± 0.6 (and $f_C^*$ would be 4.3 ± 0.7), which is 20 % higher than the actual value. The cliffs would be found to contribute to ~29 % of the tongue ablation. This might partially explain why previous studies found significantly higher values of $f_c$, and stresses the need to estimate and take into account the ice flow emergence, even for almost stagnant glacier tongues like Changri Nup Glacier (see Discussion below)."

P12 L19. Consider 'the' debris-cover anomaly
Modified accordingly

P12 L22. This emphasizes the problem with your p calculation – it is not comparing ice cliff to debris, but ice cliff to drbis-and-cliff mixtures. Your values of p will increase with this correction. I.e. total melt due to cliffs was 440000m3 for 2015-2016, and they covered an area of 113000m2. Total melt for the whole glacier was 1,918,000m3 over an area of 1.49 km2. Thus the non-cliff melt was 1478000m3 over an area of 1.377km2. And thus p is 3.6 (20% higher). Can you also calculate what p would be neglecting your emergence velocity estimation (for comparison to the studies mentioned?
As mentioned earlier in this response: $f_C^* = 1.2\ f_c$ for this year on Changri Nup (in agreement with the reviewer's calculation!). We added the influence of neglecting the emergence velocity on $f_C^*$ as well.

P12 L29. This is a very good point, but highlights a key difficulty for the paper. The authors have not demonstrated that the 'debris-cover anomaly' is applicable to Changri Nup at all! That is to say – the authors have not demonstrated that Changri Nup's debris-covered area is indeed thinning at a rate comparable to clean ice glaciers at the same elevation (the point of the debris-cover anomaly). Vincent et al 2016 has already demonstrated that the surface mass balance of Changri Nup is lower than it would be if debris were not present. Here you demonstrate that ice cliffs cannot bring the debris area's mass balance to the same level, but does Changri Nup even fit the debris-cover anomaly in the first place? This is not so problematic for your analyses and paper, but for the generalisation of your results to other areas (P13 L1-2 especially)
We both agree and disagree with the reviewer. Figure R1 shows that Changri Nup Glacier fits within a regional pattern of "debris cover anomaly". Moreover, we think that our calculation related to the cliff area equivalent ablation is true independently of the debris-cover anomaly, as it is based only on field measured ablation rates (for the debris-free surface) and the ablation rates measured in this study. Consequently, we decided to keep the lines 20-26 of page 12 unchanged. However, we understand the reviewer's concern about the generalization to the debris-cover anomaly, which implies additional assumptions, such as the reduced emergence velocity for debris-covered tongues. That's why we substantially modified the rest of this section.

P13 L4. I think this section needs to be tidied up with respect to nomenclature, in particular replace 'tongue' with 'ablation area'.
We prefer to keep the word tongue, because the glacier tongue is not the same as the ablation area.

P13 L8. This hypothetical analysis is very worthwhile, but as stated in the text, 'has already been shown by Banerjee (2018)'. Please properly reference that study early in this section (you can state that you provide the first field evidence supporting this hypothesis) and reduce this text accordingly. I recommend that you expand the discussion of the responsibility of reduced emergence velocity vs enhanced ablation (how important are cliffs and ponds for mass balance, then?) or consider more fully how the mass balance and emergence velocity (thus thinning rates) of both systems will continue to evolve. Is the apparent parity of thinning rates a temporary feature in this evolution, or should we expect this to perpetuate?
This section has been substantially modified in the revised version of the article. It is challenging to discuss the enhanced ablation of ponds, because we know very little about them on Changri Nup Glacier. We tried to map them, but with limited success because we can't distinguish between the large ponds, that are deep enough to produce enough ablation and the shallow ponds, which play a much more minor role (see our response above). While we appreciate the suggestion to orient the discussion towards the future evolution of these processes, we definitely think that we do not have enough elements to discuss this. The revised section reads:

[revised manuscript text omitted]

P13 L22-23. The manuscript has demonstrated that emergence velocities (and the difference between emergence velocity for clean-ice and debris-covered areas) are a key part of the debris-cover anomaly, but the manuscript has certainly not demonstrated that this is always (or even generally!) the reason for the thinning rate parity. Consequently I respectfully but strongly think that your statement should be modified, e.g. 'In conclusion, we have demonstrated that emergence velocity differences are as important as ice cliffs and supraglacial ponds in the calculation of melt rates for debris covered glaciers, and that the 'debris cover anomaly' is in part due to the confusion of thinning rates and net ablation.'
We modified this sentence. See our modified version just above.

P13 L24. This section is very out of place with regards to the underlying theme of the manuscript, especially as your discussion up until now focuses on cliffs not being important. I suggest as a segue to emphasize that melt rates are substantially higher than without ice cliffs, and that the primary analysis of the study is thus of benefit for modelling studies (otherwise why automatically delineate cliffs at all?).

We removed this section

P14 L6. Please include a mention of where Brun et al (2016) falls in this spectrum.
Modified accordingly

P14 L7-10. This is an important consideration that should be expanded upon. Your analysis including flow correction is without a doubt more sophisticated and 'correct' than prior efforts, but it extremely limited in its transferability because of the field data requirements. While emergence velocity is clearly an important and neglected aspect of studies addressing debris-covered glacier mass balance, it is extremely difficult to assess (and thus also the reliance on overall thinning rates rather than mass balance). It is not enough to say 'more data would be helpful' when you advocate abandonment of an entire train of thought; rather, I think it is important to acknowledge why such data do not already exist (why debris thickness has prevented widespread ice thickness measurement through debris), and to address alternative methods of assessing emergence velocity (e.g. networks of ablation stakes combined with dGPS).

The new paragraph reads: "A significant obstacle to applying our method to other glaciers is the need to estimate the emergence velocity, which requires an accurate determination of the ice fluxes entering the glacier tongues. **The measurement of ice thickness with GPR systems is already challenging for debris-free glaciers, as it requires to drag emitter, receiver and antennas along transects of the glacier surface. It is even more challenging for debris-covered glaciers, as the hummocky surface prevent the operators from dragging a sledge.** More field campaigns dedicated to ice thickness and velocity measurements (Nuimura et al., 2011, 2017) or the development of airborne ice thickness retrievals through debris are recommended. **The precise retrieval of emergence velocity pattern using a network of ablation stakes combined with DGPS is a promising alternative, in particular if combined with detailed ice flow modeling (e.g., Gilbert et al., 2016).**"

P14 L24. There is no discussion of this point, but I think it would be useful to expand upon (briefly). What do we do with your results? How does this affect models of debris covered glacier mass balance and/or dynamics?

The revised sentence reads: "**The main limitation of our study is its short spatial and temporal extent. It would be very worthwhile to obtain longer-term and multiple sites quantification of the relative ice-cliff contribution to net ablation. Then a compilation of these data would allow to develop empirical relationships for cliff enhanced ablation, which could** be included into debris-covered glacier mass balance models."

Figure 6. These are normalised change in the volume change (rather than cliff volume), correct?

Modified accordingly

Figure 10. I like the simplicity of Figure 10, but it is deceptive in its simplicity (the scales are of course arbitrary). It would be worthwhile to emphasize that this is one hypothetical transient state (another would be to double b_dot for debris free glaciers, the end-member with no increase in w_e in either case). It also would be worthwhile to highlight here the fraction of b_dot due to ice cliffs (the focus of the study), and to emphasize that w_e is the least measured aspect of the chart.
Modified accordingly

Anderson, L. S. and Anderson, R. S.: Modeling debris-covered glaciers: response to steady debris deposition, The Cryosphere, 10(3), 1105–1124, doi:10.5194/tc-10-1105-2016, 2016.

[revised manuscript text omitted]

---

## Author Comment (AC2) · 4 Jul 2018

**Response to anonymous referee 2**

We thank referee 2 for the detailed review. In this response, the reviewer's comments are in black standard font. Our response is in standard blue font and the modifications to the manuscript are in blue bold font.

This study evaluated ablation at ice cliff of Changri, debris-covered glacier tongue using UAV-image and dem (and also Preades). They consider the emergence velocity and also evaluated several kinds of errors carefully. They concluded that recent elevation changes at tongue of Changri Glacier is mainly due to lower emergence velocities, not ablation at ice cliffs. In particular, Figure 4 and 5 are very impressive for me, because it's ideal data to analyze ablation process of debris-covered glaciers (Off course we have to consider distribution of emergence velocity). I think this result can be analyzed for other target. I'm looking forward to read other papers. I have some comments as follows. I hope my comments will help to improve your manuscript.

We thank the anonymous referee 2 for her/his positive appreciation of our work.

<Specific comments> Page2 L21 '35 (Sakai et al., 1998)' » Please refer Table 2 in Sakai et al.(2000) p = 256/26=9.8. The value 35 calculated from Sakai et al.(1998) is inaccurate value.
Modified accordingly

Page 2 L26 'but it has typically been neglected in the calculation of p.' » Which previous study neglected emergence velocity? Please address the references.
As this applies to all the studies listed in the beginning of the section, we added "**, for all the above-mentioned studies.**" at this sentence.

Page 2 L26 '5.7–6.4 _ 3.9 m a-1' » 6.4 _ 3.9 m a-1 is the value of mass balance in Nuimura et al.(2011)
Thanks for pointing out this mistake, it has been corrected and the correct values are now reported.

Page 2 L28-31 'Emergence velocities will affect the thinning rates of debris-covered ice and ice cliffs equally. But since the cliffs ablate at higher rate, their thinning rate is relatively less influenced than the thinning rate of debris-covered ice. As a consequence, the ratio of the cliff thinning rate divided by the mean tongue thinning rate will overestimate p.' » Those explanation is a little bit ambiguous expression. Please write more clearly.
We modified this paragraph, which now reads as: "**Neglecting the emergence velocities (i.e. comparing thinning rates instead of ablation rates) introduces a systematic overestimation of $f_c$. This is due to the fact that cliffs ablate at higher rate than the rest of the glacier tongue: ice cliff thinning rates are thus less influenced than the thinning rates of debris-covered ice when neglecting the emergence velocity. As a consequence, the ratio of the cliff thinning rate divided by the mean tongue thinning rate will overestimate $f_c$. To correctly estimate $f_c$ and the fraction of total ice cliff net ablation, thinning rates need to be corrected with the emergence velocity.**"

Page 3 '2 Study area' »There are basic information of study area in Vincent et al.(2016). But, I recommend that ELA around the Changri glacier and altitudes (Max, min) information are necessary, here.
Modified accordingly

Page 6 L6 I cannot find out the location of cross section in Fig. 1 or 2.
We added them on the figures.

Page 7 section 4.2 and 4.3 Ice cliff is unstable. Sometimes they disappear or newly emerge in one melting season. Are there any ice cliffs diminished or emerged? And you have neglected those ice loss in this study?

The cliff outlines were updated for each year. Globally, we observed little change in the total area covered by ice cliffs (69 876, 71 826 and 69 357 ± 14 000 m² for Nov. 2015, 2016 and 2017, respectively). However, the reviewer is right and there were some substantial changes observed for individual cliffs. This is now discussed in details in section 6.1 ("**Cliff evolution and** comparison of two years of acquisition") and we added a table in the supplement (Table S2) showing the evolution of individual cliffs:

"**6.1- Cliff evolution and** comparison of two years of acquisition

**The total area occupied by ice did not vary significantly from year to year, ranging from 70 ± 14 × 10³ m² in November 2017 to 72 ± 14 × 10³ m² in November 2016. The twelve individual cliffs surveyed showed large variations in area within the course of one year, with a maximum increase of 57 % for the large cliff 06 and a decrease of 34 % for cliff 03 and 09 (Table S2). The total area of these twelve cliffs increased by 8 % in one year. Interestingly, over the same period, Watson et al. (2017) observed only declining ice cliff area on the tongue of Khumbu Glacier (~6 km away). All the large cliffs (most of them are included in the twelve cliffs surveyed with the terrestrial photogrammetry) persisted over these two years of survey, including the south or south-west facing ones (Table 1) , although south facing cliffs are known to persist less then non south facing ones (e.g., Buri and Pellicciotti, 2018). However, we observed the appearance and disappearance of small cliffs, and marginal areas became easier to classify as either ice cliff or debris-covered areas, highlighting the challenge in mapping regions covered by thin debris (e.g., Herreid and Pellicciotti, 2018).**

**We calculated backwasting rates for the twelve cliffs monitored with terrestrial photogrammetry for the period November 2015-November 2016 (Table 1). The backwasting rate is sensitive to cliff area changes (because it is calculated as the rate of volume change divided by the mean 3D area) and should be interpreted with caution for cliffs that underwent large area changes (e.g., cliffs 01, 02, 03, 06, 09 and 11; Table S2). The backwasting rates ranged from 1.2 ± 0.4 to 7.5 ± 0.6 m a⁻¹ , reflecting the variability in terms of ablation rates among the terrain classified as cliff (Fig. 9). The lowest backwasting rates are observed for cliffs 11 and 12, located on the upper part of the tongue, roughly 100 m higher than the other cliffs (Fig. 1 and Table 1). The largest backwasting rates were observed for cliff 01, which expanded significantly between November 2015 and November 2016. The backwasting rates are lower than those reported by Brun et al. (2016) on Lirung Glacier (Langtang catchment) for the period May 2013-October 2014, which ranged from 6.0 to 8.4 m a⁻¹ and lower than those reported for surviving cliffs by Watson et al. (2017) on Khumbu Glacier for the period November 2015-October 2016, which ranged from 5.2 to 9.7 m a⁻¹. These differences are likely due to temperature differences between sites. Indeed, the cliffs studied here are at higher elevation (5320-5470 m a.s.l.) than the two other studies (4050-4200 m a.s.l. for Lirung Glacier and 4923-4939 m a.s.l. for Khumbu Glacier).**"

Page 8 '4.4.1. Emergence velocity' 'The debris-covered part of the tongue has an area of 1.49 _ 0.16 km2'(Page3 line 16) The uncertainty is induced assuming that there are 20 m uncertainty in the glacier outline Vincent et al.(2016). I think estimation of the tongue area is difficult. I have never been to the Changri Glacier, therefore, I'm not sure the confidence of glacier outline at the terminus. But, in general, it is difficult to estimate outline of glacier terminus at debris-covered glacier. Then, we have to measure ice depths at two cross sections, and calculate emergence velocity between the two cross sections to avoid large error due to glacier area estimation. You can discuss.

This issue was extensively discussed in Vincent et al. (2016), who produced the glacier outline. We agree with the reviewer that debris-covered glacier outline mapping is a timely issue and consequently, we added: "**It is notoriously difficult to delineate debris-covered glacier tongues (e.g., Frey et al., 2012). In this case, we assumed an uncertainty in the outline position of ± 20 m, leading to a relative uncertainty in the glacier area of 11 %, which is higher than the 5 % of Paul et**

**al. (2013). In this case, the uncertainty on the glacier outline is not the main source of uncertainty in $w_e$, but for automatically delineated glacier outlines, this would be an important source of uncertainty.**"

Page8 L13-14 ' The maximum net ablation measured with stakes within the period 2014–2016 on the tongue of Changri Nup was chosen as an upper limit equal to 2.22 m a-1' » Please explain why you can choose the maximum net ablation measured with stakes can be assumed to be the maximum emergence velocity.

For a glacier in imbalance, the ablation is higher than the emergence velocity and the glacier surface thins. Consequently, the ablation can be seen as an upper bound for the emergence velocity. In this case, we took a rather extreme value for the uncertainty on the emergence. We added:
"**For a thinning glacier, the net ablation is higher than the emergence velocity (Hooke, 2005), consequently, the net ablation can be used as a proxy for the upper bound for the emergence velocity.**"

P13 L3-23 and Fig. 10 ãˇAAˇ In this discussion, you have compared debris-covered and debris-free glaciers in equilibrium and transient (shrinking) regime. But, your target is ice loss at ice cliff. Almost assumptions are based on part of other studies. Further, I cannot accept some assumptions, Ex. ice flux is same at both debris-covered and debris-free glaciers. Usually, debris-covered glaciers are large, and debris-free glaciers is small. Further, each altitude are different. Then, I think we cannot discuss without the observation of debris-free glaciers.

The paragraph referred to by the reviewer has substantially been modified. We added the following sentence about this specific comment: "**The ice flux at the ELA is expected to be driven by accumulation processes, and consequently it is reasonable to assume similarity for both debris-covered and debris-free glaciers.**"

Reference Sakai A, Takeuchi N, Fujita K, Nakawo M (2000) Role of supraglacial ponds in the ablation process of a debris-covered glacier in the Nepal Himalayas. International Association of Hydrological Sciences, Publication No. 264 (Symposium at Seattle 2000 - Debris-covered glacier), 119-130.

Brun, F., Buri, P., Miles, E. S., Wagnon, P., Steiner, J. F., Berthier, E., Ragettli, S., Kraaijenbrink, P., Immerzeel, W. W. and Pellicciotti, F.: Quantifying volume loss from ice cliffs on debris-covered glaciers using high-resolution terrestrial and aerial photogrammetry, J. Glaciol., 62(234), 684–695, doi:10.1017/jog.2016.54, 2016.

Frey, H., Paul, F. and Strozzi, T.: Compilation of a glacier inventory for the western Himalayas from satellite data: methods, challenges, and results, Remote Sens. Environ., 124, 832–843, doi:http://dx.doi.org/10.1016/j.rse.2012.06.020, 2012.

Herreid, S. and Pellicciotti, F.: Automated detection of ice cliffs within supraglacial debris cover, The Cryosphere, 12(5), 1811–1829, doi:10.5194/tc-12-1811-2018, 2018.

Hooke, R. L.: Principles of glacier mechanics, Cambridge university press., 2005.

Paul, F., Barrand, N. E., Baumann, S., Berthier, E., Bolch, T., Casey, K., Frey, H., Joshi, S. P., Konovalov, V., Le Bris, R., Mölg, N., Nosenko, G., Nuth, C., Pope, A., Racoviteanu, A., Rastner, P., Raup, B., Scharrer, K., Steffen, S. and Winsvold, S.: On the accuracy of glacier outlines derived from remote-sensing data, Ann. Glaciol., 54, 171–182, doi:10.3189/2013AoG63A296, 2013.

Watson, C. S., Quincey, D. J., Smith, M. W., Carrick, J. L., Rowan, A. V. and James, M. R.: Quantifying ice cliff evolution with multi-temporal point clouds on the debris-covered Khumbu Glacier, Nepal, J. Glaciol., 63(241), 823–837, 2017.

---

## Author Comment (AC3) · 4 Jul 2018

**Response to anonymous referee 3**

We thank anonymous referee 3 for the detailed review. In this response, the reviewer's comments are in black standard font. Our response is in standard blue font and the modifications to the manuscript are in blue bold font.

— SUMMARY ———-

Brun et al. present an analysis aiming at determining the actual role of ice cliffs in the so-called "debris-cover anomaly". The analysis in performed on Changri Nup Glacier, Everest region, and combines both in-situ and remote-sensing data for a two-year period. The results are bold: The debris-cover anomaly - the authors say - does not exist, but is the result of confusion around the concepts of "thinning rates" and "net ablation". The "anomaly" – the authors explain – only comes to happen because past studies failed to account for emergence velocities. In reality, the similar thinning rates observed for debris-covered and debris-free glaciers are a result of the difference between net ablation and ice flow emergence being coincidentally similar for the two types of glaciers. As much as I agree with the first part of the interpretation, I do believe that the second part is too weakly backed-up: The authors try to generalize their field-based results in the discussion section, but the result is not convincing. Some general statements (e.g. debris covered glacier have smaller accumulation-area-ratio and are generally smaller) should be better corroborated (inventories for doing that exist by now).

A part from the above, the manuscript has a very high standard: The topic is of high relevance and actuality, the introduction is well written, the text is easy to follow, the technical analysis is clearly done by experts, the relevant literature is cited in an exemplarily manner, and the figures are illustrative. I am fully convinced that the manuscript will be an important addition to the glaciological literature once it is revised.

We thank the anonymous referee 3 for her/his positive appreciation of our work. We understand his/her concern about the disproportion between the strength of our conclusion and the weakness of the theoretical arguments we advanced. We addressed this concern by revising a large part of the discussion (see details in our response to the general comments below)

— GENERAL COMMENTS ———-

1) Sample size

The fact that all claims are built on one glacier (sample size = 1) is a clear handicap for the general conclusions the authors are aiming at. For making the point that much of the debris-cover anomaly is due to a confusion of concepts, the sample size is not an issue. This can be shown with one single example, and this is really the paper's merit. Where it becomes more difficult is when the authors stat arguing that "[for debris-covered glaciers,] the combination of reduced emergence velocities and lower ablation coincidently sum up to similar thinning rates as [for] debris-free glaciers" (p. 13, L. 19-20). Arguing for a regional-scale "coincidence" seems at least adventurous with the single data-point at hand.

I see two ways of solving this: Either (1) the authors try to get hold of published data that exist for other glaciers, or (2) they refine their theoretical argumentation (Sec. 6.3) and back up some of the as-yet little-supported claims (see next comment) with remote-sensing and inventory data.

We thank the reviewer for the two suggestions and respond to his/her comment together with the next comment.

2) "Theoretical considerations"

The "theoretical considerations" presented in Sec. 6.3 is the part of the manuscript that I found the least convincing. Unfortunately, it is the crucial one.

The problem is that the arguments seem to be much based on qualitative considerations, whilst the author's point is much focused on quantitative statements. Two examples: (1) "debris-covered glaciers have lower accumulation-area ratios than debris-free glaciers" (p. 13 L. 9). The claim is

backed up with a reference (Scherler et al., 2011) but it would be ways more convincing to have some actual numbers corroborating this. These can either be re-presented from the original publication or re-compiled from inventory data and remote sensing products (I'm fully aware that the second option would be much more work-intensive). Ideally, a distribution of AARs would be shown for both debris-free and debris-covered glaciers, and the difference quantified. (2) "the glacier response time of a debris covered glacier is longer compared with a debris-free glacier (Rowan et al., 2015), therefore the clean tongue will shrink faster than the debris-covered tongue, further enhancing the difference between ADC and ADF." (p. 13 L. 15-16). Well, again, although a reference is given, it would be so much more convincing having two distributions shown, and a difference quantified. In this case, however, I'm not even sure whether this is necessary since what actually would matter are the present, actual sizes of the debris-free and debris-covered tongues – and not their response times. I think that fixing this section is the only major work that is in front of the authors.

We agree that some claims in this section 6.3 were supported only with qualitative considerations, preventing our conclusions from being convincing. We re-wrote a large part of this section 6.3 (see below) being more cautious and following the reviewer's suggestions, we also backed up our arguments with quantitative statements.
A distribution of AARs for debris-covered and debris-free glaciers could not be shown because ELAs are not available everywhere with enough accuracy. However, to quantify the fact that debris-covered tongues are most of the time larger than debris-free ones, we plotted the minimum elevation as a function of debris cover percentage for all glaciers in HMA larger than 2 km$^2$ (approx. 6500 glaciers) (new fig. 10, added in the revised manuscript). We can see that the larger the percentage of coverage by debris, the lower the glaciers flow, which is an indication that debris-covered glaciers have on average a larger ablation area than the debris-free glaciers. Concerning the response time, we agree that this was not backed up, and not really necessary in our analysis, and in turn it has been removed.

[revised manuscript text omitted]

3) Introduction
A rather minor issue: The concept of "emergence velocity" is, obviously, of central importance to the paper. Since one of the main conclusions is that there is confusion around the term, I think it would make much sense to provide a clear definition in the introduction. Some indication on how the quantity is typically calculated from field data (or other types of data) may also be helpful for the one or the other reader.
We modified the method section to include a more detailed description of the emergence velocity in section 4.1. The alternative methods to measure the emergence velocity are mentioned in the discussion and in the conclusion.

— LINE-BY-LINE COMMENTS ————-
What follows is a series of line-by-line comments of various nature, ranging from comprehension questions to stylistic corrections and including some specific suggestions for issues the authors may want to think about or change.
P.2 L.14: Maybe a detail but I see a danger of the "p" being referred to as "p-value" at some stage. This would obviously be extremely misleading, since the term is reserved for something very specific in statistics.
We now name this quantity $f_C$ and defined $f_C^*$ which is the cliff ablation enhancement factor compared to non-cliff area (instead of the average glacier tongue).

P.2 L.15: I was confused by the mixture for plurals ("cliffs") an singulars ("cliff"). At that stage, I even briefly asked myself if "p" was something defined at the cliff-scale (i.e. one "p" for every cliff). Please avoid the confusion by using consistent wording.
Cliffs is plural, and cliff is singular. In some cases however it is necessary to use the singular to denote the singular sum of the plurals: e.g. *net ice cliff ablation* refers to ablation from all ice cliffs. We have attempted to clarify this and be consistent throughout the revised manuscript.

P.3, L15: Remove "in the same outline" (there is no danger of misunderstanding that)
Modified accordingly

P.4, L.7: "using a [not "the"] Structure from Motion algorithm"

Modified accordingly

P.5, L.13-14: I was wondering whether the relatively large offset determined for stable terrain (-7 or so meters) requires a short comment/explanation?
This offset is usual, and due to the fact that Pléiades DEMs are derived from orbital parameters only (i.e., without GCPs). Consequently, while the geometry of the DEM is robust it is somehow "floating" in the 3D space, with offsets on each components that can be up to ~10 m. We added: "**This vertical offset is expected, as the DEMs are derived from the orbital parameters only (Berthier et al., 2014)**"

P.5, L.30: "The velocities measured with Pléiades match well with the field data" –> I may have missed it, but I don't think any in-situ velocity measurements were described so far? (The only reference to such measurements seems to be at P.3 L.14, but I understood that info only to be a side note on how the glacier outline was derived in another publication?)
We clarified this point: "The velocities measured with Pléiades match well with the field data **(ablation stake displacements measured with a DGPS between November 2015 and November 2016)**, with the…"

P.6, L15-16: Can a word be spent in discussing the implication of assuming a homogeneous w_e? That quantity is a distributed field, and I have the impression that assuming a similar w_e for all ice-cliffs that are considered is an important assumption? Some discussion is found later, but here is where the question arises
We added: "**It is noteworthy that some spatial variability is expected for $w_e$, however, we have no means to assess it.**"

P.6, L.26: I'm not sure to understand what "deformed" means in this case. "Deformed" how?
"deformed" meant non homogeneous of the individual points of the PCs: "we deformed the PCs, **by displacing its individual points,** for…"

P.7, L.19-20. I found the concept of analogous points somewhat abstract. Would it make sense to provide a figure with a visual example?
We added a supplementary figure (fig. S5) showing this:

[Figure]

Fig. S5 - Examples of the methodological processing for cliff 05, located on a slow flowing area (left panels) and cliff 11, located in a fast flowing area (right panels). For all the panels the cliff outlines are represented in UTM45/WGS84. a- influence of the glacier flow correction, and comparison with a uniform translation. B- example of analogous points needed for the triangulation regularization. c- difference between the individual cliff outlines and the cliff footprint needed to calculate the cliff contribution for gridded data (DEMs).

P.8, L.2-3: Also in this case, a visualization would probably make it easier to understand what is meant exactly.
See the figure above

P.9, L.26-27: "We experimentally determined L = 150 m for the UAV and L = 150 m for the Pléiades data" –> "We experimentally determined L = 150 m for both the UAV and Pleiades data" (or should one of the two "150" read something else?)
Modified accordingly

P.11, L.9: The unit of -3.88+/-0.27 is missing
Modified accordingly

P.11, L.10: Here and elsewhere: In light of the estimated uncertainty, it would make sense to state 440+/-54 x 10^3 m3/a (instead of 439 689 +/- 54 000 m3/a).
Modified accordingly

P.11, L.26: I'm not following: Is the stated value (1.51+/-0.21 m/a) already corrected for emergence?
Yes, we added: "**after correction for the emergence**"

P.12, L.7-8: The last part of the sentence is rather involved. Can't you simply say that the cliffs seem to contribute a constant share to the total ablation?
Modified accordingly

P.12, L.14: Here and below: Consider replacing "original" with "actual".
Modified accordingly

P.12, L.15: Remove "Doing the same".
Modified accordingly

P.12, L.17: Replace "emergence velocity" with "ice flow emergence" (saying "the influence of velocity" sounds somewhat odd).
Modified accordingly

P.12, L.24-25 An alternative (simpler?) wording would be "Since ice cliffs typically cover a very limited area, thus, it is unlikely that they can explain the debris-cover anomaly."
The new sentence reads: "Since ice cliffs typically cover a very limited area (Herreid and Pelliciotti, 2018), it is unlikely that they can enhance the ablation of debris-covered tongues enough to reach the ablation of ice-free tongues."

P.12, L.27: Check the wording: "englacial hydrology" is not an "ablation-related processes" (it's rather a "discipline", as e.g. glaciology)
Modified accordingly

P.12, L.28, sentence starting with ".Yet this does not: : :" –> Split the sentence somewhere; it is very long.
Modified accordingly

P.13, L.1-23: This is the part that really needs revision.

P.13, L.7: The unit of "density" should be kg/m3 (not m2)
Modified accordingly

P.13,L.10: Well, the comparison is somewhat "cheated", as it should certainly include areas of the same size.
This section was largely modified, please refer to the new version at the beginning of this response.

P.13, L.12: Not entirely sure what "both variables" is referring to. To mass balance and emergence velocity?
This section was largely modified, please refer to the new version in the beginning of this response.

P.13, L.20: If I read this correctly, you imply that the similar thinning rates for debris-free and debris-covered glaciers are only observed now, and that this was different in the past and will be different in the future. Is this correct? If so, state that explicitly.
We do not claim this. We just try to reason within a transient framework that is somehow close to the recent situation (i.e., corresponding to the geodetic observations of the last decades).

P.13, L.30-31: I don't understand the sentence. Especially the two "in" within the parenthesis create confusion.
This section was removed according to reviewer's 1 comment.

P.13, L.32: Why "nevertheless"? What's the logical link to the previous sentence?
This sentence has been moved to section 6.1 and "nevertheless" has been removed.

P.14, L.2: "i.e. the p factor defined in this study" should live in a parenthesis (the sentence is difficult to understand at the moment). Also try to split the sentence as it is very long.
Modified accordingly

P.14, L.5: "models [: : :] are not directly comparable with the observations" –> Explain (or at least give a hint) why not.
"For instance, the $f_C$ values from models (Buri et al., 2016; Juen et al., 2014; Reid and Brock, 2014; Sakai et al., 1998) are not directly comparable with the observations (Brun et al., 2016; Thompson et al., 2016), **because they usually require additional assumptions about e.g., the sub-debris ablation or emergence velocity**."

P.14, L.5: Not sure to understand what you mean with "flow components". Please clarify. In light of the claims provided above, moreover, I'm not sure to understand the latter part of the sentence (the one that "advocates for a more consistent framework"). This may be clearer after revision.
We removed this sentence.

P.14, L.10: As much as I agree with the statement, I don't think it is appropriate making it "yours": Geophysicists are working on that since years, after all.
"More field campaigns dedicated to ice thickness and velocity measurements (Nuimura et al., 2011, 2017) or the development of airborne ice thickness retrievals through debris **are needed, as stressed by the outcome of the Ice Thickness Models Intercomparison eXperiment (Farinotti et al., 2017).**"

P.14, L.16: "or englacial conduits" –> That's a very speculative claim, isn't it? I would suggest to flag it as such.
"Other contributions, such as ablation from supra-glacial lakes**, or even from englacial conduits**, are potentially…"

P.14, L.18: "we hypothesize that" –> In the last section (Sec. 6) the claims were stated in a much more decided way. Why this caution here? The text should be coherent in what the level of trust in the results is concerned.
We adjusted the rest of the manuscript on the level of confidence of the conclusion.

P.14, L.18-19: "the debris-cover anomaly could be a result of lower emergence velocities and reduced ablation" –> This is basically the main claim of the paper. Whether it is suitable of having it in the conclusion section or not very much depends on how convincing Sec. 6 will be after revision.

Ok

P.14, L.22: ": : :our suggested framework would inform estimates of ice cliff ablation: : :" –>not sure to understand how "inform" is used here. Can you reword?
"A comparison of $f_C$ or $f_C^*$ values calculated for other debris-covered glaciers under our suggested framework would be informative, in order to compare estimates of ice cliff ablation for other and potentially much larger debris-covered tongues."

P.14, L.24: "[: : : it] is required to include these results into debris-covered glacier mass balance models" –> I'm not entirely sure but it looks like you advocate for mass balance models to include a "p"-factor? If this is actually your message, please be more explicit in saying that.
This sentence was not really clear in the original manuscript. It is revised as: "**It would be very worthwhile to obtain longer-term and multiple sites quantification of the relative ice-cliff contribution to net ablation. Then a compilation of these data would allow developing empirical relationships for cliff enhanced ablation, which could be included into debris-covered glacier mass balance models.**"

P.14, L.28-29: If what I understood the message correctly, the sentence could be adjusted to "Two research directions could be (a) to extensively measure ice thicknesses and (b) to install networks of stake measurements to assess the spatial variability of ice flow emergence."
Modified accordingly

— COMMENTS TO FIGURES ————-
Fig. 1: (a) The red box in the upper-right inset is misleading, since it is not the part enlarged in the main figure. Consider replacing it with a red dot or similar. (b) Last parenthesis of the caption: Why "measured"? (The word can simply be removed.)
Modified accordingly

Fig. 2: Please tell what coordinates are used. The black line is the same as the red-dashed one in Fig. 1, I guess? (Readers shouldn't be guessing ;-) )
Modified accordingly

Fig. 3: (a) In the caption, spell out what U_s, w_s, etc are. (b) "Local slope" is misleading; it looks more like the local tangent of the surface (and my guess is that "nalpha =local slope").
Modified accordingly

Fig. 4+5: (a) Can the colour-bar be stretched and more values be added? At the moment, it is difficult to tell what colour corresponds to, e.g. -2.5 m.  (b) What is the meaning of "raw" in "raw elevation change"? (c) Please indicate the time span between the UAV surveys (or the dates of the surveys as such). (d) "from flow for" –> Do you mean "for iceflow from"? (e) For consistency, the last sentence should read "Zoom in the dashed rectangle of panel a (c,d)".
Modified accordingly

Fig. 6: (a) Please (re-) state what "normalized" means in this case. (b) State the period over which the changes refer to. (c) "In the latter" –> "In panel b" (d) "because it is more than 150%" –> "(since the value is >150%)"
Modified accordingly

Fig. 7: State the period over which the changes refer to.
Modified accordingly

Fig. 8: The red markers should be crosses (area), and not dots (volume).

Modified accordingly

Fig. 10: (a) Please simplify the third sentence (the one starting with "In the transient state,: : :"). As far as I understand, it simply means that all but the blue w_e are taken from Vincent et al (2016)? (b) "ratio of net ablation" –> Not sure to understand that. A "ratio" is between something and something else, I would say.
Modified accordingly

Tab. 1: (a) "A_SD" never shows up in the table. Thus, no need of introducing the symbol. (b) "The main aspects" –> Why plural? If there is a share of aspects implied in what the table shows, please state that.
The cliffs exhibit multiple aspects. We made this point clearer in the revised version.

Tab. 2: (a) I don't understand the meaning of the reference. Just remove? (b) Please explain in the caption what "virtual" means. (c) In the caption, provide a hint for why some values are "N/A".
This is clarified in the revised version.

Tab. 3: What is "B/H"? The caption should tell.
It means "base to height ratio", it was added in the caption.

Anderson, L. S. and Anderson, R. S.: Modeling debris-covered glaciers: response to steady debris deposition, The Cryosphere, 10(3), 1105–1124, doi:10.5194/tc-10-1105-2016, 2016.

Banerjee, A.: Brief communication: Thinning of debris-covered and debris-free glaciers in a warming climate, The Cryosphere, 11(1), 133–138, doi:10.5194/tc-11-133-2017, 2017.

Benn, D. I., Bolch, T., Hands, K., Gulley, J., Luckman, A., Nicholson, L. I., Quincey, D., Thompson, S., Toumi, R. and Wiseman, S.: Response of debris-covered glaciers in the Mount Everest region to recent warming, and implications for outburst flood hazards, Earth-Sci. Rev., 114(1–2), 156–174, doi:10.1016/j.earscirev.2012.03.008, 2012.

[revised manuscript text omitted]

---

## Author Comment (AC4) · 4 Jul 2018

**Response to anonymous referee 4**

We thank the referee 4 for the detailed review. In this response, the reviewer's comments are in black standard font. Our response is in standard blue font and the modifications to the manuscript are in blue bold font.

Summary:
The authors estimate the total ablation associated to supraglacial ice cliff melt on the debris-covered tongue of Changri Nup Glacier, Nepalese Himalaya, based on high resolution topographical data. They use terrestrial photogrammetry surveys on selected cliffs for validation and UAV- and satellite-imagery for the entire cliff population on that glacier for two consecutive years. They then derive the contribution of ice cliff melt relative to glacier melt on the tongue of Changri Nup Glacier by taking into account emergence velocity and estimate ice cliff melt to be ~3 times higher than the average melt of the glacier tongue. They conclude that ice cliffs cannot explain the debris-cover anomaly and that the anomaly in turn could be a result of lower emergence velocities and reduced ablation.

General comments:
The main outcome of this study is that UAV- and especially high-resolution satellite imagery can be used to estimate glacier-wide volume losses associated to ice cliff melt, as the authors showed by a sophisticated analysis of various topographic datasets. The second important conclusion of the study is the fact that emergence velocities have so far not been considered carefully enough in terms of glacier ablation estimates and should be investigated further. However, as much as I appreciate the topic of the paper including all its careful analysis, I think the authors' main conclusion of explaining the debris-cover anomaly with reduced emergence velocities in general is not appropriate and a bit out of the context of the paper, given the limited sample size of just one glacier tongue. I suggest the authors adapt the title accordingly and focus more on the nice outcome of the cliff volume loss estimates at the glacier scale, especially in the discussion of the manuscript. Further, I think the balance between ice cliffs and emergence velocity is not given, as the main part of the paper regarding methods description and results, is mostly about ice cliffs and in contrast the discussion/conclusion part is mostly about emergence velocities. The processing of the ice cliff data is well described in general and the results are elaborated carefully by taking into account uncertainties. I miss a better description of one of the key improvements compared to an earlier study, the correction for distortion. Further, I am not convinced that the calculation of the p-factor is feasible or should be done in a different way. All in all my impression of the paper in terms of quality, writing, and relevance in the context of the actual literature is good and it fits into the scope of The Cryosphere, but I have major comments that should be addressed. If these are addressed, I am sure the paper will be a useful contribution to the glaciological community. See further comments below.
We thank referee 4 for her/his positive appreciation of our work. See our response to the comments below.

Main issues:

1) Explanation of "debris-cover anomaly"

I like that you bring the emergence velocity component into the focus of the analysis and interpretation of debris-covered glaciers. It is clear that this upward movement of parts of the glacier tongue has been neglected so far in most of the studies related to debris-covered glaciers especially. In the case of Changri Nup Glacier the emergence velocity is, based on your observations, very small and thus similar downwasting rates of debris-free and debris-covered glacier surfaces might be explained by this differential emergence. However, I am not convinced by the reasoning of the authors to generalize this result and explain the debris-cover anomaly solely by the confusion of glacier elevation changes and net ablation. It is not clear if Changri Nup Glacier has ablation rates

similar to that of debris-free glaciers (of the same elevation range) at all, but this would be the case for the debris-cover anomaly.

The "debris cover anomaly" (i.e. similar thinning rates over debris-covered and debris free glaciers at similar elevations, although ablation is expected to be reduced over debris covered glaciers compared with debris-free glaciers) is to our opinion an interesting but fuzzy concept, which has been used to motivate previous studies that looked for processes responsible for enhanced ablation on debris-covered tongues. Based on the data of Brun et al. (2017), we show that the thinning rates of debris-covered areas are comparable to thinning rates of debris-free areas for glaciers in the Khumbu region (Figure R1). The thinning rate of Changri Nup Glacier agrees well with this regional pattern and therefore we do conclude that the tongue of Changri Nup Glacier is a representative "debris-anomaly" glacier.

[Figure]

*Figure R1: rate of elevation change for debris-free and debris-covered ice in the Khumbu region, based on Brun et al. (2017) data. . The brown histogram represents the hypsometry of the debris-covered ice and it is stacked above the blue histogram, which represents the hypsometry of the debris-free ice. The thinning rate for the debris-covered part of Changri Nup is overlaid in grey.*

For sure emergence is an important component and might explain many issues related to this topic. But I think, just based on one single glacier making a general conclusion is too ambitious and therefore the title of the manuscript should be adapted accordingly, away from the debris-cover anomaly more towards the volume loss of the ice cliffs. The study still presents interesting points and provides nice results, such as: estimation of the volume loss associated to ice cliff melt at the glacier scale; incorporating rotational behavior of ice cliffs in volume loss estimates; sophisticated comparison to net ablation by taking into account also glacier emergence.

We agree with the reviewer's comment and now avoid unsupported generalization, as also recommended by other reviewers. Consequently, the title of the manuscript was revised and now reads: "**Ice cliff contribution to the tongue-wide ablation of Changri Nup Glacier, Nepal, Central Himalaya**". Additionally, section 6.3 regarding the debris-cover anomaly has been mostly rewritten, with much more caution regarding the generalization of our conclusions.

The manuscript now focuses more on the cliff evolution and contribution to tongue-wide ablation. Indeed, we extended the discussion section 6.1 ("**Cliff evolution and** comparison of two years of acquisition") and we added a table in the supplement (Table S2) showing the evolution of individual cliffs:

"**The total area occupied by ice did not vary significantly from year to year, ranging from 70 ± 14 × $10^3$ m$^2$ in November 2017 to 72 ± 14 × $10^3$ m$^2$ in November 2016. The twelve individual cliffs surveyed showed large variations in area within the course of one year, with a maximum increase of 57 % for the large cliff 06 and a decrease of 34 % for cliff 03 and 09 (Table S2). The total area of these twelve cliffs increased by 8 % in one year. Interestingly, over the same period, Watson et al. (2017) observed only declining ice cliff area on the tongue of Khumbu Glacier (~6 km away). All the large cliffs (most of them are included in the twelve cliffs surveyed with the terrestrial photogrammetry) persisted over these two years of survey, including the south or south-west facing ones (Table 1) , although south facing cliffs are known to persist less then non south facing ones (e.g., Buri and Pellicciotti, 2018). However, we observed the appearance and disappearance of small cliffs, and marginal areas became easier to classify as either ice cliff or debris-covered areas, highlighting the challenge in mapping regions covered by thin debris (e.g., Herreid and Pellicciotti, 2018).**

**We calculated backwasting rates for the twelve cliffs monitored with terrestrial photogrammetry for the period November 2015-November 2016 (Table 1). The backwasting rate is sensitive to cliff area changes (because it is calculated as the rate of volume change divided by the mean 3D area) and should be interpreted with caution for cliffs that underwent large area changes (e.g., cliffs 01, 02, 03, 06, 09 and 11; Table S2). The backwasting rates ranged from 1.2 ± 0.4 to 7.5 ± 0.6 m a$^{-1}$ , reflecting the variability in terms of ablation rates among the terrain classified as cliff (Fig. 9). The lowest backwasting rates are observed for cliffs 11 and 12, located on the upper part of the tongue, roughly 100 m higher than the other cliffs (Fig. 1 and Table 1). The largest backwasting rates were observed for cliff 01, which expanded significantly between November 2015 and November 2016. The backwasting rates are lower than those reported by Brun et al. (2016) on Lirung Glacier (Langtang catchment) for the period May 2013-October 2014, which ranged from 6.0 to 8.4 m a$^{-1}$ and lower than those reported for surviving cliffs by Watson et al. (2017) on Khumbu Glacier for the period November 2015-October 2016, which ranged from 5.2 to 9.7 m a$^{-1}$. These differences are likely due to temperature differences between sites. Indeed, the cliffs studied here are at higher elevation (5320-5470 m a.s.l.) than the two other studies (4050-4200 m a.s.l. for Lirung Glacier and 4923-4939 m a.s.l. for Khumbu Glacier).**"

2) Calculation of p
I am not sure if your calculation of the p ratio makes much sense. Like this you compare ice cliff melt to the melt of the entire debris-covered glacier tongue. This means you compare ice cliff melt to a mixture of subdebris- and ice cliff melt (the glacier tongue). Wouldn't it be more feasible to compare ice cliff melt to subdebris melt (i.e, exclude ice cliff areas)? In order to be comparable to the previous studies you cited, you should check if they also compared ice cliff melt to the ice cliff-subdebris mixture or to subdebris melt only.

This point was also mentioned by other reviewers, consequently the $p$ factor is now named $f_C$ factor to avoid a confusion with the "$p$-value". The $f_C$ factor has the same definition as $p$, but we added the definition of a new factor, named $f_C^*$, which is the ratio of the cliff ablation divided by the non-cliff ablation (denoted by the subscript NC):

$$f_C^* = \frac{\Delta V_C}{A_C} \frac{A_{NC}}{\Delta V_{NC}} = f_C \frac{\Delta V_T}{\Delta V_T - \Delta V_C} \frac{A_T - A_C}{A_T}$$

Based on our data, for Changri Nup Glacier, $\frac{\Delta V_T}{\Delta V_T - \Delta V_C} = \frac{1}{1-0.23} = 1.30$ and $\frac{A_T - A_C}{A_T} = \frac{1-0.07}{1} = 0.93$, consequently $f_C^* = 1.2\, f_C$.

For the consistency between the studies mentioned, we interpreted the studies as follow:
- Juen et al. 2014 : "Although the ice cliffs occupy only 1.7% of the debris covered area, the melt amount accounts for approximately 12% of the total sub-debris ablation" -> $f_C^* = \frac{12}{1.7} = 7.1$
- Reid and Brock 2014: "Analysis of the DEM indicates that ice cliffs account for at most 1.3% of the 1m pixels in the glacier's debris-covered zone, but application of a distributed model indicates that ice cliffs account for ~7.4% of total ablation." -> $f_c = \frac{7.4}{1.3} = 5.7$
- Buri et al. 2016: "Although only representing 0.09% of the glacier tongue area, the total melt at the two cliffs over the measurement period is 2313 and 8282m3, 1.23% of the total melt simulated by a glacio-hydrological model for the glacier's tongue. -> $f_c = \frac{1.23}{0.09} = 13.7$
- Sakai et al 1998: From the abstract: "The ice cliff melt amount reaches 69% of the total ablation at debris covered area, although the area of ice cliffs occupies less than 2% of the debris covered area" -> $f_c = \frac{69}{2} = 35$
- Sakai et al 2000: From their Table 2: ratio of the "absorbed heat at each type of surface during the observation period (167 days)", including the "whole debris-covered zone" -> $f_c = \frac{256}{26} = 9.8$ or looking only at the debris -> $f_C^* = \frac{256}{21} = 12.2$
- Brun et al 2016: "The ice cliffs lose mass at rates six times higher than estimates of glacier-wide melt under debris, which seems to confirm that ice cliffs provide a large contribution to total glacier melt." -> $f_c = 6$
- Thompson et al 2016: "Although ice cliffs cover only ~5% of the area of the lower tongue, they account for 40% of the ablation." -> $f_c = \frac{40}{5} = 8$

As most of the studies were already within the framework of the original definition of $p/f_C$, we decided to keep the focus on this factor, instead of $f_C^*$. This example demonstrates the importance of a consistent framework for comparing these studies.

Consequently, we decided to present both factors $f_c$ and $f_c*$ in the revised manuscript, in order to avoid any confusion.

Also, it is not clear how many ice cliffs you detected on the entire debris-covered glacier tongue in both seasons, does this number vary? Did you observe the formation of new features over time or disappearance of them? This might also affect the p ratio, in case the cliff areas are not excluded in its calculation.
We added some information regarding the year-to-year evolution of the cliffs, in section 6.1 (see above). For the calculation of the factors $f_c$ and $f_c*$ we used the cliff footprint. Consequently, the dynamic changes in the cliff areas are taken into account.

Additionally, could you derive melt rates for the cliff surface (perpendicular to their surface) instead of pure elevation change rates? This would be helpful in order to compare your results to previous studies.
We added the backwasting distance in Table 1. We calculated it as individual cliff volume loss from terrestrial photogrammetry (i.e., only for the period Nov. 2015 – Nov. 2016), because this is the acquisition for which we know the mean cliff 3D area. The backwasting rate is compared with other studies in section 6.1.

3) Correction with local field of displacement

From the text I cannot follow how the rotational component of the ice cliffs/glacier surface are implemented in the calculations of the ice cliff volume changes (Section. 4.2). Since this is a main improvement of this study compared to an earlier one using a similar method, much more emphasis should be given to explain this implementation. How do you use the local field of displacement? Where does the 3D flow field that you use for the correction come from? This is a key comment that should be addressed for the paper clarity. A schematic figure would be helpful. Also, can you estimate how much the difference is compared to the previous method used in Brun et al. 2016? More importantly, can you validate the new method to show that it is appropriate and sound? From the short description provided: i) the method cannot be reproduced; ii) there is no evidence that it is the correct approach. You should discuss this new method in the discussion part of the manuscript, as it seems to be a clear advancement from previous studies, where a simple homogeneous direction of displacement was assumed.

Actually, taking into account the heterogeneity in the field of displacement has only little effect on the outcomes, because finally in our case study of Changri Nup Glacier, the rotational effect is limited. We added a supplementary figure (Fig. S5, see below) showing this. We added: "**This would be an important methodological refinement for ice cliffs on fast flowing glaciers with a rotational component, but has minor influence for the cliffs of interest in this study (Fig. S5)**". The main interest on this new method is its ability to handle gridded data, while the method of Brun et al. (2016) worked only with 3D (TINs) data. Taking into account the rotational effect is still an important improvement compared with Brun et al. (2016) especially if the method is used on fast moving and turning glaciers, which is not the case on Changri Nup or on Lirung glaciers (this study and Brun et al., 2016). Figure S5 helps also to visualize the method's development as well.

[Figure]

Fig. S5 - Examples of the methodological processing for cliff 05, located on a slow flowing area (left panels) and cliff 11, located in a fast flowing area (right panels). For all the panels the cliff outlines are represented in UTM45/WGS84. a- influence of the glacier flow correction, and comparison with a uniform translation. B- example of analogous points needed for the triangulation regularization. c- difference between the individual cliff outlines and the cliff footprint needed to calculate the cliff contribution for gridded data (DEMs).

4) Uncertainty of emergence velocity assumptions
As mentioned above in the general comments, the balance between ice cliff associated melt estimates and emergence velocity assumptions is not well elaborated in the manuscript. E.g. I miss a more in depth discussion of the uncertainties related to the calculation of the emergence velocities in the methods section, such as the one associated with having only one cross section profile for

glacier thickness estimates, or general uncertainties in ice flux assumptions (glacier thickness measurements, bed topography, subglacial conditions, distribution of emergence etc.). As stated in line 19, p.10, "the main source of uncertainty on the cliff volume change is the uncertainty on the emergence velocity", but this is not discussed later on in the text.

We added a paragraph about the uncertainty in the calculation for the emergence velocity (section 4.1):

"The emergence velocity refers to the upward flux of ice relative to the glacier surface in an Eulerian reference system (Cuffey and Paterson, 2010). For the case of a glacier in steady-state (i.e., no volume change at the annual scale), the emergence velocity balances exactly the net ablation for any point of the glacier ablation area (Hooke, 2005). For a glacier out of its steady state (as Changri Nup Glacier) the thinning rate observed in the ablation area is the sum of the net ablation and the emergence velocity (Hooke, 2005). On debris-covered glaciers, while the thinning rate is relatively straightforward to measure from DEM differences, for example, the ablation is highly spatially variable and difficult to measure (e.g., Vincent et al., 2016). In order to evaluate the mean net ablation of Changri Nup Glacier tongue from the thinning rate, we estimate the mean emergence velocity ($w_e$) for the period November 2015-November 2016 and for the period November 2016--November 2017 using the flux gate method of Vincent et al. (2016). As the ice flux at the glacier front is 0, the average emergence velocity downstream of a cross-section can we calculated as the ratio of the ice flux through the cross-section ($\Phi$ in m$^3$ a$^{-1}$), divided by the glacier area downstream of this cross-section ($A_T$ in m$^2$):**

$$w_e = \frac{\Phi}{A_T}$$

**This method requires an estimate of ice flux through a cross-section of the glacier, and is based here on measurements of ice depth and surface velocity along a profile upstream of the debris-covered tongue (Figs. 1 and 2). The ice flux is the product of the depth-averaged velocity ($\bar{u}$ in m a$^{-1}$) and the cross-sectional area. For the period November 2015-November 2016 (resp. November 2016-November 2017), the glacier slowed down compared with the 2011-2014 period and the centerline velocity was equal to 10.8 m a$^{-1}$ (resp. 11.1 m a$^{-1}$), leading to an assumed mean surface velocity along the upstream profile of 8.1 ± 0.6 m a$^{-1}$ (resp. 8.3 ± 0.6 m a$^{-1}$), as the centerline velocity is usually 70 to 80 % of the mean surface velocity along the cross-section (e.g., Azam et al., 2012; Berthier and Vincent, 2012). We used the relationship between the centerline velocity and the mean velocity, instead of an average of the velocity field along the cross section, because the image correlation was not successful on a relatively large fraction (~ 30 %) of the cross section. Converting the surface velocity into a depth-averaged velocity requires assumptions about e basal sliding and a flow law (Cuffey and Paterson, 2010). Little is known about the basal conditions of Changri Nup Glacier, but Vincent et al. (2016) assumed a cold base, and therefore no sliding. This leads to $\bar{u}$ being approximated as 80 % of the surface velocity, additionally assuming n = 3 in Glen's flow law (Cuffey and Paterson, 2010). As an end-member case, assuming that the motion is entirely by slip implies $\bar{u}$ equals to the surface velocity (Cuffey and Paterson, 2010). Consequently, we followed Vincent et al. (2016) and assumed no basal sliding, but we took the difference between the two above-mentioned cases as the uncertainty on $\bar{u}$. This leads to $\bar{u}$ = 6.5 ± 1.6 m a$^{-1}$ (resp. 6.6 ± 1.7 m a$^{-1}$) for the period November 2015-November 2016 (resp. November 2016-November 2017).**

**Assuming independence for the cross-sectional area ($\sigma_S$) and the depth-averaged velocity ($\sigma_{\bar{u}}$), the uncertainty on the ice flux ($\sigma_\Phi$) can be estimated as:**

$$\frac{\sigma_\Phi}{\Phi} = \sqrt{\frac{\sigma_{\bar{u}}^2}{\bar{u}} + \frac{\sigma_S^2}{S}}$$

**Given the above mention values for the depth-averaged velocity, the cross-sectional area and the associated uncertainties, the relative uncertainty of the ice flux is ~30 %. As a result, for the period**

**November 2015-November 2016 (resp. November 2016-November 2017), the incoming ice flux was thus 499 700 ± 150 000 m$^3$ a$^{-1}$ (resp. 503 840 ± 150 000 m$^3$ a$^{-1}$). The glacier tongue area was considered unchanged at 1.49 ± 0.16 km$^2$, corresponding to $w_e$ = 0.33 ± 0.11 m a$^{-1}$ (resp. 0.34 ± 0.11 m a$^{-1}$). It is notoriously difficult to delineate debris-covered glacier tongues (e.g., Frey et al., 2012). In this case, we assumed an uncertainty in the outline position of ± 20 m, leading to a relative uncertainty in the glacier area of 11 %, which is higher than the 5 % of Paul et al. (2013). In this case, the uncertainty on the glacier outline is not the main source of uncertainty in $w_e$, but for automatically delineated glacier outlines, this would be an important source of uncertainty. The updated emergence velocity is ~20 % lower than estimated for the 2011-2015 period (Vincent et al., 2016), due to both the thinning and deceleration of the glacier. As the difference in $w_e$ between November 2015-November 2016 and November 2016-November 2017 is insignificant, we consider $w_e$ to be constant and equal to $w_e$= 0.33 ± 0.11 m a$^{-1}$ for the rest of this study. It is noteworthy that some spatial variability is expected for $w_e$, however, we have no means to assess it."**

Also, the reduction in emergence velocity of _20% compared to the period 2010-2015 (line 16, p. 6) is striking, isn't it? Can you try to explain it more convincingly? If glacier emergence is so (relatively) variable, this might also have implications on your assumption of generally explaining the debris-cover anomaly by lower emergence velocities.

This reduction in emergence velocity is likely due to negative mass balances over this 2010-2017 period. Indeed, over the period 2010-17, we observe a strongly continuous negative glacier-wide mass balance of the debris-free West Changri Nup glacier of -1.36 m w.e. yr$^{-1}$ (-1.24 m w.e. yr$^{-1}$ between 2010 and 2015 from Sherpa et al., 2017; and unpublished data from P. Wagnon, for the period 2015-2017). Since Changri Nup and West Changri Nup glaciers are located near-by and since the time lag between negative surface mass balances and decreasing velocities is expected to be short (Vincent et al., 2000), the reduction in velocity between 2010-15 and 2015-17, and as a consequence in emergence velocity, is not surprising on Changri Nup glacier.
Nevertheless, we agree that the emergence velocity is likely to be variable both in time and space over Changri Nup Glacier (and elsewhere), but we do not have any clue to quantify this variability. That is the reason why we performed a sensitivity test over this emergence velocity, using a very large range of possible values (section 4.3.1 and fig. 6).

Azam, M. F., Wagnon, P., Ramanathan, A., Vincent, C., Sharma, P., Arnaud, Y., Linda, A., Pottakkal, J. G., Chevallier, P., Singh, V. B. and Berthier, E.: From balance to imbalance: a shift in the dynamic behaviour of Chhota Shigri glacier, western Himalaya, India, J. Glaciol., 58, 315–324, doi:10.3189/2012JoG11J123, 2012.

Berthier, E. and Vincent, C.: Relative contribution of surface mass-balance and ice-flux changes to the accelerated thinning of Mer de Glace, French Alps, over 1979-2008, J. Glaciol., 58, 501–512, doi:10.3189/2012JoG11J083, 2012.

Brun, F., Buri, P., Miles, E. S., Wagnon, P., Steiner, J. F., Berthier, E., Ragettli, S., Kraaijenbrink, P., Immerzeel, W. W. and Pellicciotti, F.: Quantifying volume loss from ice cliffs on debris-covered glaciers using high-resolution terrestrial and aerial photogrammetry, J. Glaciol., 62(234), 684–695, doi:10.1017/jog.2016.54, 2016.

Brun, F., Berthier, E., Wagnon, P., Kaab, A. and Treichler, D.: A spatially resolved estimate of High Mountain Asia glacier mass balances from 2000 to 2016, Nat. Geosci., 10, 668–673, 2017.

Cuffey, K. M. and Paterson, W. S. B.: The physics of glaciers, Academic Press., 2010.

Frey, H., Paul, F. and Strozzi, T.: Compilation of a glacier inventory for the western Himalayas from satellite data: methods, challenges, and results, Remote Sens. Environ., 124, 832–843, doi:http://dx.doi.org/10.1016/j.rse.2012.06.020, 2012.

Herreid, S. and Pellicciotti, F.: Automated detection of ice cliffs within supraglacial debris cover, The Cryosphere, 12(5), 1811–1829, doi:10.5194/tc-12-1811-2018, 2018.

Hooke, R. L.: Principles of glacier mechanics, Cambridge university press., 2005.

Paul, F., Barrand, N. E., Baumann, S., Berthier, E., Bolch, T., Casey, K., Frey, H., Joshi, S. P., Konovalov, V., Le Bris, R., Mölg, N., Nosenko, G., Nuth, C., Pope, A., Racoviteanu, A., Rastner, P., Raup, B., Scharrer, K., Steffen, S. and Winsvold, S.: On the accuracy of glacier outlines derived from remote-sensing data, Ann. Glaciol., 54, 171–182, doi:10.3189/2013AoG63A296, 2013.

Sherpa, S. F., Wagnon, P., Brun, F., Berthier, E., Vincent, C., Lejeune, Y., Arnaud, Y., Kayastha, R. B. and Sinisalo, A.: Contrasted surface mass balances of debris-free glaciers observed between the southern and the inner parts of the Everest region (2007-2015), J. Glaciol., 63(240), 637–651, 2017.

Vincent, C., Vallon, M., Reynaud, L. and Le Meur, E.: Dynamic behaviour analysis of glacier de Saint Sorlin, France, from 40 years of observations, 1957-97, J. Glaciol., 46, 499–506, doi:10.3189/172756500781833052, 2000.

Vincent, C., Wagnon, P., Shea, J. M., Immerzeel, W. W., Kraaijenbrink, P., Shrestha, D., Soruco, A., Arnaud, Y., Brun, F., Berthier, E. and Sherpa, S. F.: Reduced melt on debris-covered glaciers: investigations from Changri Nup Glacier, Nepal, The Cryosphere, 10(4), 1845–1858, doi:10.5194/tc-10-1845-2016, 2016.

Watson, C. S., Quincey, D. J., Smith, M. W., Carrivick, J. L., Rowan, A. V. and James, M. R.: Quantifying ice cliff evolution with multi-temporal point clouds on the debris-covered Khumbu Glacier, Nepal, J. Glaciol., 63(241), 823–837, 2017.

---

## Author Comment (AC5) · 4 Jul 2018

**Summary of the response to the four reviewers' comments**

We thank the scientific editor and the four reviewers for their detailed reviews of our article submitted to The Cryosphere. The main improvements in the manuscript are:

- As suggested by reviewer 3, we processed additional data in order to improve the discussion section about the debris cover anomaly (section 6.3). The discussion section 6.3 has been substantially modified (see below and see the detailed responses to each of the reviewers' comments), in particular we are now more careful about the generalization of our findings. However, we added a figure (fig. 10) showing the minimum glacier elevation as a function of the debris cover percentage (based on the data of Kraaijenbrink et al., 2017). The minimum glacier elevation decreases with the percentage of debris cover. We interpret this relationship as an indirect hint that debris-covered tongues are larger than debris-free tongues and that the ablation is reduced on these debris-covered tongues, because they can exist at lower elevation than the debris-free tongues.

[Figure]

*Figure 10: Glacier minimum elevation as a function of the percentage of debris cover for the glaciers larger than 2 km² in High Mountain Asia. The black crosses represent individual glaciers and the red diamonds shows the mean of the glacier minimum elevation. For instance, the first diamond represent the mean of the glacier minimum elevation for glaciers with a percentage of debris cover between 0 and 0.51% (5th percentile).*

- We added a supplementary figure S5, illustrating some specific aspects of our new methodological developments

[Figure]

*Figure S5 - Examples of the methodological processing for cliff 05, located on a slow flowing area (left panels) and cliff 11, located in a fast flowing area (right panels). For all the panels the cliff outlines are represented in UTM45/WGS84. a-influence of the glacier flow correction, and comparison with a uniform translation. B- example of analogous points needed for the triangulation regularization. c- difference between the individual cliff outlines and the cliff footprint needed to calculate the cliff contribution for gridded data (DEMs).*

- We added a supplementary table S2, showing the changes in area between Nov. 2015 and Nov. 2016 for the twelve cliffs surveyed with the terrestrial photogrammetry

Tab. S2 – 3D area changes of the twelve field monitored cliffs

| Cliff ID | 3D area 2015 [m²] | 3D area 2016 [m²] | Relative area change (%) |
|----------|-------------------|-------------------|--------------------------|
| Cliff 01 | 6126 | 8961 | 46 |
| Cliff 02 | 1135 | 1496 | 32 |
| Cliff 03 | 3650 | 2415 | -34 |
| Cliff 04 | 1915 | 1788 | -7 |
| Cliff 05 | 11323 | 11265 | -1 |
| Cliff 06 | 4099 | 6435 | 57 |
| Cliff 07 | 749 | 756 | 1 |
| Cliff 08 | 1286 | 1278 | -1 |
| Cliff 09 | 2897 | 1918 | -34 |
| Cliff 10 | 2659 | 2192 | -18 |
| Cliff 11 | 466 | 707 | 52 |
| Cliff 12 | 818 | 732 | -11 |
| **Total** | **37124** | **39942** | **8** |

- The cliff ablation enhancement factor, named *p* in our original submission is now named $f_c$ to avoid any confusion with the "p-value" as suggested by reviewer 3. Following, the suggestion of reviewer 1 and 4, we added the definition and computation of the $f_C^*$ factor, which compares the cliff and non cliff ablation (instead of the cliff and whole glacier tongue).
- We changed the structure of parts of the data and method sections. Section 3.4.2 is now entitled "Ground penetrating radar data", and the method section is now separated into three main subsections: 4.1-Emergence velocity; 4.2-Ice cliff backwasting calculation; 4.3-Sources of uncertainty on the ice cliff backwasting.
- In order to better balance the focus of the paper (comments from reviewers 1, 2 and 4), we extended the section 6.1 with a description of the cliff evolution and compared backwasting rates with published values. We extended Table 1 with values of mean elevation and backwasting rates for individual cliffs.
- Reviewers 1, 3 and 4 legitimately criticized the extrapolation we made based on a single glacier. We substantially modified section 6.3 ("Ice cliff ablation and the debris-cover anomaly"), in order to modify our previous statements, which were probably too strong with regards to the small sample studied here (n=1, as pointed out by reviewer 3). We changed the title of the paper, which is now "Ice cliff contribution to the tongue-wide ablation of Changri Nup Glacier, Nepal, Central Himalaya". Moreover, we backed up some of our theoretical arguments, based on a compilation of data from Kraaijenbrink et al. (2017) shown in figure 10.

Additional changes:

- The family name of Dibas Shrestha was misspelled (missing "h") in the original submission

- Silvan Raggetli brought to our attention that the "debris-cover anomaly" was never observed in the Langtang catchment, due to insufficient hypsometric overlap between debris-free and debris-covered ice. We modified the text accordingly.
- The signs greater than and smaller than were inverted in equation 5. It is corrected in the revised version.

---

## Referee Report (RR1)

*Review of* **"Ice cliff contribution to the tongue-wide ablation of Changri Nup Glacier, Nepal, Central Himalaya"** *by Brun et al.*

**Main comment:**

The authors have made substantial adaptions to the previous version of the manuscript in a detailed response to the initial reviews. The clarity in the new version of the manuscript and in their response to the reviewers is greatly acknowledged.
Specifically, in response to my review, the authors have:
1) changed the title of the paper and rewritten a section regarding the debris-cover anomaly in order to avoid unsupported generalization of their results;
2) put more emphasis on cliff evolution and the cliffs' contribution to the glacier ablation;
3) greatly clarified the former term $p$ (now $f_c$)
4) added better explanations of their methodological processing for two ice cliffs, especially by showing Fig. S5.
5) clarified the uncertainty in emergence velocity estimates and its reduction over time.

With these modifications and those in response to the other reviews, the resulting manuscript is a substantial contribution to the field.

**Technical corrections:**

P3, L4: …rates than the rest…
P4, L24: …we refer to Vincent…
P7, L16: …of the basal sliding…

---

## Editor Decision (ED1)

[revised manuscript text omitted]

---

## Author Response (AR2)

Dear Francesca,

We thank you for your positive appreciation of our work and of our efforts to address the comments of the reviewers. We are happy to see that our original publication triggered a vivid debate within the debris-covered glacier community, and we believe that the review process clearly improved our article.

We thank the two reviewers who evaluated our manuscript a second time and made very relevant comments. Please find our point-by-point response to their comments below and, attached, the revised manuscript.

Best regards,

Fanny and co-authors

**Response to anonymous referee #4**

The authors have made substantial adaptions to the previous version of the manuscript in a detailed response to the initial reviews. The clarity in the new version of the manuscript and in their response to the reviewers is greatly acknowledged.

Specifically, in response to my review, the authors have:

1) changed the title of the paper and rewritten a section regarding the debris-cover anomaly in order to avoid unsupported generalization of their results;

2) put more emphasis on cliff evolution and the cliffs' contribution to the glacier ablation;

3) greatly clarified the former term p (now fc)

4) added better explanations of their methodological processing for two ice cliffs, especially by showing Fig. S5.

5) clarified the uncertainty in emergence velocity estimates and its reduction over time.

With these modifications and those in response to the other reviews, the resulting manuscript is a substantial contribution to the field.

We thank anonymous reviewer #4 for this second evaluation of our manuscript and her/his positive appreciation of our work.

Technical corrections:

P3, L4: …rates than the rest…

Modified accordingly

P4, L24: …we refer to Vincent…

Modified accordingly

P7, L16: …of the basal sliding…

Modified accordingly

**Response to anonymous referee #1**

The manuscript by Brun et al has improved and they have addressed the reviewers' comments (including my own) carefully. I appreciate the changes the authors have made to balance the manuscript in terms of content, more carefully account for uncertainty in emergence flux, and more clearly present their hypothetical explanation of the debris-cover anomaly, along with the opportunities for future research to better generalise its implications. My comments at this point are primarily for clarity of the manuscript, rather than major changes to analysis or discussion.

We thank anonymous reviewer #1 for this second evaluation of our manuscript and her/his positive appreciation of our work. Modifications to the text are highlighted in bold font.

P2L7-8. May want to make it clear that this is still part of the 'first' hypothetical explanation. Also, order inconsistent for the citations

We added "**Additionally,** other processes…" to link this sentence to the previous one.

P2L8. May want to state 'Second'

Modified accordingly

P2L15-25. It may be helpful to note that these two enhancement factors are useful in very distinct ways. f_c is a useful perspective for DEM-differencing (you know the total volume loss, but not the portion attributable to cliffs) while f_c* is more meaningful for modelling (you can calculate subdebris ablation and scale with f_c* to estimate the additional melt due to cliffs). In either case you need to know the cliffs' area.

Re-written accordingly. The new paragraph reads as "$f_C^*$ has the advantage of not including the ice cliff contributions in the total tongue ablation**, it is thus useful for modeling studies where sub-debris and cliff ablation are estimated independently or in order to scale the ice cliff ablation from the sub-debris ablation.** $f_C$ has the advantage of being directly linked to the total ice cliff contributions to ablation."

P2L33. Remove 'typically', as emergence velocity has been neglected in all such cases.

Modified accordingly

P3L1-3. The Nuimura et al 2011 values for Khumbu are difficult to compare to the other two because they are derived much farther up-glacier. An option to consider would be to mention that emergence velocities are usually strongly positive in the upper ablation area (i.e. just below an icefall, as in the observations of Nuimura et al 2011) but decaying towards the terminus (as in the other two studies). Another option would be to reference Rounce et al (2018) who use the flux-gate method to estimate divergence flux for the whole glacier length.

Here we politely disagree with the reviewer's interpretation of decaying emergence velocities towards the terminus. Data from Nuimura et al. (2017) show a rather constant emergence velocity on the debris-covered tongue of Lirung glacier (see their figure 6c below), which is consistent with the fact that the surface mass balance of debris-covered tongues is probably highly variable on small spatial scales (~100 m), but not very dependent on the elevation (e.g., Banerjee and Shankar, 2013). Consequently, the pattern of modeled flux divergence obtained by Rounce et al. (2018) could be an artifact due to the fact that they rely on ice thickness data obtained from modeling, which itself relies on strong hypothesis on the surface mass balance gradient (Huss and Farinotti, 2012).

[Figure]

*Figure R1: Longitudinal profiles of components in the continuity equation for the four areas of the debris-covered area of Lirung Glacier (figure 6c from Nuimura et al., 2017)*

However, we agree with the reviewer that the three emergence velocities we cited do not match each other exactly, as one of them is the debris-covered tongue average (Vincent et al., 2016) and the others are calculated for various sections of debris-covered tongues (Nuimura et al., 2011, 2017). We added this sentence: "**However, we stress the fact that these emergence velocities have been measured at different locations of these debris-covered tongues (in particular close to the clean ice/debris transition on Khumbu Glacier), on glaciers with very different dynamics.**"

P3L24. Suggest …through avalanche deposition…

Modified accordingly

P3L10. Suggest '…and thus can only be acquired…'

Modified accordingly

P4L21. Change for clarity to '… were spatially well-distributed…' or simply 'were spatially distributed'

Modified accordingly (spatially well-distributed)

P6L15. Suggest this section be named 'Cross section ice thickness' as this is the data you are updating (rather than the GPR data itself).

Modified accordingly

P6L21-23. I appreciate the modifications that authors have undertaken to better account for uncertainty in their emergence velocity estimates. I think a slight further modification (to the text only) is needed. The location is slightly different to that of Azam et al (2012), and this uncertainty of ice thickness depends on ice thickness itself (ie larger magnitude of uncertainty for thicker ice), the properties of the ice, and post-processing settings. Simply pulling the +/-15m value from Azam et al (2012) thus may not transfer well. As this is not an unreasonable value given the ice characteristics, you can probably leave the value as-is. But please revise the text to indicate clearly that, considering the glaciers to be reasonably comparable, you estimate this value as +/-15 m, as in Azam et al (2012).

Modified accordingly, the paragraph now reads: "Following **Azam et al. (2012), who measured the ice thickness of Chhota Shigri Glacier (15.48 km² flowing from 5830 to 4050 m a.s.l. with a maximum measured ice thickness of ~270 m) using the same methods,** we estimated that the absolute uncertainty on the ice thickness is ± 15 m,…"

P6L28. …out of its steady state ('such' as Changri Nup)…

Modified accordingly

P8L25. I suggest you use the term 'longitudinal gradient' rather than 'glacier mean slope', which is quite different.

Modified accordingly

P13L17. Worth noting that cliff total area did not change significantly 'between your November orthoimages' – it may also have changed during the seasons as cliffs became buried by debris and others emerged. This is an important (but understandable) limitation of the study, as of course you could not observe the cliffs continuously. However, you do see signs of cliffs disappearing or emerging, which indicates that your estimates may be slightly lower than the total cliff-associated contribution. There could also be transient cliffs that emerge and disappear within 1 year, which would only show as a zone of heightened elevation loss. This would be clarified by, for example, including your cliff outlines on either Figure 4 or 5 (possibly as a third column).

Thanks for the suggestion, we added the cliff footprint on figure 4, which now looks as:

[Figure]

*Figure R2: Panels showing maps of elevation change from UAV (**a, d**) before flow correction and (**b, c, e, f**) after flow correction over the period 23/11/2015--16/11/2016. **Black outlines***

*on panels c and f are the cliff footprints. Panels d, e and f are zooms of the panels a, b and c, respectively.*

P14L14. For Changri Nup Glacier. It will be interesting to see how these values compare to those for other glaciers. You not the low backwasting rates (e.g. relative to Lirung and Khumbu) and the higher elevation of Changri Nup. Do you think this may be a key part of the reason that your enhancement factors are so much lower? I.e. no doubt that emergence velocity plays a role (as you demonstrate) but of course enhancement factors will vary based on climatological setting. Possibly worth including a line in your discussion.

This is a very interesting point, thanks for sharing your thoughts about it. Spontaneously, we would think that the enhancement factor should be quite invariable… but it's true that the different setting of Changri Nup Glacier can explain part of the difference. We are a bit reluctant to compare our estimates of $f_c$ and $f_C^*$ with Brun et al. (2016) and Thompson et al. (2016) because they did not correct for the emergence velocity, and consequently we compared our $f_c$ and $f_C^*$ values not corrected from emergence velocity at the end of section 6.2: "**The values of $f_c$ and $f_C^*$ not corrected from the emergence velocity can be compared to the previous observational estimates. Both Brun et al. (2016) and Thompson et al. (2016) found values higher than our estimates. Part of the difference might arise from the different climatological settings, as Lirung and Ngozumpa glaciers are located at lower elevation than Changri Nup Glacier.**"

P15L4-6. These processes are certainly in play for Changri Nup, they 'do apply'. What 'does not apply' is the idea that these processes (and cliffs) bring ablation for the debris-covered area to the same level as for debris-free ice. Please clarify this sentence. Also, the debris-covered area 'experiences' a reduced ablation rate (not 'responsible').

We rephrased this sentence as "**Yet the contribution of these processes is not sufficient to enhance the ablation of the debris-covered tongue of Changri Nup Glacier at the level of clean ice ablation, as Vincent et al. (2016) already showed that the insulating effect of debris dominates for this glacier.**"

P15L25. I assume that this is Pearson's r? Perhaps italicise for clarity.

Modified accordingly

P15L26. Glaciers can also advance to lower elevations if they have a higher balance flux (e.g. larger accumulation area). Thus, it is the combination of the Scherler (2011) results with Fig 10 that make your point – I think this needs to be made clearer.

We added "Additionaly" at the sentence "**Additionally**, based on the data of…" and clarified the last sentence of the paragraph "**The combination of these two observations hints** at both reduced ablation and a larger tongue for debris-covered glaciers."

P15L27. Suggest '…that the ablation area of glaciers with considerable debris-cover is usually larger than for those without, …'. As written, the text does not account for glaciers with small or 'minor' debris-covered areas. No formal definition of a debris-covered glacier is given for this discussion, for example. A bit of revision as suggested will clarify that you're discussing only these glaciers with considerable debris mantles.

Modified accordingly

P16L5. …potentially important share…

Modified accordingly

P16L12. Yes, for instance differences in glacier and DC area hypsometry may have a considerable effect on f_c.

We added the sentence "**The definition of debris-covered tongues, the nature of their surface and their hypsometry might have a considerable effect on** $f_c$**.**"

P16L29. Suggest 'along englacial conduits' – the conduits themselves do not lead to ablation, but water running through them.

Modified accordingly

P16L32. I think you still need to be explicit that the cliffs play an important role in this – without them (but accounting for the emergence velocity) thinning rates would not be at parity for debris-covered and debris-free areas. Thus it is the combination of the hotspots of ablation with reduced emergence velocities that 1) heighten ablation above the Ostrem curve estimates and 2) alias the heightened ablation as equivalent thinning. Both aspects are important.

We added this sentence at the end of the paragraph: "**However, ice cliffs are still hot-spots of ablation and consequently of enhanced thinning; without them, the thinning rates of debris-covered and clean ice might not be similar.**"

P17L5. I think this is largely due to the near-constant coverage of cliffs over your study period. It would also be very interesting to see how variable cliff coverage is overall, as this will further inform whether cliffs always account for ~25% of ablation.

We specified this constant "Though our results cover only two years of data, **the area occupied by ice cliffs and their relative contribution to ablation (**$f_c$**) remained almost constant while** net ablation totals **differed** by 25%."

Banerjee, A. and Shankar, R.: On the response of Himalayan glaciers to climate change, J. Glaciol., 59, 480–490, doi:10.3189/2013JoG12J130, 2013.

Brun, F., Buri, P., Miles, E. S., Wagnon, P., Steiner, J. F., Berthier, E., Ragettli, S., Kraaijenbrink, P., Immerzeel, W. W. and Pellicciotti, F.: Quantifying volume loss from ice cliffs on debris-covered glaciers using high-resolution terrestrial and aerial photogrammetry, J. Glaciol., 62(234), 684–695, doi:10.1017/jog.2016.54, 2016.

Huss, M. and Farinotti, D.: Distributed ice thickness and volume of all glaciers around the globe, J. Geophys. Res. Earth Surf., 117(F4), F04010, doi:10.1029/2012JF002523, 2012.

Nuimura, T., Fujita, K., Fukui, K., Asahi, K., Aryal, R. and Ageta, Y.: Temporal Changes in Elevation of the Debris-Covered Ablation Area of Khumbu Glacier in the Nepal Himalaya since 1978, Arct. Antarct. Alp. Res., 43(2), 246–255, doi:10.1657/1938-4246-43.2.246, 2011.

Nuimura, T., Fujita, K. and Sakai, A.: Downwasting of the debris-covered area of Lirung Glacier in Langtang Valley, Nepal Himalaya, from 1974 to 2010, Quat. Int., doi:http://dx.doi.org/10.1016/j.quaint.2017.06.066, 2017.

Rounce, D. R., King, O., McCarthy, M., Shean, D. E. and Salerno, F.: Quantifying debris thickness of debris-covered glaciers in the Everest region of Nepal through inversion of a sub-debris melt model, J. Geophys. Res. Earth Surf., 2018.

[revised manuscript text omitted]

---

## Author Response (AR3)

Dear Francesca,

Thank you for your thorough reading of our paper. Your comments greatly improved the clarity of the paper and the quality of the English. We added a reference to Miles et al. in press, which is cited as Miles et al. 2018 in the paper because this was the automatic format provided by GRL website. We modified the text as you suggested everywhere.

Additionally, Joe Shea corrected the English in multiple places and, as for all papers published in TC, the manuscript will undergo a typesetting and correction process, while being edited. This will certainly guarantee the quality of the English.

You will find a track-change version of the manuscript appended below.

Best regards,

Fanny and co-authors

**Ice cliff contribution to the tongue-wide ablation of Changri Nup Glacier, Nepal, Central Himalaya**

Fanny Brun[1,2], Patrick Wagnon[1,3], Etienne Berthier[2], Joseph M. Shea[3,4,5], Walter W. Immerzeel[6], Philip D.A. Kraaijenbrink[6], Christian Vincent[1], Camille Reverchon[1], Dibas Shrestha[7], and Yves Arnaud[1]

[1]Univ. Grenoble Alpes, CNRS, IRD, Grenoble INP, IGE, F-38000 Grenoble, France
[2]LEGOS, Université de Toulouse, CNES, CNRS, IRD, UPS, F-31400 Toulouse, France
[3]International Center for Integrated Mountain Development, Kathmandu, Nepal
[4]Center for Hydrology, University of Saskatchewan, Saskatoon, Canada
[5]Geography Program, University of Northern British Columbia, Prince George, Canada
[6]Department of Physical Geography, Faculty of Geosciences, Utrecht University, Utrecht, the Netherlands
[7]Central Department of Hydrology and Meteorology, Tribhuvan University, Kathmandu, Nepal

**Correspondence:** Fanny Brun (fanny.brun@univ-grenoble-alpes.fr)

**Abstract.** Ice cliff backwasting on debris-covered glaciers is recognized as an important  mass loss process that is potentially responsible for the so-called "debris-cover anomaly", i.e. the fact that debris-covered and debris-free glacier tongues appear to have similar thinning rates in Himalaya. In this study, we  quantify the total contribution of ice cliff backwasting to the net ablation of the tongue of  Changri Nup Glacier

5    , Nepal, between 2015 and 2017. Detailed backwasting and surface thinning rates were obtained from terrestrial photogrammetry collected in November 2015 and 2016,  unmanned air vehicle (UAV) surveys conducted in November 2015, 2016 and 2017, and Pléiades tri-stereo imagery obtained in November 2015,  2016, and  2017.

10   UAV- and Pléiades-derived ice cliff volume loss estimates were, respectively, 3  and 7%  less than the value calculated from the reference terrestrial photogrammetry. Ice cliffs cover between 7  of the total map view area  of the Changri Nup tongue. Yet from November

15  2015 to November 2016 (November 2016 to November 2017), ice cliffs contributed to 23  +/- 5% ( 24  +/- 5%) of the total  ablation observed on the tongue. Ice cliffs therefore have a net ablation rate 3.1 ± 0.6 (resp. 3.0 ± 0.6) times higher than the average glacier tongue surface. However, on Changri Nup Glacier, ice cliffs still cannot compensate for the reduction of ablation due to debris-cover.  In addition to cliffs enhancement, a combination of reduced ablation and lower emergence velocities  could be responsible for the  debris covered anomaly on debris-covered tongues.

**1 Introduction**

Ablation areas in High Mountain Asia (HMA) are heavily debris-covered, meaning that a potentially large part of melt water originates from ice ablation of debris-covered glacier tongues (Kraaijenbrink et al., 2017). Numerous studies have demonstrated that a debris layer thicker than 5–10 cm has a dominant insulating effect and dampens the ablation of ice beneath it (e.g., Østrem, 1959; Nicholson and Benn, 2006; Reid and Brock, 2010; Reznichenko et al., 2010; Lejeune et al., 2013). Yet counter-intuitively, similar thinning rates (change in glacier surface elevation over time) have been observed for clean ice and debris-covered ice at similar elevations across HMA (Gardelle et al., 2013; Kääb et al., 2012). This 'debris-cover anomaly' (Pelicciotti et al., 2015) has been observed in the Khumbu region (Nuimura et al., 2012), the Kangri Karpo Mountains (Wu et al., 2018),  and at Kanchenjunga (Lamsal et al., 2017) and Siachen (Agarwal et al., 2017) glaciers.

Two main hypotheses have been proposed to explain this anomaly. First, while ablation rates are reduced by thick debris, ice cliffs  act as local hot spots for melt and thus could contribute disproportionately to the tongue-averaged ablation (Sakai et al., 1998, 2002; Reid and Brock, 2014; Immerzeel et al., 2014; Pellicciotti et al., 2015; Steiner et al., 2015; Buri et al., 2016a). Additionally, other processes linked to supraglacial and englacial water systems could lead to substantial ablation  (e.g., Sakai et al., 2000; Miles et al., 2016, 2018; Benn et al., 2017; Watson et al., 2018). Second, debris-covered tongues  likely have a lower emergence velocity compared with debris-free tongues (Anderson and Anderson, 2016; Banerjee, 2017). As a result,  similar thinning rates (surface mass balance rate minus emergence velocity)  could potentially be observed for debris-covered and clean ice glaciers, though the measured mass balance rates would be more negative for clean ice.

[revised manuscript text omitted]

30 and Pléiades imagery collected over the tongue of Changri Nup Glacier, Nepal between 2015 and 2017. From the terrestrial photogrammetry, 3D models of 12 cliffs are created to calculate reference ice cliff volume losses from 2015 to 2016. We introduce a new method to calculate ice volume losses based on DEM differencing  and geometric changes (e.g. ice emergence) induced by glacier flow. The new method is validated with  terrestrial photogrammetric estimates of ice

cliff volume loss and applied to the entire Changri Nup Glacier tongue  to estimate the fraction of  tongue-wide net ablation due to ice cliffs.

**2 Study area**

This study focuses on the debris-covered part of the tongue of the Changri Nup Glacier,  located in the Everest region of Nepal (Fig. 1). The glacier accumulates mass partly through avalanche deposition from the surrounding steep slopes (up to ∼6700 m a.s.l.) and flows down to 5250 m a.s.l. The local equilibrium line altitude (ELA)  calculated for the nearby debris-free West Changri Nup Glacier is approximately 5600 m (Sherpa et al., 2017). We use the same glacier tongue outline as Vincent et al. (2016), which was derived from a combination of UAV imagery,  velocities measured on the ground and field observations. This outline is substantially different from the outline available in the Randolph Glacier Inventory 6.0 (Pfeffer et al., 2014), which  erroneously connects the debris-covered Changri Nup Glacier and the debris-free West Changri Nup Glacier .

Debris covers an area of $1.49 \pm 0.16$ km$^2$ (Fig. 1)  on the tongue of Changri Nup Glacier. Twelve ice cliffs were ground-surveyed (Table 1 and Fig. 1),  and the analysis was then extended to more than 140 ice cliffs of various sizes (Fig. 1). The map view area of  all ice cliffs was $70 \pm 14 \times 10^3$ m$^2$, $72 \pm 14 \times 10^3$ m$^2$ and $70 \pm 14 \times 10^3$ m$^2$ in November 2015, in November 2016 and in November 2017, respectively (see section 4.4.4 for the uncertainty assessment of the cliff map view areas).

**3 Data**

**3.1 Terrestrial photogrammetry**

 Terrestrial photographs of 12 ice cliffs (Table 1) were collected during two field campaigns: 24–28 November 2015 and 9–12 November  2016, using survey methods similar to those described in Brun et al. (2016) and Watson et al. (2017). Between 200 and 400 photographs of each ice cliff were taken from various camera positions using a Canon EOS5D Mark II digital reflex camera with a Canon 50 mm f/2.8 fixed focal length lens (Vincent et al., 2016). For each ice cliff, we  derived point clouds (PCs) and triangulated irregular networks (TINs) with Agisoft Photoscan 1.3.4  Professional (Agisoft, 2017). In order to align the photographs and georeference the final point clouds and derived products, between 7 and 17 ground control points (GCPs) made of pink fabric were spread around each cliff. GCP positions were surveyed with a Topcon differential global positioning system (DGPS) unit with a precision of ∼  0.10 m. All markers were used as GCPs and therefore no independent markers were available for validation. After optimization of the photographs alignment, the marker residuals were on average  0.27 m for the 2015 campaign and  0.18 m for the 2016 campaign. The 3D area of the surveyed cliffs ranged from 600 m$^2$ to more than 11 000 m$^2$ (Table 1).

**3.2 UAV photogrammetry**

UAV imagery of Changri Nup Glacier was obtained in November 2015, November 2016, and November 2017 using the Sony Cyber-shot WX DSC-WX220 mounted on the fixed-wing eBee UAV manufactured by senseFly (Table 2). Aerial imagery was processed using a Structure from Motion (SfM) procedure in Agisoft Photoscan (see Vincent et al. (2016) and Kraaijenbrink et al. (2016) for details) to produce dense point clouds. Orthomosaics (0.10 m resolution) and DEMs (0.20 m resolution) were produced for each year. Additional mission and processing details for each year are given below.

In 2015, five separate eBee flights between 22–24 November were flown to cover the surface of the glacier. The data were georeferenced using a set of 24 GCPs that were spatially well-distributed and measured using the Topcon DGPS (Fig. S1). Based on 10 additional independent GCPs, the error of the 2015 UAV product was determined to be 0.04 m horizontal and 0.10 m vertical, which is in the range of expected accuracy (Gindraux et al., 2017).

On 10 November 2016, Changri Nup was surveyed with three eBee flights. To georeference the 2016 UAV imagery, we distributed a total of 17 markers on the glacier and measured their coordinates with the Topcon DGPS. Unfortunately, due to time constraints, the resulting spatial distribution of the markers was suboptimal (Fig. S1). Using only these markers as GCPs had considerable consequences for processing accuracy, and we therefore defined 16 additional virtual tie points. Tie point coordinates were sampled from the November 2015 UAV orthomosaic and DEM (Fig. S1), and we selected specific features on boulders that were (a) clearly identifiable on both the 2015 and 2016 image sets, and (b) located on stable terrain (Immerzeel et al., 2014), which we determined from visual inspection of the Pléiades orthoimages and DEMs.

In 2017, three separate flights were used to survey the glacier on 23 November, and 30 GCPs were collected (Fig. S1). Residuals, based on 6 independent check points, were 0.10 m in horizontal and 0.14 m in vertical.

**3.3 Pléiades tri-stereo photogrammetry**

[revised manuscript text omitted]

Assuming  that cross-sectional area ($\sigma_S$) and the depth-averaged velocity ($\sigma_{\bar{u}}$)  are independent, uncertainty in the ice flux ($\sigma_\Phi$) can be estimated as:

$$\frac{\sigma_\Phi}{\Phi} = \sqrt{\frac{\sigma_{\bar{u}}^{2}}{\bar{u}} + \frac{\sigma_S^{2}}{S}} \qquad (4)$$

Given the above mentioned values for the depth-averaged velocity, the cross-sectional area and the associated uncertainties, the relative uncertainty  in the estimated ice flux is $\sim$30 %. As a result, for the  November 2015–November 2016  and November 2016–November 2017  periods, the ice flux was  499 700 ± 150 000 m$^3$ a$^{-1}$  and 503 840 ± 150 000 m$^3$ a$^{-1}$  respectively. 
[revised manuscript text omitted]
 2016.  This results in calculated values of $f_C$  = 4.5 ± 0.6 (and $f_C^*$  = 5.4 ± 0.7), which is 50 % higher than the actual value.  Ice cliffs would thus contribute to ∼34 % of the total tongue ablation. For the period November 2016–November 2017, the  same assumption results in $f_C$  = 3.6 ± 0.6 (and $f_C^*$  = 4.3 ± 0.7),  and an ice cliff contribution of ∼29 %  to the total tongue ablation.  Neglecting

30   $w_e$ might partially explain why previous studies found significantly higher values of $f_C$, and  our results stress need to estimate and take into account  ice flow emergence, even for  nearly-stagnant glacier tongues like Changri Nup Glacier (see Discussion below).

[revised manuscript text omitted]

Consequently, the qualitative picture we can draw is that the ablation area of glaciers with considerable debris-cover is usually larger than for debris-free glaciers ($A_{DC} > A_{DF}$). This results in lower emergence velocities ($w_{e,DC} = \Phi/A_{DC} < \Phi/A_{DF} = w_{e,DF}$). If the glacier is in mass and dynamical equilibrium, in both debris-covered and debris-free cases, the thinning rate at any elevation is 0, because the emergence velocity compensates the surface mass balance. However, both $w_e$ and $\dot{b}$ will be lower for the debris-covered tongue (Fig. 11). In an unbalanced regime with consistent negative mass balances, as mostly observed in High Mountain Asia (Brun et al., 2017), similar thinning rates between debris-free and debris-covered tongues could be the combination of reduced emergence velocities and lower ablation for debris-covered glaciers (Fig. 11). Evidence for reduced debris-covered glacier velocities and loss of connectivity between accumulation and ablation areas (Neckel et al., 2017) will lead to further reductions in both ice fluxes and $w_e$.

In conclusion, our field evidence shows that enhanced ice cliff ablation alone could not lead to a similar level of ablation for debris-covered and debris-free tongues. While other processes can substantially increase the ablation of debris-covered tongues, we highlight the potentially important role of the neglected emergence velocity in the explanation of the so-called 'debris-cover anomaly'.

**6.4 Applicability to other glaciers**

Determining the total ice cliff contribution to the net ablation of the tongue (i.e., the $f_C$ factor defined in this study) of a single glacier has limited value by itself, because we do not know the variability between glaciers. In particular, it is too early to conclude if the range of $f_C$ values reported in the literature reflects inconsistencies amongst the different methods, or is actually a reflection of variability between glaciers. For instance, model-based $f_C$ values (Sakai et al., 1998; Juen et al., 2014; Buri et al., 2016b; Reid and Brock, 2014) are not directly comparable with the observations (Brun et al., 2016; Thompson et al., 2016), because they usually require additional assumptions about e.g., the sub-debris ablation or emergence velocity. The definition of debris-covered tongues, the nature of their surface, and their hypsometry might also have a considerable effect on $f_C$.

A significant obstacle to applying our method to other glaciers is the need to estimate the emergence velocity, which requires an accurate determination of the ice fluxes entering the glacier tongues. The measurement of ice thickness with GPR systems is already challenging for debris-free glaciers, as it requires transmitter, receiver and antennaes must be pulled along transects on the glacier surface. It is even more challenging for debris-covered glaciers, as the hummocky surface prevents the operators from dragging a sledge. More field campaigns dedicated to ice thickness and velocity measurements (Nuimura et al., 2011, 2017) or the development of airborne ice thickness retrievals through debris are needed, as

5   stressed by the outcome of the Ice Thickness Models Intercomparison eXperiment (Farinotti et al., 2017). The precise retrieval of emergence velocity pattern using a network of ablation stakes combined with DGPS is a promising alternative, in particular if combined with detailed ice flow modeling (e.g., Gilbert et al., 2016).

**7   Conclusions**

In this study, we estimate the total contribution of ice cliff to the total net ablation of a debris-covered glacier tongue for

10   two consecutive years, taking into account the emergence velocity. Ice cliffs are responsible for 23-24 $\pm$ 5 % of the total net ablation for both years, despite a tongue-wide net ablation approximately 25 % higher in the second year. On Changri Nup Glacier, the fraction of total net ablation from ice cliffs is too low to explain the so-called "debris-cover anomaly". Other contributions, such as ablation from supra-glacial lakes, or along englacial conduits, are potentially large and have yet to be quantified. For the specific case of Changri Nup Glacier they are likely not large enough to compensate for

15   the reduced ablation (Vincent et al., 2016). Consequently, we hypothesize that the "debris-cover anomaly" could be a result of lower emergence velocities and reduced ablation, which leads to *thinning rates* comparable to those observed on clean ice glaciers. However, ice cliffs are still hot-spots of ablation and consequently of enhanced thinning; without them, the thinning rates of debris-covered and clean ice might not be similar.

Our method requires high-resolution UAV or satellite stereo imagery, and is restricted to glaciers where thickness estimates at

20   a cross section upstream of the debris-covered tongue are available, and emergence velocity can be estimated. A comparison of cliff ablation enhancement factor ($f_C$ or $f_C^*$) values calculated for other debris-covered glaciers under our suggested framework would be informative, in order to compare estimates of ice cliff ablation for other and potentially much larger debris-covered tongues. Though our results cover only two years of data where net ablation totals differed by 25%, the area occupied by ice cliffs and their relative contribution to ablation ($f_C$) remained almost constant.

25   A main limitation of our study is its short spatial and temporal extent, and it would be worthwhile to obtain longer-term estimates of the relative ice-cliff contribution to net ablation at multiple sites. These estimates would lead to the development of empirical relationships for cliff enhanced ablation that could be included in debris-covered glacier mass balance models.

In line with a previous study (Vincent et al., 2016), we stress the need for more research about the emergence velocity of debris-covered (and nearby debris-free) tongues, as the assumption that thinning rates are equal to net ablation rates is incorrect, and can lead to inaccurate conclusions. 
[revised manuscript text omitted]

|---|---|---|---|
| 22/11/2015 | 0.36;0.26;0.10 | -4.3 | 0.3 |
| 13/11/2016 | 0.47;0.28; 0.20 | 6.6 | 3.7 |
| 24/10/2017 | 0.34;0.25;0.09 | 1.0 | 4.2 |